   

THE EMBO JOURNAL

# TDP1 phosphorylation by CDK1 in mitosis promotes MUS81-dependent repair of trapped Top1-DNA covalent complexes

Srijita Paul Chowdhuri & Benu Brata Das 🆔 ✉

## Abstract

Topoisomerase 1 (Top1) controls DNA topology, relieves DNA supercoiling during replication and transcription, and is critical for mitotic progression to the G1 phase. Tyrosyl-DNA phosphodiesterase 1 (TDP1) mediates the removal of trapped Top1-DNA covalent complexes (Top1cc). Here, we identify CDK1-dependent phosphorylation of TDP1 at residue S61 during mitosis. A TDP1 variant defective for S61 phosphorylation (TDP1-S61A) is trapped on the mitotic chromosomes, triggering DNA damage and mitotic defects. Moreover, we show that Top1cc repair in mitosis occurs via a MUS81-dependent DNA repair mechanism. Replication stress induced by camptothecin or aphidicolin leads to TDP1-S61A enrichment at common fragile sites, which over-stimulates MUS81-dependent chromatid breaks, anaphase bridges, and micronuclei, ultimately culminating in the formation of 53BP1 nuclear bodies during G1 phase. Our findings provide new insights into the cell cycle-dependent regulation of TDP1 dynamics for the repair of trapped Top1-DNA covalent complexes during mitosis that prevents genomic instability following replication stress.

**Keywords** Topoisomerase 1; TDP1; MUS81; MiDAS; CDK1
**Subject Categories** Cell Cycle; DNA Replication, Recombination & Repair

## Introduction

To ensure faithful chromosome segregation, mammalian cells need to complete DNA replication in S-phase prior to their entry into mitosis (Shiloh, 2003). Incompletely replicated, or unresolved, chromosomes from the S-phase can often persist into mitosis, where they present a potential threat to the faithful segregation of sister chromatids (Mankouri et al, 2013). Indeed, it is becoming increasingly apparent that the transition from S-phase to M-phase perhaps encounters a less stringent checkpoint, and cells frequently enter mitosis with under-replicated or unrepaired chromosomes following replication stress (RS) (Belotserkovskaya and Jackson,

2014; Blackford and Stucki, 2020; Mankouri et al, 2013; Minocherhomji et al, 2015b; Orthwein et al, 2014). Certain regions of the human genome are intrinsically difficult to replicate, and they are particularly susceptible to RS, like common fragile sites (CFSs), which are prone to form gaps and breaks on metaphase chromosomes. The expression of late replicating CFS on metaphase chromosomes suggests that these sites also fail to complete DNA replication in the S- and G2 phase or suffer breakages that are carried over to mitosis (Chan et al, 2009). Defects in mitotic chromosomal segregation can lead to abnormal structures such as micronuclei or chromatin bridges, which are prevalent in human cancers. Topoisomerase 1 (Top1) has been associated to CFS stability, and Top1 deficiency increases DNA breaks at CFS loci (Arlt and Glover, 2010; Pladevall-Morera et al, 2019).

Top1 is essential for the release of DNA supercoiling during replication and transcription and faithful segregation of chromosomes (Capranico et al, 2017; Champoux, 2001; Pommier et al, 2016). Top1 generates transient and reversible Top1-linked DNA single-strand breaks (SSBs) (Top1 cleavage complexes; Top1cc) during catalysis, which can be preferentially trapped by the anticancer drug camptothecin (CPT) and its clinical derivatives topotecan and irinotecan (Chowdhuri et al, 2023; Pommier, 2006). Abortive Top1ccs are converted to DNA double-strand breaks (DSBs) upon replication and transcription collisions (Pommier, 2006), which triggers cell cycle arrest and cell death (Das et al, 2016). Tyrosyl-DNA Phosphodiesterase 1 (TDP1) typically hydrolyzes the phosphodiester bond between DNA 3′-end and the tyrosyl moiety of Top1 to repair Top1cc (Ashour et al, 2015; Bhattacharjee et al, 2022b; Kawale and Povirk, 2018; Pommier et al, 2014). Accordingly, genetic inactivation of TDP1 causes hypersensitivity to CPT, while homozygous mutation of TDP1 causes the neurodegenerative syndrome, spinocerebellar ataxia with axonal neuropathy (SCAN1) resulting from elevated levels of Top1cc in post-mitotic neurons (Das et al, 2010; Huang et al, 2013; Interthal et al, 2005; Interthal et al, 2001; Katyal et al, 2007; Murai et al, 2012; Pommier et al, 2014; Takashima et al, 2002; Vance and Wilson, 2002). The phosphodiesterase activity of TDP1 also repairs other blocked 3′-DNA lesions, including oxidative DNA damage and ionizing radiations (IR) (Huang et al, 2013; Katyal et al, 2007; McKinnon and Caldecott, 2007; Murai et al, 2012). The N-terminal region of TDP1 spanning 1–148 amino acids is not required for

Laboratory of Molecular Biology, School of Biological Sciences, Indian Association for the Cultivation of Science, 2A & B, Raja S. C. Mullick Road, Jadavpur, Kolkata, West Bengal 700032, India. ✉E-mail: pcbbd@iacs.res.in

in vitro catalytic activity of TDP1 (Interthal et al, 2001) however, plays a critical role in subcellular localization, turnover, stability, recruitment of TDP1 at DNA damage sites, and interactions with its repair partners such as PARP1, XRCC1, and Ligase III in response to Top1cc-induced DNA damage (Bhattacharjee et al, 2022a; Bhattacharjee et al, 2022b; Das et al, 2009; Das et al, 2014; El-Khamisy et al, 2005; Hudson et al, 2012; Kawale and Povirk, 2018; Pommier et al, 2014; Rehman et al, 2018).

Top1 undergoes cell cycle-specific phosphorylation that regulates its activity during mitotic transcription to remove super-coiling that allows completion of transcription during prometaphase and reloading of RNAPII at promoters during mitotic exit, facilitating the progression into G1 (Wiegard et al, 2021). Intriguingly, cells deficient for p53 when exposed to CPT undergo mitotic catastrophe (Tse and Schwartz, 2004), suggesting the failure to activate the G2/M checkpoint and the entry of cells with unrepaired Top1cc into mitosis. However, whether TDP1 repairs mitotic Top1cc remains unknown.

Human cells have developed a strategy for "unscheduled" DNA synthesis (termed MiDAS), which is a break-induced repair (BIR) mechanism that operates in early mitosis to rescue under-replicated loci, and involves POLD3, RAD52, and MUS81 as key players (Bhowmick et al, 2023; Bhowmick et al, 2016; Macheret et al, 2020; Malkova and Ira, 2013; Minocherhomji et al, 2015a). MUS81-EME1 is a structure-specific endonuclease that cleaves late replication intermediates at CFSs during early mitosis to trigger DNA repair synthesis that ensures faithful chromosome segregation (Bhowmick et al, 2023; Bhowmick et al, 2016; Calzetta et al, 2020; Di Marco et al, 2017; Minocherhomji et al, 2015a). Several alternative pathways exist in cells to ameliorate the deleterious effects of trapped Top1cc-induced replication stress (Zhang et al, 2022). MUS81 can also resolve 3'-flap structures, generated from Top1cc at the stalled replication (Regairaz et al, 2011; Wu and Wang, 2021). However, it is still unclear how trapped Top1cc on the mitotic chromosome are repaired.

Cyclin-dependent kinase 1 (CDK1) is a serine/threonine kinase that phosphorylates an array of target proteins and play key roles in coordinating DNA repair with cell cycle transitions (Brown et al, 2015; Diril et al, 2012; Holt et al, 2009). Defects in CDK1 activation lead to cell cycle progression beyond metaphase and deleterious late mitotic events. Intriguingly, CDK1-mediated phosphorylation of SLX4 causes folding of the SAP domain, which facilitates its binding with MUS81 and stimulates robust cleavage of DNA replication and recombination structures during mitosis. (Palma et al, 2018; Payliss et al, 2022). Human Top1 also undergoes mitotic phosphorylation by CDK1, although the functional implication of the phosphorylation remains unclear (Hackbarth et al, 2008). However, unscheduled activation of CDK1 leads to mitotic catastrophe in cells deficient for checkpoint after CPT treatment (Szmyd et al, 2019).

Post-translational modifications (PTMs) of TDP1 are critical part of the DNA damage response that accounts for the subcellular localization, turnover, stability, recruitment and modulation of catalytic activity of TDP1 at DNA damage sites (Bhattacharjee et al, 2022a; Bhattacharjee et al, 2022b; Das et al, 2009; Das et al, 2014; Hudson et al, 2012; Kawale and Povirk, 2018; Pommier et al, 2014; Rehman et al, 2018). However, there are no evidence for mitotic regulation of TDP1. Here we report the functional coupling of CDK1 with TDP1 that facilitates phosphorylation of TDP1 at S61

residue in G2/M boundary. TDP1-S61 phosphorylation increased during early mitosis, then declined in telophase to near basal levels in G1. Our results represent the first demonstration of a cell cycle-dependent regulatory phosphorylation of TDP1 at S61 in human cells and its implications in maintaining chromosomal stability.

## Results

### TDP1 is phosphorylated at serine 61 during mitosis

To identify new post-translational modifications of TDP1, we immunoprecipitated ectopic FLAG-TDP1 from cells grown in the presence or absence of CPT and analyzed them by mass spectrometry (MS). This MS analysis of FLAG-TDP1 detected S61 as a phosphorylated residue on TDP1 (Fig. 1A; Appendix Fig. S1A), which was identified as independent of CPT-induced DNA damage. The S61 residue of human TDP1 is phylogenetically conserved across vertebrate species as a proline-directed phosphorylation site (Fig. 1A) and lies within a conserved motif, which is the preferred substrate for Cyclin-dependent kinase 1 (CDK1) (Dephoure et al, 2008). Because CDK1 is a mitotic regulatory kinase, we hypothesized TDP1-S61 phosphorylation as a mitotic event. To that effect, we generated a phospho-specific antibody that recognizes the epitope HKRKI(S*)PVKFSN (the asterisk denotes phosphorylation) spanning the S61 residue of TDP1 and tested its ability to detect immunoprecipitated ectopic human FLAG-TDP1 complexes from MCF7 cells synchronized to mitosis by thymidine and nocodazole (Noc) treatment as outlined in the protocol (Fig. 1B). Figure 1C shows that the pS61-TDP1 antibody recognizes a single band with a molecular weight similar to that of the FLAG-TDP1 immunoprecipitated from cells synchronized to mitotic phase only (Fig. 1C; panel + Noc). Furthermore, we were able to abolish the pS61-TDP1 signal on TDP1 in mitotic cells by applying broad-spectrum phosphatase ($\lambda$-phosphatase) prior to immunoprecipitation (Appendix Fig. S1B). The antibody specificity for phosphorylated S61 residue on TDP1 was validated using FLAG-TDP1 mutant version (S61A) that failed to detect the pS61-TDP1 signal immunoprecipitated from the mitotic cells (Fig. 1C). Knockdown of TDP1 by siRNA abrogated the endogenous pS61-TDP1 signal in MCF7 cells synchronized to mitotic phase (Fig. 1D), confirming the specificity of the antibody for pS61 of TDP1. We further confirmed mitotic phosphorylation of TDP1-S61 by using a pan-anti-MPM2 antibody that recognizes epitopes in proteins phosphorylated during mitosis (Kuang et al, 1989). Notably, the phosphorylation signal on TDP1 was abrogated in the immunoprecipitated FLAG-TDP1^S61A, indicating S61 as the targeted residue for the mitotic phosphorylation (Appendix Fig. S1C; phosphorylated TDP1).

In order to monitor the temporal kinetics of TDP1 phosphorylation at S61, we pulled down FLAG-TDP1 variants (WT and S61A) ectopically expressing in MCF7 cells synchronized to mitosis with thymidine-nocodazole (Fig. 1B) and monitored the phosphorylation of this site after release from the noc-arrest at indicated time points in early mitotic (0 h), late mitotic (1 h), and G1 (5 h) phases (see FACS profile; Fig. 1E) using anti-pS61-TDP1 (Fig. 1F) and anti-MPM2 antibody (Appendix Fig. S1D). We found that TDP1-S61 phosphorylation peaked upon release from early mitosis before declining in late mitosis to return to near basal levels in G1 (Fig. 1F; Appendix Fig. S1D), which mirrored the CDK1 kinase activity

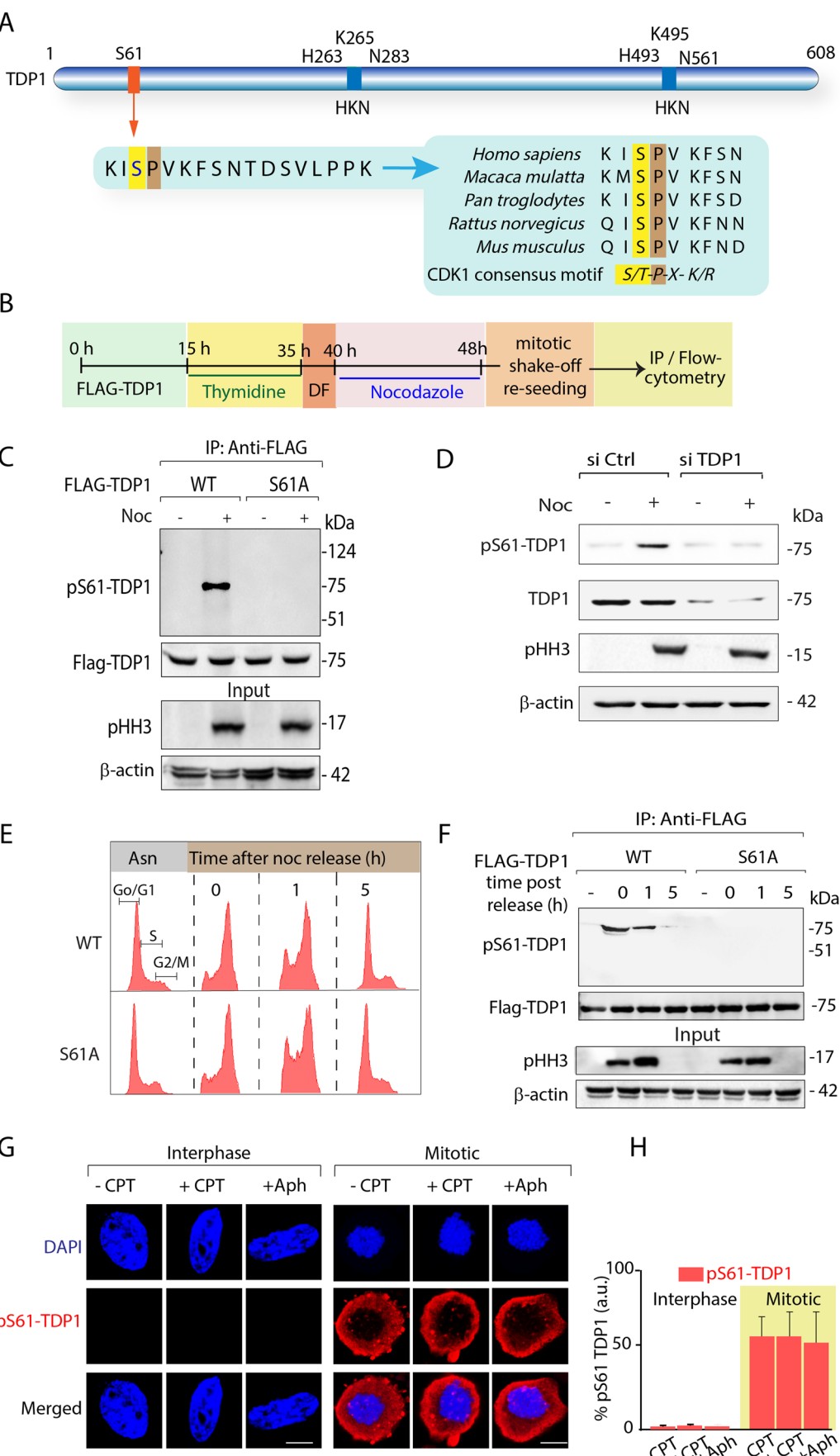

**Figure 1.  Human TDP1 is phosphorylated at S61 during early mitosis.**

(A) Schematic representation of human TDP1 showing the serine phosphorylation site (S61) and the catalytic residues (HKN motifs). The sequence alignment of amino acids of TDP1 in the region flanking the S61 phosphorylation site (a conserved CDK1 substrate phosphorylation site S/T-P-X-K/R is highlighted) from human (*Homo sapiens*), monkey (*Macaca mulatta*), chimpanzee (*Pan troglodytes*), rat (*Rattus norvegicus*), mouse (*Mus musculus*) shows the phylogenetic conservation of S61. (B) Schematic representation for the protocol followed for synchronization of MCF7 cells expressing ectopic FLAG-TDP1 variants (WT or S61A) to mitotic phase for immunoprecipitation or PI-RNase based flow cytometry analysis (DF, drug free). (C) Ectopic FLAG-TDP1^WT or FLAG-TDP1^S61A in MCF7 cells were left asynchronous or synchronized to mitosis, following immunoprecipitation with anti-FLAG antibody, the immune complexes were analyzed by western blotting. Note: The TDP1-S61 phospho-specific antibody (pS61-TDP1) recognizes a single band with a molecular weight corresponding to TDP1 in the mitotic phase only. The same blot was stripped and reprobed with anti-FLAG antibody (FLAG-TDP1). Aliquots (10%) of the input were probed with phospho-histone H3 at Ser10 (anti-pHH3) to indicate mitotic phase and β-actin as loading control. (D) MCF7 cells were transfected with Si Ctrl or Si TDP1 for 72 h, left asynchronous or synchronized to mitosis with nocodazole treatment, and western blotting was performed using anti-pS61-TDP1 antibody. The level of TDP1 knockdown was confirmed by anti-TDP1 antibody with β-actin as a loading control. (E) Flow cytometry profile of the MCF7 cells expressing ectopic FLAG-TDP1 variants (WT and S61A), synchronized to the mitosis phase, and harvested at indicated time points after release from noc. (F) Following mitotic synchronization, FLAG-TDP1 variants were immunoprecipitated at the indicated time points [0, 1 and 5 h post release in drug-free media from nocodazole (Noc) block] using the anti-FLAG antibody, and immune complexes were blotted with the anti-pS61-TDP1 antibody. The same blot was stripped and reprobed with anti-FLAG antibody. Aliquots (10%) of the input indicate the mitotic marker phospho-histone H3 (anti-pHH3) and β-actin as loading control. (G) Immunolocalization of endogenous pS61-TDP1 (red) in MCF7 cells treated with or without CPT (15 nM, 24 h) or aphidicolin (0.4 μM, 24 h) synchronized to interphase or mitosis and detected with anti-pS61-TDP1 antibody. Cells at interphase and mitosis were tallied on the basis of their chromatin morphology, as indicated by DAPI staining (blue). (H) Bar graph showing the intensity of pS61-TDP1 for interphase and mitotic cells following RS with CPT or APH. Intensities from 40 nuclei per stage were expressed as mean ± SD. $n = 3$ biological replicates. a.u. arbitrary unit. Scale bars, 10 μm. Source data are available online for this figure.

(Appendix Fig. S1K) (Brown et al, 2015; Diril et al, 2012; Holt et al, 2009). This is congruent with our hypothesis that CDK1 might be the kinase responsible for phosphorylating TDP1 at S61. We further confirmed that FLAG-TDP1^S61A expression in MCF7 cells did not perturb the cell cycle progression (Fig. 1E).

Next, to follow the subcellular distribution of endogenous pS61-TDP1, we performed immunocytochemistry using the phospho-peptide antibody (pS61-TDP1) in MCF7 cells at interphase or mitosis in the absence or presence of RS induced by CPT or aphidicolin (APH) (Minocherhomji et al, 2015b). Figure 1G–H; Appendix Fig. S1E,F,S1J show that the induction of pS61-TDP1 was detected only in the mitotic cells and was not detected in the interphase cells, consistent with western blotting analysis (Appendix Fig. S1G–I and Fig. 1F; panel +5 h). Furthermore, the subcellular distribution of pS61-TDP1 was shown to be identical regardless of RS generated by CPT or APH (Fig. 1G,H). Notably, pS61-TDP1 localization was excluded from the mitotic chromosomes and showed cytosolic distribution in the M-phase (Fig. 1G and see the field image in Appendix Fig. S1E). Together, we conclude TDP1-S61 is phosphorylated in mitosis, and pS61-TDP1 cellular distribution indicates an inverse correlation with TDP1 enrichment on mitotic chromosomes.

## CDK1 binds with TDP1 to phosphorylate at S61

CDK1 plays a key role in the regulation of the mitotic DNA damage response (Brown et al, 2015; Diril et al, 2012; Holt et al, 2009) and it has been shown to phosphorylate Top1 (Hackbarth et al, 2008). This prompted us to test whether CDK1 interacts with TDP1. Co-immunoprecipitation (co-IP) of ectopically expressed FLAG-TDP1 in MCF7 cells pulled down endogenous CDK1 (Fig. 2A) both in the presence as well as in the absence of CPT, indicating TDP1-CDK1 binding is independent of DNA damage. Further, we co-immunoprecipitated endogenous TDP1 from MCF7 cells and confirmed the association between endogenous TDP1-CDK1 in cells (Fig. 2B). To test whether CDK1 interacts with TDP1, we performed co-IP with FLAG-TDP1 in the presence of the benzonase nuclease (Fig. 2C) (Rehman et al, 2018). We found that

the TDP1-CDK1 association was resistant to benzonase, indicating a protein-protein interaction, not mediated through DNA or RNA (Fig. 2C). We also observed similar levels of endogenous CDK1 in the immune complexes of both FLAG-TDP1 variants (WT and S61A) confirming the dispensable nature of pS61-TDP1 in mediating CDK1 and TDP1 interaction (Fig. 2D). We further established the presence of TDP1 in the CDK1-complex using reverse co-IP in cells ectopically expressing HA-tagged CDK1 (Fig. 2E), confirming the specific association between TDP1 and CDK1. In addition, we co-immunoprecipitated endogenous CDK1 from MCF7 cells and confirmed its association TDP1 in cells (Fig. 2F). Further, the interaction between CDK1 and TDP1 was not abrogated in the presence of the CDK1 inhibitor (RO3306), suggesting CDK1-TDP1 binding is independent of CDK1 catalytic activity (Fig. 2A,E,F).

Next, to determine whether CDK1 is involved in TDP1 phosphorylation at S61, we ectopically expressed and immunoprecipitated FLAG-TDP1^WT from CDK1 knockdown cells synchronized to the mitotic phase. Figure 2G shows that CDK1 depletion resulted in a marked reduction in pS61-TDP1, suggesting that TDP1 not only physically interacts with CDK1 (Fig. 2A–F) but is also phosphorylated at S61 in vivo by CDK1 during the mitotic phase (Fig. 2G). Further inhibition of CDK1 catalytic activity by treatment with RO3306, resulted in a marked reduction in pS61-TDP1, supporting the role of CDK1 in phosphorylating TDP1 at S61 residue (Fig. 2H). To obtain further evidence for TDP1 phosphorylation at S61 by CDK1, we performed the in vitro kinase assay with recombinant 6X-His-TDP1 protein variants (WT and S61A) as substrates for immune-precipitated HA-CDK1 as the source of the kinase in the presence of ATP. The pS61-TDP1 antibody detected phosphorylation of TDP1^WT by CDK1, but not of TDP1^S61A (Fig. 2I), confirming that CDK1 phosphorylates TDP1 in vitro at S61. Under Similar conditions, we demonstrated that CDK2 failed to phosphorylate TDP1 at the S61 residue using in vitro kinase assays, confirming CDK1 as the bona fide kinase phosphorylating TDP1 at the S61 residue (Appendix Fig. S2). Taken together, these results confirmed CDK1-TDP1 physically interacts to catalyze TDP1 phosphorylation at S61 during mitosis.

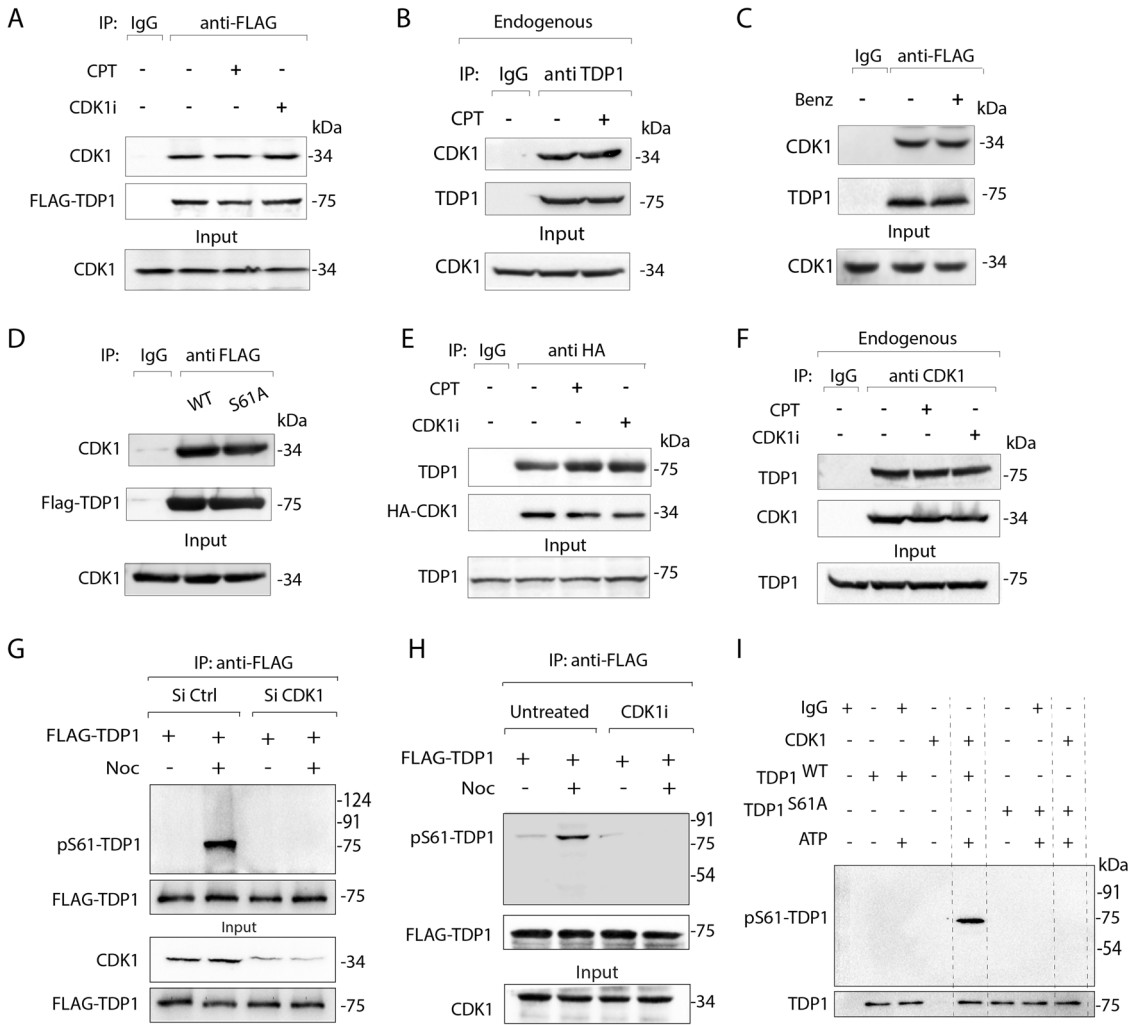

**Figure 2.  CDK1 physically interacts with TDP1 to catalyze S61 phosphorylation.**

(A) MCF7 cells ectopically expressing FLAG-TDP1 treated with or without CPT (5 μM, 3 h) or CDK1i (9 μM, 16 h) were co-immunoprecipitated (co-IP) using anti-FLAG antibody. Immune complexes were blotted with anti-CDK1 antibodies. The same blot was stripped and reprobed with anti-FLAG antibody to show the expression of FLAG-TDP1. Aliquots (10%) of the input show the level of CDK1 prior to immunoprecipitation. (B) Endogenous TDP1 was immunoprecipitated using anti-TDP1 antibody from MCF7 cells treated with or without CPT (5 μM, 3 h). Immune complexes were blotted with anti-CDK1 antibody. The same blot was stripped and reprobed with anti-TDP1 antibody to show TDP1 pull-down. Aliquots (10%) of the input show the level of CDK1 prior to immunoprecipitation. (C) Same as (A), except the cell lysates were pretreated with benzonase (nuclease) prior to co-IP as indicated. Note: The TDP1-CDK1 interaction is resistant to benzonase. (D) MCF7 cells ectopically expressing FLAG-TDP1 variants (WT and S61A) were co-immunoprecipitated (co-IP) using anti-FLAG antibody. Immune complexes were blotted with anti-CDK1 antibody. The same blot was stripped and reprobed with anti-FLAG antibody to show the expression of FLAG-TDP1 variants. Aliquots (10%) of the input show the level of CDK1 prior to immunoprecipitation. (E) MCF7 cells ectopically expressing HA-CDK1 were treated with or without CPT (5 μM, 3 h) and CDK1i (9 μM, 16 h) and immune-precipitated using anti-HA antibody. The immune complexes were blotted with anti-TDP1 antibody. The same blot was stripped and reprobed with anti-HA antibody. Aliquots (10%) of the input show the level of TDP1 prior to immunoprecipitation. (F) Endogenous CDK1 was co-IPed using anti-CDK1 antibody from MCF7 cells treated with or without CPT (5 μM, 3 h) or CDK1i (9 μM, 16 h). Immune complexes were blotted with anti-TDP1 antibody. The same blot was stripped and reprobed with anti-CDK1 antibody. Aliquots (10%) of input show TDP1. (G) FLAG-TDP1 was expressed in MCF7 cells transfected with Si CDK1 to knockdown CDK1 or Si Ctrl as indicated. Following nocodazole treatment (200 ng/ml, 8 h), ectopic FLAG-TDP1 was co-IPed using anti-FLAG antibody, and the immune complexes were blotted with pS61-TDP1 antibody. The same blot was stripped and reprobed with anti-FLAG antibody. Aliquots (10%) of the input show the level of CDK1 knockdown and FLAG-TDP1 prior to immunoprecipitation. (H) FLAG-TDP1 was expressed in MCF7 cells left untreated or treated with CDK1 inhibitor (RO3306) as indicated. Following nocodazole treatment (200 ng/ml, 8 h), ectopic FLAG-TDP1 was co-IPed using anti-FLAG antibody, and the immune complexes were blotted with pS61-TDP1 antibody. The same blot was stripped and reprobed with anti-FLAG antibody. Aliquots (10%) of the input show the level of CDK1 prior to immunoprecipitation. (I) In vitro kinase assays with HA-CDK1 immunoprecipitated from MCF7 cells in the presence of ATP. The substrates were recombinant 6xHis-tagged TDP1 variants (WT or S61A). Western blotting against the anti-TDP1 antibody shows the amount of substrate in each reaction. Protein molecular weight markers (kDa) are indicated on the right. Source data are available online for this figure.

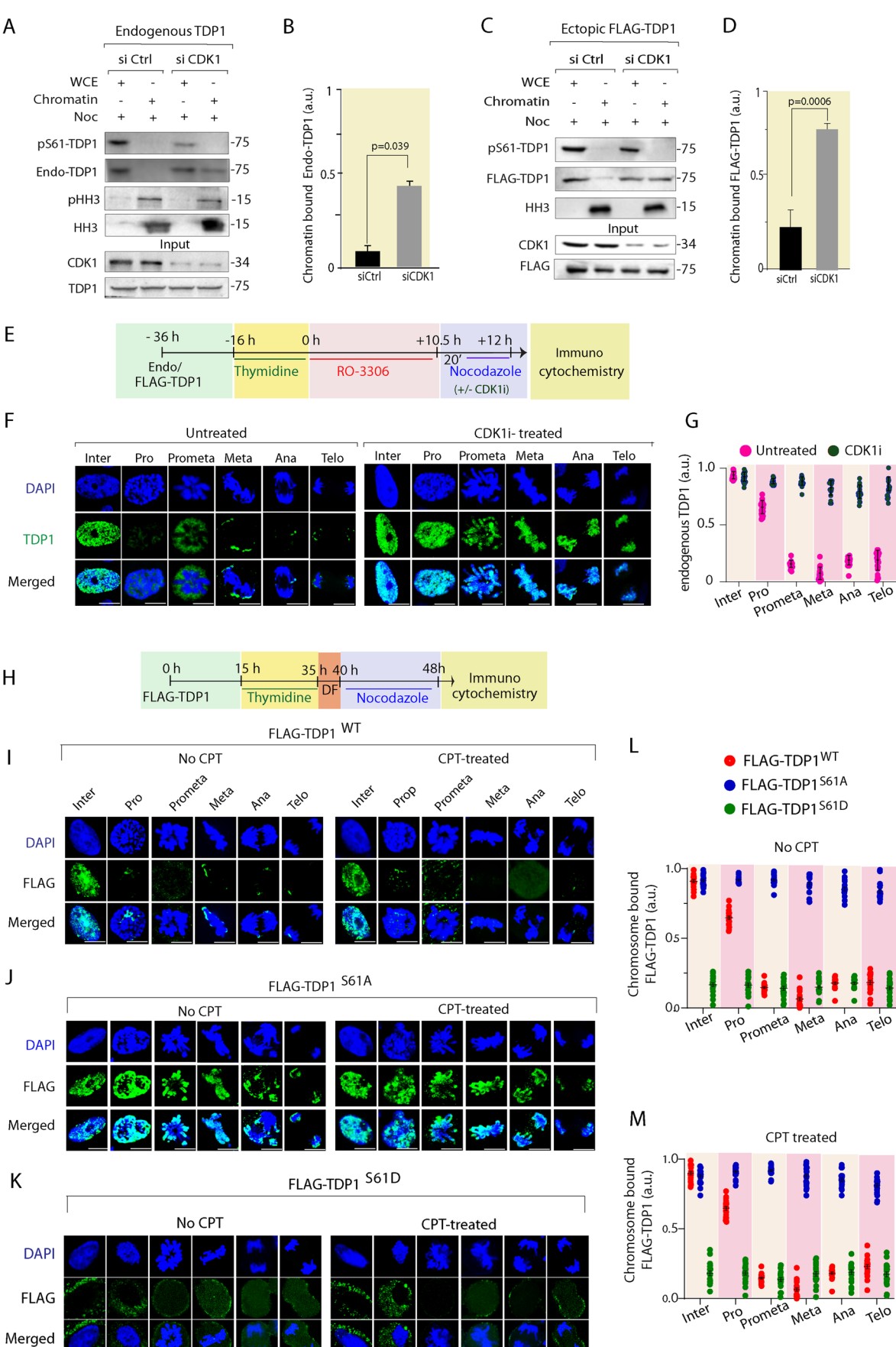

◄

**Figure 3. TDP1 phosphorylation at S61 by CDK1 abrogates chromosomal enrichment of TDP1 during mitosis.**

(A) Chromatin fractions were prepared from Noc synchronized MCF7 cells transfected with Si Ctrl or Si CDK1 and were blotted with pS61-TDP1 and anti-TDP1 antibodies. Aliquots (10%) show levels of endogenous CDK1 and TDP1 in the whole-cell (WCE) lysates prior to chromatin fractionation. (B) Densitometric analysis of chromatin-bound endogenous TDP1 in mitotic cells upon CDK1 knockdown. Data are mean ± SD, $n = 3$ biological replicates. *$P < 0.05$ (one-way ANOVA). (C) TDP1$^{-/-}$ MEFs were co-transfected with FLAG-TDP1$^{WT}$ and Si CDK1 or Si Ctrl and were synchronized with Noc. Chromatin fractions from mitotic cells were analyzed by western blotting against the anti-FLAG antibody and the pS61-TDP1 antibody. Aliquots (10%) show levels of endogenous CDK1 and ectopic FLAG-TDP1$^{WT}$ in the whole-cell (WCE) lysates prior to chromatin fractionation. (D) Densitometric analysis of chromatin-bound ectopic FLAG-TDP1 in mitotic cells upon CDK1 knockdown. Data are mean ± SD, $n = 3$ biological replicates. ***$P < 0.001$ (one-way ANOVA). (E) Schematic representation of the protocol followed to study the cell cycle stage-dependent colocalization of endogenous TDP1 with the chromosomes in the presence and absence of CDK1 inhibitor (RO3306) as indicated. (F) MCF7 cells were fixed at different time intervals after release from G2/M arrest as outlined in the protocol and stained with anti-TDP1 antibody to detect endogenous TDP1 (green) and the chromosomes by DAPI (blue). (G) The fluorescence intensity of chromosome-bound endogenous TDP1 was quantified. Staining intensities from 20 to 30 nuclei/stage were expressed as mean ± SD. a.u. arbitrary unit. (H) Schematic representation of the protocol followed to study the cell cycle stage-dependent colocalization of FLAG-TDP1 variants (WT, S61A and S61D) with the chromosomes in the presence and absence of CPT as indicated. (I–K) TDP1$^{-/-}$ MFEs complemented with FLAG-TDP1$^{WT}$ (I) or FLAG-TDP1$^{S61A}$ (J) or FLAG-TDP1$^{S61D}$ (K) were fixed after release from nocodazole arrest, and FLAG-TDP1 was immunodetected with anti-FLAG antibody (green) in the presence or absence of CPT (15 nM; 24 h) as indicated. Note: FLAG-TDP1$^{S61A}$ was readily detected on the different phases of mitotic chromosomes stained with DAPI (blue). (L, M) The fluorescence intensity of the chromosome-bound FLAG-TDP1 variants (WT or S61A or S61D) in the absence of CPT (upper panel) or the presence of CPT (bottom panel) was quantified. Staining intensities from 20 to 30 nuclei per stage were expressed as mean ± SD; a.u. arbitrary unit. All the results are expressed as mean ± SD for at least three independent experiments ($n = 3$). Scale bars, 10 μm. Source data are available online for this figure.

## S61 phosphorylation of TDP1 promotes its dissociation from mitotic chromosomes

To investigate the role of the CDK1-mediated phosphorylation of TDP1 at the S61 residue in its chromosomal recruitment and/or dissociation during mitosis, we knocked down CDK1 by siRNA in MCF7 cells and synchronized the cells to the mitotic phase with nocodazole, followed by chromatin fractionation and western blotting (Wu and Wang, 2021). A 60–70% knockdown of CDK1 was aimed for allowing cells to progress into mitosis because CDK1 deficiency prevents mitotic entry (Diril et al, 2012). We utilized two independent approaches to detect the chromatin-bound fraction of TDP1, either by western blotting with endogenous TDP1 or by analyzing the chromatin-bound fraction of ectopic FLAG-TDP1 expressed in CDK1 knockdown cells synchronized to the mitotic phase. The relative purity of the chromatin-bound fractions was further assessed by analysis of phospho-histone H3 and total histone H3, which were only detectable during mitosis in the chromatin fractions respectively (Fig. 3A). The endogenous pS61-TDP1 was detected in CDK1-proficient cells (siCtrl) that were synchronized to mitotic phase and was markedly reduced following CDK1 knockdown (Fig. 3A,B), which is in agreement with CDK1's role in phosphorylating TDP1 at S61. We also noted that the endogenous pS61-TDP1 signal was only detected in the whole-cell extract (WCE) and was markedly reduced in the chromatin fraction independent of CDK1. We also detected a marked reduction of the endogenous TDP1 in the chromatin-bound fraction (Fig. 3A, endo-TDP1), in keeping with the immunolocalization of pS61-TDP1 in the soluble fraction during M-phase (Fig. 1G–H). In contrast, endogenous TDP1 was enriched in the chromatin fraction (~four-fold; Fig. 3B) in CDK1 knockdown cells, a situation mimicking defective TDP1 phosphorylation at S61. Further, we also confirmed the enrichment of ectopic FLAG-TDP1 (~three- to fourfold; Fig. 3C,D) in the chromatin fraction in cells deficient for CDK1 (Fig. 3C) compared to the CDK1-proficient cells by western blotting with anti-FLAG antibodies (Fig. 3C). Therefore, we surmised that TDP1 defective for phosphorylation at S61 is enriched on the mitotic chromosomes.

To further test the effect of CDK1-mediated phosphorylation at S61 on the chromosomal recruitment of endogenous TDP1 during

cell cycle, we used RO3306 to inhibit CDK1 activity and performed confocal immunofluorescence microscopy with anti-TDP1 antibody. MCF7 cells were synchronized following the protocol outlined in Fig. 3E. The cells at the indicated stages of mitosis were selected on the basis of their chromatin morphology, as confirmed by DAPI staining (Wu and Wang, 2021). We noted a similar pattern of colocalization for endogenous TDP1 on chromatin (DAPI) in the interphase cells, which was independent of RO3306 treatment (Fig. 3F, untreated and CDK1i-treated, panels *interphase*). However, TDP1 was not detected on mitotic chromosomes (Fig. 3F untreated, panel *prophase* to *telophase*). In contrast, treatment with RO3306 resulted in a marked increase in the chromosomal localization of endogenous TDP1 during mitosis (prophase to telophase) when compared to the untreated mitotic cells (Fig. 3F,G). The immunodetection of TDP1 on the chromosomes during mitosis with the CDK1 inhibitor (Fig. 3F) corroborated with chromatin enrichment of TDP1 in CDK1 knockdown cells (Fig. 3A–D). Together, CDK1 inhibition or knockdown during mitosis resulted in the occlusion of TDP1 on the mitotic chromosomes; thereby further strengthening our hypothesis that CDK1-mediated S61 phosphorylation plays a key role in ousting TDP1 from chromosomes during mitosis.

In the next step, we set out to dissect the chromosomal recruitment of the FLAG-TDP1$^{S61A}$ mutant during mitosis. To do so, we used TDP1$^{-/-}$ MEFs complemented with FLAG-TDP1$^{WT}$ (TDP1$^{-/-/WT}$) or phosphomutant FLAG-TDP1$^{S61A}$ (TDP1$^{-/-/S61A}$) or phosphomimetic FLAG-TDP1$^{S61D}$ (TDP1$^{-/-/S61D}$), synchronized to the mitosis (Fig. 3H) in the presence and absence of a low dose of CPT, which generates replication stress, and protein localization was monitored during interphase and stages of mitosis using confocal immunofluorescence microscopy (Fig. 3I–M). In contrast to interphase cells, we detected a marked reduction of the colocalization of FLAG-TDP1$^{WT}$ in the early prophase to telophase of chromosomes (DAPI) (Fig. 3I,L,M). However, the FLAG-TDP1$^{S61A}$ was readily detectable on the chromosomes during prophase to telophase independent of replication stress with CPT (Fig. 3J,L,M), suggesting CDK1-mediated S61 phosphorylation of TDP1 promotes its dissociation from mitotic chromosomes. In keeping with pS61-TDP1, the phosphomimetic FLAG-TDP1$^{S61D}$ was also delocalized from the chromosomes, emphasizing the role

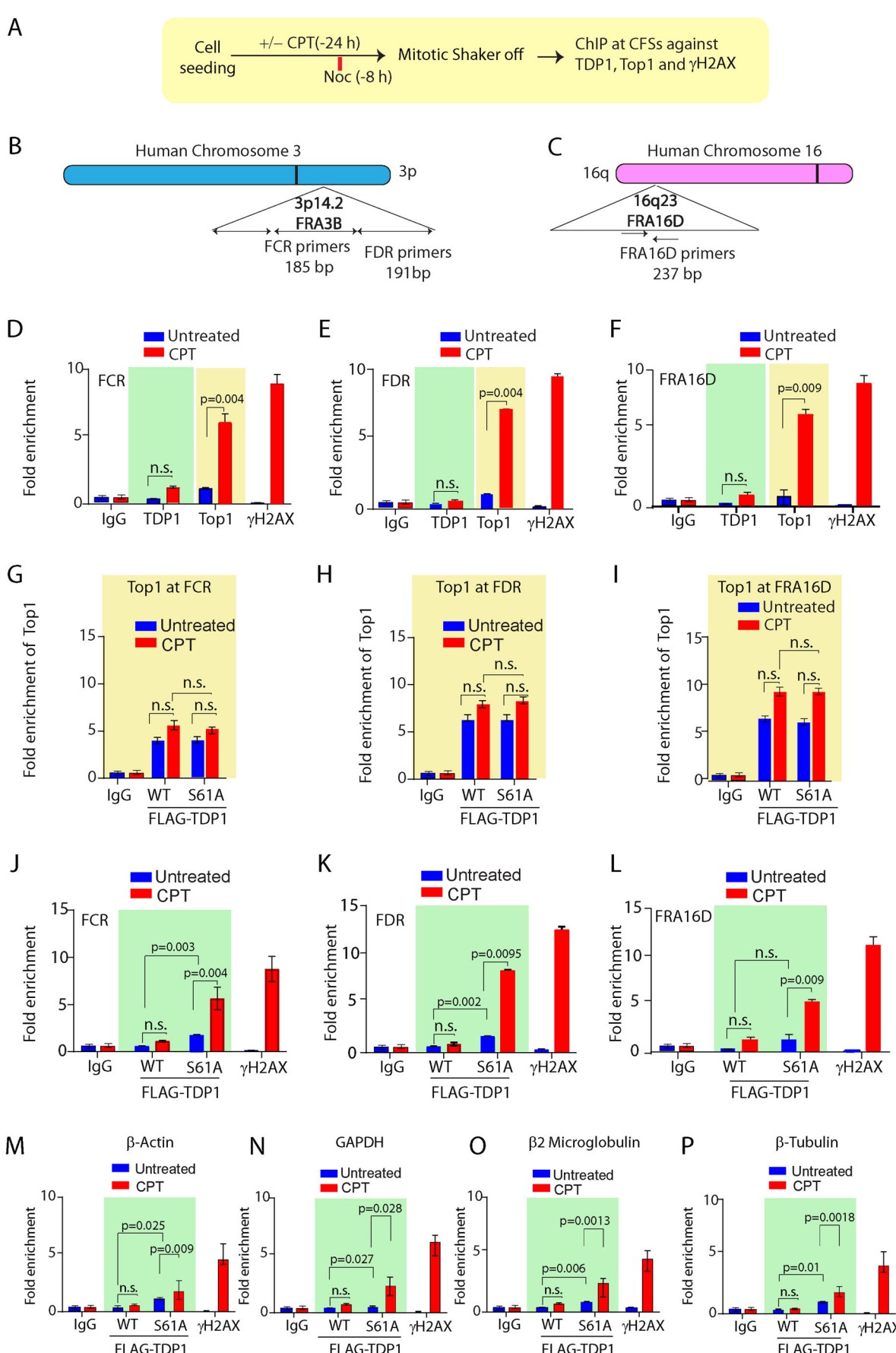

**Figure 4.  TDP1^S61A is enriched at the CFS during mitosis.**

(A) A schematic representation of the protocol for synchronization of MCF7 cells followed by chromatin immunoprecipitation (ChIP) with indicated antibodies at the CFS and non-CFS loci. (B, C) Genomic organization of the FRA3B and FRA16D regions, along with the primer sets of the distal (FDR) and central (FCR) region within the FRA3B locus; FRA16D locus, is designated. (D–F) Endogenous Top1 but not TDP1 preferentially localizes to CFSs upon CPT (15 nM; 24 h) treatment during mitosis. Quantification of cross-linked FRA3B-FCR, FRA3B-FDR and FRA16D loci chromatin-immunoprecipitated from MCF7 cells using the specified antibodies (Top1 and TDP1). The DSB marker γH2AX antibody was used as a positive control for FRA3B and FRA16D enrichment post CPT treatment. Fold enrichment over IgG was determined and is shown for each primer pair for the ChIP. Data are mean ± SD, $n = 3$ biological replicates. ns, non-significant ($P > 0.05$); **$P \leq 0.01$ (one-way ANOVA). (G–I) Quantification of cross-linked FRA3B-FCR, FRA3B-FDR, and FRA16D loci chromatin-immunoprecipitated from MCF7 cells transfected with FLAG-TDP1 variants (WT or S61A) using the specified antibodies (endogenous Top1). Fold enrichment over IgG was determined and is shown for each primer pair for the ChIP. (J–L) Quantification of cross-linked FRA3B-FCR, FRA3B-FDR, and FRA16D loci chromatin-immunoprecipitated from MCF7 cells ectopically expressing the FLAG-TDP1 variants (WT and S61A) using the specified antibodies with or without CPT treatment (15 nM; 24 h). The DSB marker γH2AX antibody was used as a positive control for FRA3B and FRA16D enrichment post CPT treatment. Fold enrichment over IgG was determined and is shown for each primer pair for the ChIP. Data are mean ± SD, $n = 3$ biological replicates. ns, non-significant ($P > 0.05$); **$P \leq 0.01$ (one-way ANOVA). (M–P) Quantification of cross-linked β-actin, GAPDH, β2-microglobulin and β-tubulin loci chromatin-immunoprecipitated from MCF7 cells ectopically expressing the FLAG-TDP1 variants (WT and S61A) using the specified antibodies with or without CPT treatment (15 nM; 24 h). The DSB marker γH2AX antibody was used as a positive control for β-actin, GAPDH, β2-microglobulin and β-tubulin enrichment post CPT treatment. Fold enrichment over IgG was determined and is shown for each primer pair for the ChIP. Data are mean ± SD, $n = 3$ biological replicates. n.s. non-significant ($P > 0.05$); *$P < 0.05$, **$P < 0.01$; (one-way ANOVA). Source data are available online for this figure.

of S61 phosphorylation in ousting TDP1 from the chromosomes (Fig. 3K–M; Appendix Fig. S3A,B).

## TDP1^S61A is trapped at CFS loci in the G2/M-phase following replication stress and triggers mitotic chromosome breaks

The chromatin-bound fraction of mutant TDP1^S61A during mitosis (Fig. 3) prompted us to investigate the sites on DNA where TDP1^S61A becomes trapped on the mitotic chromosomes. Common fragile sites (CFSs) are specific regions of the genome that are susceptible to replication stress and exhibit gaps or breaks on metaphase chromosomes under conditions that partially inhibit DNA replication (Li and Wu, 2020; Minocherhomji et al, 2015b; Özer and Hickson, 2018). We tested the enrichment of endogenous Top1 and TDP1 on mitotic chromosomes at three different CFS loci: FRA3B-FDR and FCR regions, and FRA16D (Fig. 4D–F), located on chromosomes 3 and 16, respectively (Fig. 4A–C), using chromatin immunoprecipitation (ChIP) followed by quantitative polymerase chain reaction (qPCR) analysis (Ghosh et al, 2019). To induce mild RS that causes CFS instability, the MCF7 cells were treated with a low dose of CPT or APH for 24 h, then synchronized to mitosis (Fig. 4A; Appendix Fig. 4SA). We noted a significant increase in endogenous Top1 enrichment but not TDP1 at the fragile sites which is consistent with the S61 phosphorylation dependent chromatin dissociation of TDP1 during mitosis following CPT treatment (Fig. 4D–F), in accordance to the role of Top1 in CFS breakage (Arlt and Glover, 2010; Sbrana et al, 1998; Tuduri et al, 2009). Subsequent enrichment of γH2AX confirmed CPT-induced DNA break sites at the CFSs (Fig. 4D–F,J–L).

In addition, we confirmed using ChIP assays (Fig. 4G–I) that ectopic expression of the FLAG-TDP1 variants (FLAG-TDP1^WT and FLAG-TDP1^S61A) in MCF7 cells had no effect on the Top1 enrichment at the CFSs, which suggests that CPT-induced Top1cc trapping on mitotic chromosomes is independent of TDP1-S61 phosphorylation. We detected FLAG-TDP1^S61A enrichment at CFSs that was markedly increased after CPT treatment (Fig. 4J–L). In addition, replication stress induced with APH similarly enriched TDP1^S61A at the CFSs together with γH2AX (Appendix Fig. S4B–D). We noted that both replication stress with CPT (Fig. 4J–L) and APH (Appendix Fig. S4B–D) treatment markedly increased the

enrichment of TDP1^S61A in all the CFSs, suggesting the S61 phosphorylation defective TDP1 are additionally trapped on the CFSs of mitotic chromosomes. However, we also detected CPT-induced enriched FLAG-TDP1-S61A at the four tested non-CFS, including β-actin (Fig. 4M), GAPDH (Fig. 4N), β2-microglobulin (Fig. 4O), and β-tubulin (Fig. 4P), albeit to a significantly lesser extent (Appendix Fig. S4F) compared to the CFSs, which are intrinsically more susceptible to RS. APH treatment also causes TDP1^S61A enrichment at β-actin (Appendix Fig. S4E) to significantly lesser extent than CFSs (Appendix Fig. S4B–D).

We subsequently hypothesized that TDP1^S61A trapping on mitotic chromosomes instigates additional DNA damage. To test this assumption, we used TDP1^−/−/WT or TDP1^−/−/S61A MEFs, arrested in mitosis and treated with CPT during mitotic progression, and analyzed γH2AX levels using immunocytochemistry (Fig. 5A–F). Indeed, we detected a significant increase in γH2AX levels in mitotic cells expressing TDP1^−/−/S61A MEFs compared to TDP1^−/−/WT (Fig. 5A) that was markedly increased after RS induced with CPT (Fig. 5B), suggesting TDP1^S61A trapping induces accumulation of DSBs. Further, CPT treatment also resulted in γH2AX-positive anaphase bridges and micronuclei, suggesting chromosomal DNA breaks during mitosis in TDP1^−/−/S61A MEFs (Fig. 5C–F).

We subsequently tested whether CPT-induced DNA damage during late S/G2 phase persists into mitosis, by co-staining with Top1cc and γH2AX antibodies in the TDP1^−/−/WT or TDP1^−/−/S61A or TDP1^−/− MEFs complemented empty vector (TDP1^−/−/EV) MEFs. CPT treatment in late S/G2 (Appendix Fig. S5B–D) phase led to marked accumulation of mitotic DNA breaks both in TDP1^−/−/EV or TDP1^−/−/S61A MEFs, while the CPT-induced γH2AX and Top1cc levels were significantly reduced in early S-phase (Appendix Fig. S5A,S5C,D) both in the TDP1^−/−/WT or TDP1^−/−/S61A MEFs, suggesting that S61 of TDP1 is dispensable for Top1cc repair during S-phase. Though, we noted a substantial increase in CPT-induced γH2AX in TDP1^−/−/S61A MEFs compared to TDP1^−/−/WT MEFs (Fig. 5G-H); however, this effect was not due to increased accumulation of Top1cc (Fig. 5G,I; Appendix Fig. S5E,F). Because TDP1 is excluded from the mitotic chromosomes (Figs. 3 and 4), therefore, it is suggestive that mitotic DNA breaks in TDP1^−/−/S61A MEFs are dependent on MUS81 activity, which might be required for the clearance of the trapped TDP1^S61A (Wu and Wang, 2021).

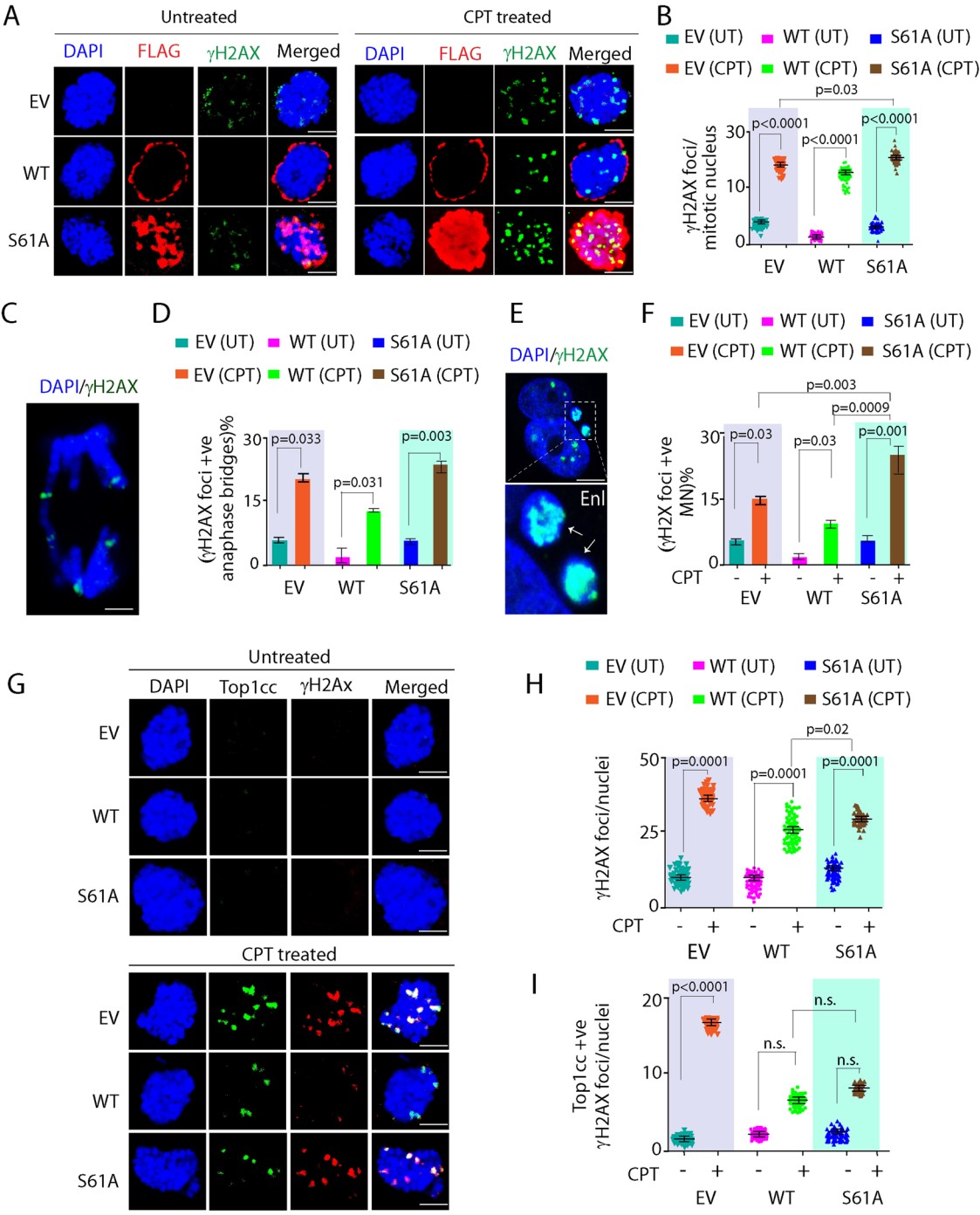

## Mitotic Top1ccs are repaired by MUS81-dependent mitotic DNA synthesis (MiDAS) independent of TDP1

Trapped Top1ccs, are broadly repaired by phosphodiesterase pathway that includes TDP1 (Huang et al, 2013; Interthal et al, 2001; Kawale and Povirk, 2018; Pommier et al, 2014); while alternative endonuclease pathways that include MUS81-EME1, XPF-ERCC1 have been implicated in the repair of trapped Top1cc (Chowdhuri and Das, 2021; Liu et al, 2002; Wu and Wang, 2021; Zhang et al, 2022). During early mitosis the nuclease activity of the structure-specific MUS81-EME1 endonuclease complex promotes a RAD52- and POLD3-mediated DNA repair synthesis (MiDAS), which serves to minimize chromosome breaks under conditions of replication stress (Bhowmick et al, 2016; Groelly et al, 2022). To test the repair of pre-mitotic trapped Top1cc-linked DNA breaks by MiDAS, we looked into BrdU or EDU incorporation on prophase DNA to mark the newly replicating cells. We ensured that the TDP1$^{-/-/WT}$ or TDP1$^{-/-/S61A}$ or TDP1$^{-/-/EV}$ MEFs had no inherent replication defect (Appendix Fig. S6A,B). The TDP1$^{-/-/WT}$ or TDP1$^{-/-/S61A}$ or TDP1$^{-/-/EV}$ MEFs were synchronized to mitosis,

**Figure 5. TDP1^S61A trapping generates mitotic DNA breaks independent of Top1ccs.**

(A) TDP1^−/− MEFs complemented with FLAG-TDP1 variants (TDP1^−/−/WT; TDP1^−/−S61A) or Empty vector (TDP1^−/−/EV) were synchronized in mitosis and treated with or without CPT (10 µM, 1 h), released in presence of nocodazole followed by immunocytochemistry with anti-FLAG (red) to detect FLAG-TDP1 and anti-γH2AX (green) antibody. Cells were counterstained with DAPI to visualize mitotic nuclei (blue). (B) Quantification for the number of γH2AX foci per mitotic nucleus calculated for 50 nuclei. Data are mean ± SD, $n = 3$ biological replicates. *$P \leq 0.05$; ****$P \leq 0.0001$ (one-way ANOVA). (C) Representative merged image showing anaphase nucleus (blue) with γH2AX (green) foci on bridges resulting from CPT treatment in TDP1^−/−S61A MEFs. The enlarged view of the anaphase bridge has been shown. (D) Quantification of γH2AX-positive anaphase bridges in TDP1^−/−/EV; TDP1^−/−/WT and TDP1^−/−S61A MEFs. Data are mean ± SD, $n = 3$ biological replicates. *$P \leq 0.05$; **$P \leq 0.01$ (one-way ANOVA). (E) Representative merged image showing G1 primary nucleus (PN) with micronuclei (MN) harboring γH2AX (green) foci resulting from CPT treatment in mitosis. The enlarged view of the MN with γH2AX in TDP1^−/−S61A MEFs has been shown. (F) Quantification of the number of γH2AX-positive G1-MN in TDP1^−/−/EV; TDP1^−/−/WT, and TDP1^−/−S61A MEFs as indicated. Data are mean ± SD, $n = 3$ biological replicates. *$P \leq 0.05$; **$P \leq 0.01$; ***$P \leq 0.001$ (one-way ANOVA). (G) TDP1^−/−/EV; TDP1^−/−/WT, and TDP1^−/−S61A MEFs were treated with CPT (15 nM; 9 h) in late S-phase as indicated to generate replication stress, synchronized in G2/M-phase followed by immunocytochemistry with anti-γH2AX (red) and anti-Top1cc (green) antibody. Cells were counterstained with DAPI to visualize mitotic nuclei (blue). (H, I) Quantifications for the number γH2AX foci (H) and Top1cc-positive γH2AX foci (I) per mitotic nucleus of TDP1^−/−/EV; TDP1^−/−/WT and TDP1^−/−S61A MEFs treated with CPT in late S calculated for 50 cells. Data are mean ± SD, $n = 3$ biological replicates. ns non-significant ($P > 0.05$); *$P \leq 0.05$; ****$P \leq 0.0001$ (one-way ANOVA). Scale bars, 10 µm. Source data are available online for this figure.

while BrdU incorporation was used to monitor MiDAS. We could detect BrdU foci on mitotic chromosomes that implicate DNA synthesis was occurring in early mitosis (Fig. 6A–D). Also, to exclusively capture DNA synthesis in mitosis, we inhibited S/G2 replication with hydroxyurea, as previously described (Macheret et al, 2020) and analyzed BrdU incorporation by immunofluorescence microscopy. We detected no BrdU or γH2AX foci in untreated cells (Fig. 6A, panel "No treatment") and these were markedly increased after CPT treatment (Fig. 6B, panel "CPT"). Under similar conditions, APH treatment shows a marked reduction in BrdU foci formation compared to cells treated with CPT (Fig. 6C, panel "APH") (Groelly et al, 2022). It is possible that MiDAS is engaged in the repair of CPT-induced Top1cc during mitosis because pS61-TDP1 is excluded (Fig. 1G,H) from mitotic chromosomes and we see the colocalization of CPT-induced BrdU and γH2AX foci (Fig. 6A–D). Notably, TDP1^−/−S61A MEFs showed a marked increase in the colocalization of BrdU and γH2AX foci upon CPT treatment (Fig. 6B) compared to TDP1^−/−/WT MEFs, suggesting that trapping of TDP1^S61A on mitotic chromosomes hyperactivates MiDAS. Intriguingly, we detected a significant reduction of both BrdU and γH2AX foci in the TDP1^−/−/WT, TDP1^−/−S61A or TDP1^−/−/EV MEFs after co-treatment with APH and CPT (Fig. 6D), suggesting replication conflicts as a potential source of these Top1cc-induced DNA breaks (Fig. 6E,F). Our study further suggests that mitotic Top1ccs are repaired primarily by MUS81-dependent endonuclease pathways.

## TDP1^S61 trapping amplifies mitotic DNA breaks by elevating MUS81 association with chromatin

Because MUS81 has been implicated in the cleavage of stalled Top1cc at the replication forks (Wu and Wang, 2021), and is an important factor in MiDAS (Bhowmick et al, 2016), we tested the recruitment of MUS81 at the CPT-induced mitotic Top1cc. We utilized two independent approaches to detect the chromatin-bound fraction of MUS81 in TDP1^−/−/WT or TDP1^−/−S61A MEFs synchronized to the mitotic phase: (i) chromatin fractionation and western blotting, and (ii) immunofluorescence staining of mitotic chromatin-bound MUS81 foci after CPT treatment. Figure 6G,H shows that CPT-induced MUS81 loading on the chromatin fraction was markedly increased in TDP1^−/−S61A MEFs compared to TDP1^−/−/WT MEFs which is consistent with the marked increase

of TDP1^S61A in the chromatin-bound fraction (Fig. 6H, left panel). Accordingly, MUS81 knockdown enriched TDP1^S61A at the FDR loci (Appendix Fig. S6C), which is consistent with the role of MUS81 in the clearance of trapped TDP1^S61A. Therefore, TDP1^S61A trapping resulted in the overloading of MUS81 on the mitotic chromosomes.

Next, we tested if MUS81 overloading triggers uncontrolled MUS81-mediated increase in DSB levels during MiDAS. CPT-induced DSBs were analyzed using γH2AX and BrdU or EdU; we already show that the number of γH2AX foci significantly increased in TDP1^−/−S61A MEFs during MiDAS, compared to TDP1^−/−/WT MEFs after CPT treatment (Fig. 6B). Notably, siRNA-mediated knockdown of MUS81 significantly reduced CPT-induced γH2AX and BrdU or EdU foci in all independent cell types (TDP1^−/−/WT, TDP1^−/−S61A, and TDP1^−/−/EV MEFs), indicating that the extensive DNA breakage of CPT-induced collapsed replication forks in mitotic cells is dependent on MUS81 but independent of TDP1 activity (Fig. 6I–L; Appendix Fig. S6D–I). Accordingly, MUS81 knockdown resulted in a significant reduction both in the EdU foci and γH2AX foci (Appendix Fig. S6D–I), suggesting the role of MUS81-mediated MiDAS in the clearance of trapped Top1ccs as well as TDP1^S61A. Because TDP1^S61A trapping accumulates mitotic DNA breaks, we tested the co-occupancy of TDP1^S61A and MUS81 at the CFS loci in MCF7 human cells. We also performed ChIP for endogenous MUS81 at CFSs in MCF7 cells expressing ectopic FLAG-TDP1^WT or FLAG-TDP1^S61A, which confirmed the co-enrichment of endogenous MUS81 together with TDP1^S61A at the CFSs, which was markedly increased after CPT treatment (Fig. 6M–P). Because both Top1cc and TDP1^S61A are enriched at CFSs after CPT treatment and cause a concomitant increase in DNA breaks (Figs. 4, 5, and 6A–F), our data suggest that MUS81 endonuclease generates DNA breaks at trapped Top1cc and TDP1^S61A sites on the mitotic chromosomes (Fig. 6M–P).

## Mitotic phosphorylation of TDP1 at serine 61 prevents genome instability

To test whether trapping of the TDP1^S61A on the mitotic chromosomes can lead to defective mitotic progression and accumulation of chromosomal aberrations (Wu and Wang, 2021), we analyzed the frequency of micronuclei (MN), bulky anaphase bridges (AB), chromatid breaks (CB), and cohesion defects (CD) in

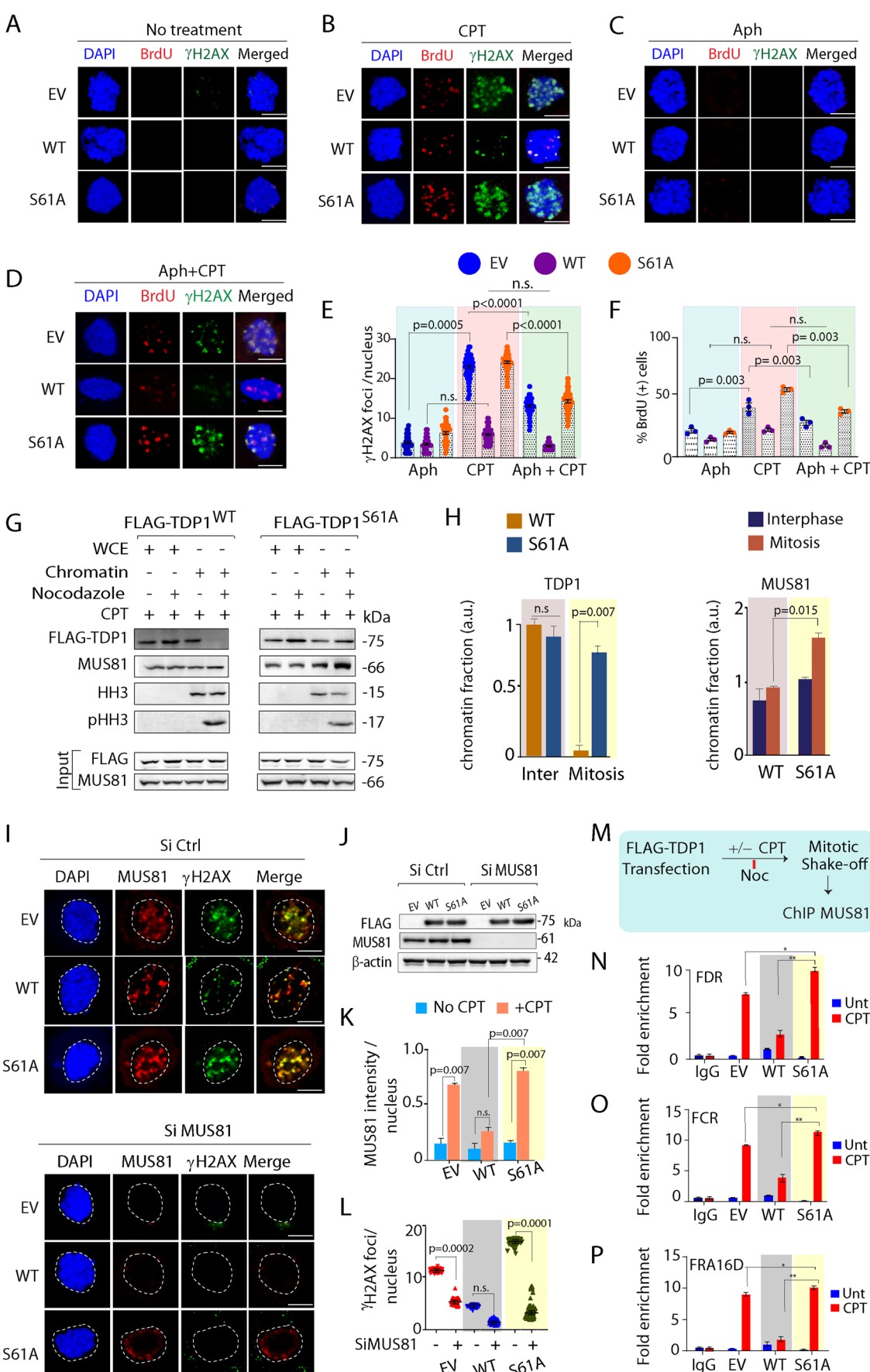

**Figure 6.   TDP1^{S61} trapping amplifies mitotic DNA breaks by elevating MUS81 chromatin enrichment.**

(A–D) TDP1^{−/−} MEFs complemented with EV or FLAG-TDP1 variants (TDP1^{−/−/WT} and TDP1^{−/−/S61A}) were treated with or without CPT (15 nM; 24 h) or Aph (0.4 μM, 24 h) alone or in combination (CPT + APH 24 h) and enriched at M-phase. Representative images show break-induced repair with newly synthesized mitotic DNA marked by BrdU foci (red). The γH2AX foci, signifying the DNA strand breaks, are shown in green. Cells were counterstained with DAPI to visualize nuclei (blue). (E, F) Quantification of γH2AX foci per nucleus and percentage of BrdU-positive mitotic nucleus obtained from immunofluorescence by confocal microscopy were calculated for 20–25 cells. Data are mean ± SD, n = 3 biological replicates. ns, non-significant ($P > 0.05$); ***$P \leq 0.001$; ****$P \leq 0.0001$ (one-way ANOVA). (G) Chromatin fractions were prepared from TDP1^{−/−/WT} and TDP1^{−/−/S61A} MEFs after CPT treatment (15 nM; 24 h) and analyzed by western blotting to detect FLAG-TDP1 variants and MUS81 with anti-FLAG and anti-MUS81 antibodies. Anti-HH3 and anti-pHH3 were used as chromosomal and mitotic markers, respectively. Protein levels of MUS81, FLAG-TDP1^{WT}, and FLAG-TDP1^{S61A} were analyzed in whole-cell lysates (WCE) to ensure equal levels of protein before chromatin fractionation. (H) Quantification showing the relative chromosomal enrichment of FLAG-TDP1 variants (WT or S61A) in interphases (Asn) and mitotic (M) chromosomes (left panel). Quantification showing the relative enrichment of MUS81 on mitotic chromosomes in interphases (Asn) and mitosis (M) for indicated cells treated with CPT (15 nM; 24 h) (right panel) n = 3 biological replicates. Data are mean ± SD, n = 3 biological replicates. ns, non-significant ($P > 0.05$); *$P \leq 0.05$; **$P \leq 0.01$ (one-way ANOVA). (I) Representative images of immunofluorescence microscopy showing induction of CPT (15 nM, 24 h)-induced γH2AX (green) and MUS81 (red) foci during mitosis in TDP1^{−/−/EV}, TDP1^{−/−/WT}, and TDP1^{−/−/S61A} MEFs, co-transfected either with Si Ctrl or Si MUS81 to knockdown MUS81. Cells were counterstained with DAPI to visualize mitotic nuclei (blue). Note: Colocalization of CPT-induced γH2AX and MUS81 foci in merged images indicates MUS81 overloading amplifies mitotic DNA breaks in TDP1^{−/−/S61A} MEFs. (J) Representative western blot analysis for the immunofluorescence microscopy in (I) to detect the levels of MUS81 and FLAG-TDP1 in TDP1^{−/−/EV}, TDP1^{−/−/WT}, and TDP1^{−/−/S61A} MEFs, co-transfected either with Si Ctrl or Si MUS81 to knockdown MUS81. β-actin has been used as loading control. (K) Quantifications of MUS81 foci on the mitotic chromosomes scored for 50 nuclei (each category) as depicted by the corresponding bar diagram. Data are mean ± SD, n = 3 biological replicates. ns, non-significant ($P > 0.05$); **$P \leq 0.01$ (one-way ANOVA). (L) Quantifications for CPT-induced γH2AX foci on mitotic nuclei in indicated cells (n = 50 cells from three biological replicates; mean ± SD). Note: siRNA knockdown of MUS81 significantly reduced CPT-induced γH2AX. ns, non-significant ($P > 0.05$); ****$P \leq 0.0001$ (one-way ANOVA). (M) A schematic representation for the protocol followed for the ChIP of endogenous MUS81 at the CFS loci in mitosis following CPT (15 nM, 24 h) treatment. (N–P) Quantification of cross-linked FRA3B-FCR, FRA3B-FDR, and FRA16D loci chromatin-immunoprecipitated from MCF7 cells transfected with empty vector (EV) or FLAG-TDP1 variants (WT or S61A) using the specified antibodies. Fold enrichment over IgG was determined and is shown for each primer pair for the ChIP. Data are mean ± SD, n = 3 biological replicates. *$P \leq 0.1$; **$P \leq 0.01$ (one-way ANOVA). Scale bars, 10 μm. Source data are available online for this figure.

TDP1^{−/−/WT}, TDP1^{−/−/S61A}, and TDP1^{−/−/EV} MEFs when exposed to a CPT (Fig. 7A–E). We found that CPT treatment markedly increased micronuclei, and chromatid breaks in the TDP1^{−/−/S61A} MEFs compared to the TDP1^{−/−/WT} or TDP1^{−/−/EV} MEFs (Fig. 7A–E), suggesting TDP1^{S61A} trapping triggers mitotic defects. Notably, TDP1^{−/−/S61A} MEFs showed a reduction in the CPT-induced chromatin breaks CB and CD when subjected to knockdown of MUS81 (Fig. 7C).

Next, to test whether the TDP1^{S61A} trapping on the mitotic chromosomes can lead to the accumulation of 53BP1 nuclear bodies in the subsequent G1 phase (Fig. 7F,G), we used TDP1^{−/−/WT}, TDP1^{−/−/S61A}, and TDP1^{−/−/EV} MEFs exposed to a low dose of CPT followed by synchronization to the next G1 phase of the cell cycle. Immunostaining of 53BP1 detected a marked increase in G1-phase 53BP1 nuclear bodies in the TDP1^{−/−/S61A} MEFs as compared to TDP1^{−/−/WT} or TDP1^{−/−/EV} MEFs, suggesting TDP1^{S61A} trapping in mitotic chromosomes propagates DNA breaks into 53BP1 bodies in G1 phase (Fig. 7F,G).

Next, we performed neutral comet assays to compare CPT-induced DSBs in TDP1^{−/−/WT}, TDP1^{−/−/S61A}, and TDP1^{−/−/EV} MEFs synchronized to mitosis (Cortés-Gutiérrez et al, 2012). Figure 7H shows that TDP1^{−/−/S61A} MEFs accumulated higher levels of CPT-induced DNA breaks (~fivefold) than TDP1^{−/−/WT} MEFs, which is due to TDP1^{S61A} trapping (Fig. 6B). Notably, MUS81 knockdown resulted in a significant decrease in mean comet length (Fig. 7H, siMUS81), indicating MUS81 deficiency reduces CPT-induced DNA breaks. Taken together our results provide evidence for the role of S61 phosphorylation of TDP1 during mitosis in the maintenance of genomic stability.

## MUS81 knockdown partly reverses TDP1^{−/−/S61A} MEFs against CPT-induced replication stress

BIR is a mutagenic mechanism that involves extensive DNA resection and mutagenic DNA synthesis and, if uncontrolled

loading of MUS81, can lead to RS-mediated cell death (Kramara et al, 2018; Malkova and Ira, 2013). MCF7 cells knockdown for MUS81 had no significant effect on CPT-induced cell survival when compared to wild-type counterpart (Fig. 7I). However, we noted MUS81 depletion, on the other hand, made TDP1^{−/−} MEFs much more susceptible to CPT, suggesting that the TDP1-dependent pathway and the endonuclease pathways operate in the repair of trapped Top1ccs (Fig. 7J) as shown previously (Marini et al, 2023). The hypersensitivity to CPT in TDP1^{−/−} MEFs caused by MUS81 depletion could be rescued by complementation with FLAG-TDP1^{WT}, further substantiating the role of TDP1 in Top1cc repair (Fig. 7J).

Because excessive DNA breaks and enhanced BIR in TDP1^{−/−/S61A} MEFs cells are dependent on MUS81 overloading (Fig. 6G,H), we investigated whether knocking down of MUS81 can rescue TDP1^{−/−/S61A} MEFs viability in response to CPT treatment. MUS81 knockdown partly restored TDP1^{−/−/S61A} MEFs survival to CPT (Fig. 7J).The rescue of the effects of TDP1^{S61A} trapping in MEFs by knockdown of MUS81 was small but reproducible in independent experiments (Fig. 7C,J). Although uncontrolled BIR causes cell death, it is plausible that regulated BIR is required for TDP1^{−/−/S61A} MEFs to survive CPT-induced DNA damage. Alternatively, MUS81's role in processing additional replication intermediates in resuming the collapsed replication forks may be needed for TDP1^{−/−/S61A} MEFs survival in response to CPT (Fig. 7J), as it is required for TDP1^{+/+} MEFs expressing TDP1^{S61A}, which further validates that excessive loading of MUS81 on the chromatin due to TDP1^{S61A} trapping leads to a marked increase in end resection causing chromosomal defects and cell death (Appendix Fig. S7).

# Discussion

In this study, we establish that principal mitotic kinase CDK1 can regulate TDP1 through phosphorylation of serine 61 during

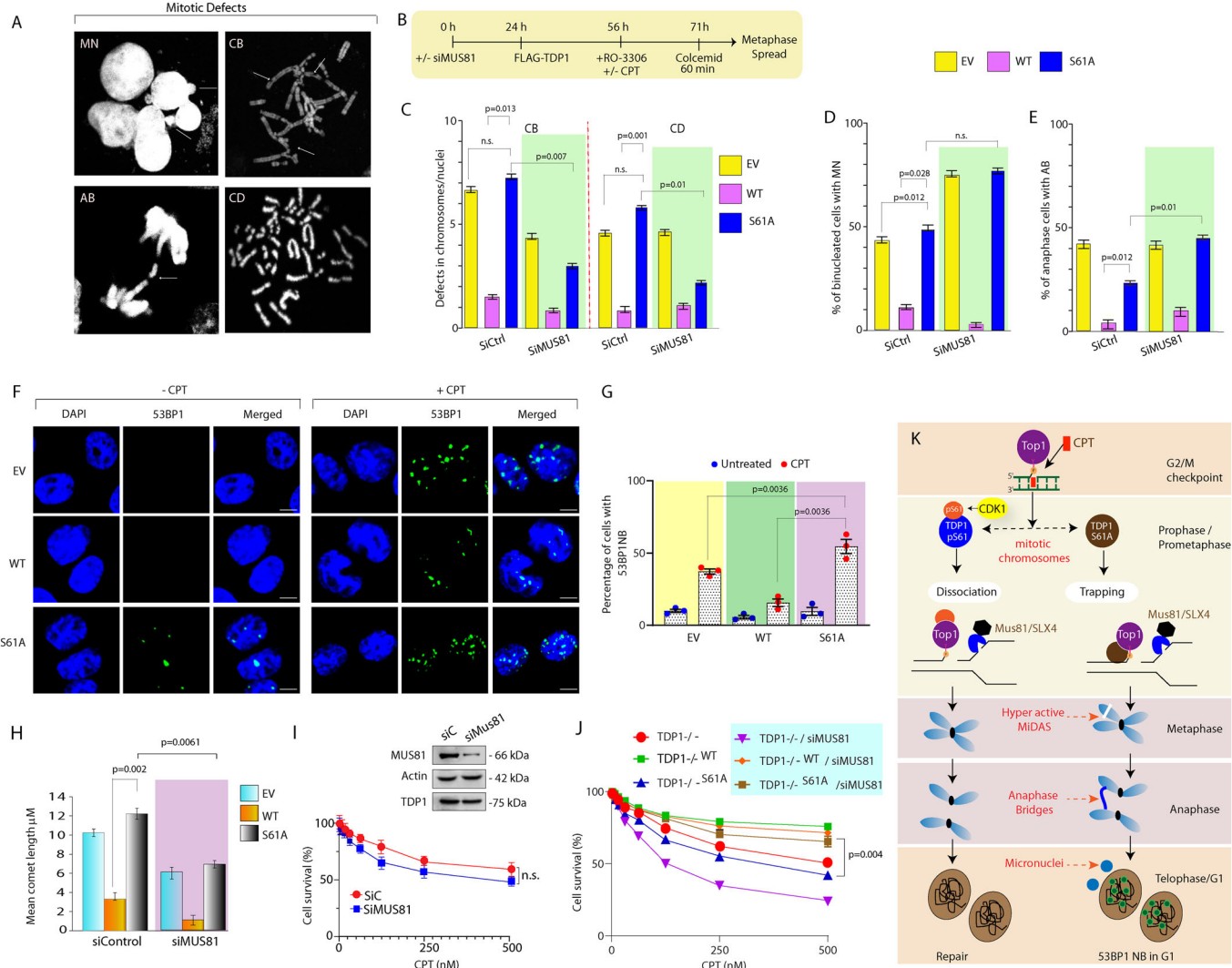

mitosis, promoting its dissociation from mitotic chromosomes, which is crucial for genome stability. TDP1 defective for S61 phosphorylation (TDP1$^{S61A}$) is trapped on the mitotic chromosomes, triggering DNA damage and mitotic defects. We detected that TDP1$^{S61A}$ is enriched at CFS loci, which are of late replicating origin, are intrinsically susceptible to RS, and also accumulate Top1ccs. Our study provides additional evidence that MUS81-dependent MiDAS is partly responsible for the clearance of mitotic Top1ccs. It is interesting to note that the significant spike in CPT-induced mitotic DNA breaks in TDP1$^{-/-/S61A}$ cells is not caused by the Top1cc elevation but due to uncontrolled MUS81 loading and excessive DNA breakage, which causes elevated MiDAS, 53BP1 nuclear bodies in G1 phase, chromosome abnormalities, and cell mortality that could be partly reversed by MUS81 depletion.

CDK1 is essential for mitotic progression and regulates mitotic DNA damage response, checkpoint activation, and mitotic catastrophe via phosphorylation of DDR proteins (Brown et al, 2015; Diril et al, 2012; Holt et al, 2009). The mechanistic implications of CDK1-mediated TDP1 phosphorylation at S61 (Fig. 2) appear to aid TDP1 temporal dynamics during mitosis

(Fig. 3). TDP1-S61 phosphorylation increased during early mitosis, then declined in late mitosis to near baseline levels in G1 (Fig. 1F), which is paralleled by CDK1 kinase activity (Appendix Fig. S1K). The mitotic chromosomes are innately refractory to pS61-TDP1 recruitment (Fig. 1G,H). Defects in TDP1-S61 phosphorylation either by CDK1 inhibition (Fig. 3E–G) or by knockdown of CDK1 (Fig. 3A–D) markedly increased (~fourfold) TDP1 in the chromatin-bound fraction during M-phase irrespective of replication stress. Accordingly, immunodetection evidenced a marked increase in retention of TDP1$^{S61A}$ on the chromosomes during prophase to telophase chromosomes (Fig. 3J,L,M), confirming CDK1-mediated S61 phosphorylation dissociates TDP1 from mitotic chromosomes. Intriguingly, we observed that replication stress with CPT or APH markedly increased the enrichment of TDP1$^{S61A}$ (Fig. 4; Appendix Fig. S4) on the mitotic chromosomes, exhibiting induction of CPT-induced DNA breaks and γH2AX on metaphase chromosomes (Fig. 5). Our results fit well with previous observations that the DDR proteins, which are the substrates of mitotic kinases, promote the dissociation of RNF8, 53BP1, Claspin, and XRCC4 at DSB-flanking chromatin sites in M-phase (Giunta

**Figure 7.   MUS81 knockdown protects TDP1$^{-/-/S61A}$ MEFs against CPT-induced replication stress.**

(A) Representative confocal microscopic images showing the mitotic defects observed in TDP1$^{-/-}$ MEFs complemented with FLAG-TDP1$^{S61A}$ following CPT treatment (15 nM; 24 h). MN micronuclei, AB anaphase bridges, CB chromatid breaks, CD cohesion defect. (B) Schematic representation of the protocol followed for the scoring of mitotic defects on metaphase chromosomes in TDP1$^{-/-}$ MEFs complemented with FLAG-TDP1 variants (TDP1$^{-/-/WT}$ and TDP1$^{-/-/S61A}$) or Empty vector (TDP1$^{-/-/EV}$), co-transfected either with Si Ctrl or Si MUS81 to knockdown MUS81. The indicated cells were synchronized in the mitotic phase following CPT treatment before metaphase spread. (C) The percentage of mitotic defects was scored for 150 metaphase spreads (each category), as depicted by the corresponding bar diagram quantification as indicated. Data are mean ± SD, $n = 3$ biological replicates. ns, non-significant ($P > 0.05$), *$P \leq 0.05$; **$P \leq 0.01$(one-way ANOVA). (D) Bar graphs showing the percentage of binucleated cells with MN in TDP1$^{-/-}$ MEFs complemented with FLAG-TDP1 variants (TDP1$^{-/-/WT}$ and TDP1$^{-/-/S61A}$) or Empty vector (TDP1$^{-/-/EV}$), co-transfected either with Si Ctrl or Si MUS81 to knockdown MUS81. Data are mean ± SD, $n = 3$ biological replicates. ns non-significant ($P > 0.05$), *$P \leq 0.05$ (one-way ANOVA). (E) Bar graphs showing the percentage of anaphase cells with AB in TDP1$^{-/-}$ MEFs complemented with FLAG-TDP1 variants (TDP1$^{-/-/WT}$ and TDP1$^{-/-/S61A}$) or Empty vector (TDP1$^{-/-/EV}$), co-transfected either with Si Ctrl or Si MUS81 to knockdown MUS81. Data are mean ± SD, $n = 3$ biological replicates. *$P \leq 0.05$; (one-way ANOVA). (F) Representative images of immunofluorescence microscopy show the induction of CPT-induced 53BP1 nuclear bodies in the G1 phase. TDP1$^{-/-}$ MEFs complemented with FLAG-TDP1 variants (WT and S61A) were treated with CPT (15 nM, 24 h) followed by G2/M arrest with 200 ng/mL nocodazole during the last 8 h of CPT treatment. Mitotic cells were harvested by shake-off and re-seeded in drug-free medium. Five hours after release, cells were fixed and stained with the anti-53BP1 antibody (green) and DAPI (blue) as nuclear stain. Scale bars, 10 μm. (G) Quantifications showing the percentage of cells detected with CPT-induced 53BP1 nuclear bodies in G1 phase calculated for 45–50 cells in indicated cell types. Data are mean ± SD, $n = 3$ biological replicates. **$P \leq 0.01$; (one-way ANOVA). (H) Quantification of CPT-induced DNA strand breaks measured by neutral comet assays in TDP1$^{-/-}$ MEFs complemented with FLAG-TDP1 variants (WT and S61A) or Empty vector (EV), co-transfected either with Si Ctrl or Si MUS81 to knockdown MUS81 arrested in mitotic phase with 200 ng/mL nocodazole added during the last 8 h of CPT treatment. CPT-induced comet tail lengths were calculated for 50 cells. Data are mean ± SD, $n = 3$ biological replicates. **$P \leq 0.01$ (one-way ANOVA). (I) Survival of MCF7 cells transfected either with Si Ctrl or Si MUS81 to knockdown MUS81 in the presence of CPT. Percentage survival was normalized for the CPT-untreated cells ± SEM. $n = 3$ biological replicates. n.s. non-significant ($P > 0.05$). Inset: western blots showing the levels of MUS81 knockdown and the corresponding TDP1 levels in the MCF7 cells. Actin was used as a loading control. (J) Survival of TDP1$-/-$ MEF cells complemented with FLAG-TDP1 variants (WT or S61A), co-transfected with expressing Si Ctrl or Si MUS81 to knockdown MUS81 and in the presence of CPT. Percent survival was normalized to the CPT-untreated cells ± SEM. $n = 3$ biological replicates. Asterisks denote statistically significant differences **$P \leq 0.01$ ($t$ test). (K) Model depicting how TDP1$^{S61A}$ trapping on mitotic chromosomes leads to hyperactivation of mitotic DNA synthesis (MiDAS), causing excessive MUS81-Eme1 endonuclease-mediated DNA resection and breaks on the mitotic DNA. These breaks manifest as bulky anaphase bridges followed by chromosomal segregation defects leading to the formation of micronuclei and the accumulation of 53BP1 nuclear bodies in G1 daughter cells. CDK1-mediated phosphorylation of TDP1 at the S61 residue promotes its exclusion from the mitotic chromosome to safeguard genomic stability. Source data are available online for this figure.

et al, 2010; Terasawa et al, 2014). RS-induced activation of CDK1-dependent phosphorylation of human 53BP1 during mitosis can eventually lead to the deactivation of 53BP1 (Belotserkovskaya and Jackson, 2014; Giunta et al, 2010; Orthwein et al, 2014). Similarly, CDK1-dependent phosphorylation of MDC1 disrupts MDC1 and γH2AX interaction, presumably to avoid checkpoint activation during mitosis (Yu et al, 2012).

RS causes cells to fail to complete replication before the S-G2 transition, allowing them to progress through the cell cycle with under-replicated DNA. These late-cell cycle stage DNA synthesis loci, which include CFSs, are among the most difficult in the human genome to replicate. Intriguingly, it is apparent that Top1 is critical for controlling the topological state of DNA during mitosis at the CFS loci (Fig. 4) (Arlt and Glover, 2010). Thus it becomes imperative that the trapped Top1ccs need clearance in mitosis for the re-establishment of the interphase transcriptome during mitotic exit (Wiegard et al, 2021). MiDAS occurs at CFSs and telomeres upon RS induced by APH and at other collapsed replication forks induced by CPT (Arlt and Glover, 2010; Chan et al, 2009; Li and Wu, 2020; Minocherhomji et al, 2015b; Özer and Hickson, 2018; Pladevall-Morera et al, 2019). Therefore, we favor the interpretation that CPT-induced DNA breaks in the condensed mitotic chromosomes are refractory to the canonical TDP1-dependent DNA repair pathways and have to rely on the nuclease activity of the structure-specific endonuclease MUS81-EME1 (Figs. 6 and 7), which has been implicated in the cleavage of stalled Top1cc at the replication forks (Regairaz et al, 2011; Wu and Wang, 2021), promotes a RAD52-mediated break-induced replication process using POLD3 at mitotic break sites (Bhowmick et al, 2023; Bhowmick et al, 2016), to minimize chromosome mis-segregation under conditions of RS. CPT-induced retention of TDP1$^{S61A}$ on mitotic chromosomes may cause replication obstacles that require MUS81 processing to prevent transcription replication collisions (Chappidi et al, 2020; Marini et al, 2023; Matos et al, 2020). In yeast,

TDP1 and RAD52 act in the same epistasis group for the repair of trapped Top1cc (Pouliot et al, 2001). Although HR is indicated as a high-fidelity repair mechanism, MiDAS is a mutagenic error-prone mechanism that leads to chromosome rearrangements and chromosomal instabilities (Wu and Wang, 2021).

Despite the N-terminal region of TDP1 spanning 1–148 aa is not required for in vitro catalytic activity (Interthal et al, 2001), the inability of TDP1$^{S61A}$ to repair mitotic Top1ccs can be attributed to the paucity of the SSBR partners associated with TDP1 like XRCC1, PNKP, Ligase III, and Polβ on the mitotic chromosomes following RS. It is interesting to note that TDP1 requires a cohort of downstream repair factors for the repair of Top1ccs (Pommier et al, 2014). The functional coupling between PARP1 and TDP1 accelerates CPT-Top1cc repair (Das et al, 2014). However, PARP1/2 gene deletion had no significant effect on RS-induced MiDAS (Richards et al, 2023), demonstrating that PARP1/2-dependent activity is not essential for BIR in mitosis, and supporting our hypothesis that RS-induced mitotic Top1cc is repaired independently of TDP1. Accordingly, TDP1$^{-/-/S61A}$ MEFs exhibited a remarkable increase in the mitotic DSBs, which were not restricted to Top1ccs generated by CPT treatment (Fig. 5). Another possibility could be due to the lack of promiscuity of TDP1 towards DNA damage substrates and a general proclivity towards a limited cohort of 3' or 5'-DNA adducts (Pommier et al, 2014). As a result, we detected an elevated enrichment of MUS81 on mitotic DNA following RS, which could be readily associated with unrestrained MiDAS in TDP1$^{-/-/S61A}$ cells (Fig. 6). Following MUS81 knockdown in these cells, there was a significant reduction in DSBs and MiDAS (Fig. 6J–M; Fig. S6D–I), allowing for a partial restoration in cell viability (Fig. 7H), suggesting the role of MUS81 endonuclease in DNA break formation for both trapped Top1cc and TDP1$^{S61A}$ from the mitotic chromosomes. Concurrent to the deleterious effects of MUS81 overloading and unrestrained MiDAS

in TDP1$^{S61A}$ expressing cells was associated with several mitotic defects like chromatid breaks, bulky anaphase bridge formation, micronuclei formation, and also the accumulation of 53BP1 nuclear bodies in the G1 daughter cells suggestive of chromosomal instability which again could be partly salvaged by MUS81 depletion (Figs. 6 and 7; Appendix Fig. S6D–I).

In conclusion, our findings show that TDP1 undergoes a unique post-translational alteration during mitosis. It demonstrates the biological importance of TDP1 phosphorylation at S61 and offers a new TDP1 post-translational regulatory mechanism (Fig. 7I). It is worth noting that melanoma and oral cavity cancer variations associated with serine-to-leucine (S61L) SNPs in TDP1 are critical to the physiological importance of S61 phosphorylation (https://portal.gdc.cancer.gov/genes). Because MiDAS is elevated in cells challenged with RS and is especially prevalent in aneuploid cancer cells with oncogene activation, inhibition of CDK1 may provide a therapeutic intervention in cancers with elevated TDP1 expression, which is known to cause chromosomal instability (Duffy et al, 2016).

# Methods

## Reagents and tools table

| Reagent or resource | Source | Identifier |
|---|---|---|
| **Antibodies** | | |
| Rabbit polyclonal anti-phospho-Histone H3 (Ser10) | Millipore | Cat# 06-570; RRID: AB_310177 |
| Mouse anti-MPM2 monoclonal antibody | Millipore | Cat# 05-368; RRID: AB_309698 |
| Mouse monoclonal anti-phospho-Histone H2A.X (Ser139) Antibody, clone JBW301 | Millipore | Cat# 05-636; RRID: AB_309864 |
| Mouse monoclonal Anti-MUS81 antibody | Abcam | Cat# ab14387; RRID: AB_301167 [MTA30 2G10/3] Cat# sc-53382; RRID: AB_2147138 |
| Rabbit polyclonal anti-histone H3 antibody | Millipore | Cat# 06-755; RRID: AB_2118461 |
| Mouse monoclonal anti-FLAG antibody (M2) | Sigma-Aldrich | Cat# F3165; RRID: AB_259529 |
| Rabbit polyclonal anti-FLAG antibody | Sigma-Aldrich | Cat# F7425; RRID: AB_439687 |
| Mouse monoclonal anti-CDK1 antibody | Cell Signaling technologies | Cat# 9116; RRID: AB_2074795 |
| Mouse anti-actin monoclonal, unconjugated, Clone actn05 (c4) antibody | Novus | Cat# NB 600-535; RRID: AB_521546 |
| Mouse anti-53BP1 monoclonal, unconjugated, Clone BP13 antibody | Millipore | Cat# 05-726 RRID: AB_309940 |
| Mouse anti-Top1cc monoclonal antibody | Millipore | Cat# MABE1084 RRID: AB_2756354 |
| Mouse anti-Top1 (C21) monoclonal antibody | Santa Cruz | Cat# sc-32736 RRID: AB_628382 |
| Mouse anti-BrdU monoclonal antibody (MoBU-1), conjugated Alexa Fluor™ 488 | Thermo Fisher Scientific | Cat# B35130 RRID: AB_2536434 |
| Rabbit serine 61-TDP1 (S61-TDP1) | Custom made | N/A |
| Rabbit phospho serine 61-TDP1 (pS61-TDP1) | Custom made | N/A |
| Goat anti-rabbit IgG (H + L) secondary antibody, HRP | Thermo Fisher Scientific | Cat# 31460, RRID:AB_228341 |

| Reagent or resource | Source | Identifier |
|---|---|---|
| Goat anti-mouse mouse IgG-h&l polyclonal, HRP-conjugated antibody | Novus | Cat# NB 7539, RRID:AB_524788 |
| Goat anti-mouse IgG (H + L) secondary antibody, Alexa Fluor 488-10 nm colloidal gold | Thermo Fisher Scientific | Cat# A-31561, RRID:AB_2536175 |
| Goat anti-rabbit IgG (H + L) highly cross-adsorbed antibody, Alexa Fluor 568-conjugated | Thermo Fisher Scientific | Cat# A-11036, RRID:AB_10563566 |
| Goat anti-mouse IgG (H + L) cross-adsorbed secondary antibody, Alexa Fluor 488 | Thermo Fisher Scientific | Cat# A-11001, RRID:AB_2534069 |
| Goat anti-mouse IgG (H + L) cross-adsorbed Secondary antibody, Alexa Fluor 568 | Thermo Fisher Scientific | Cat# A-11004, RRID:AB_2534072 |
| **Bacterial and virus strains** | | |
| *E. coli* BL21 DE3 | Thermo Fisher Scientific | Cat# EC0114 |
| *E. coli* DH5α | Thermo Fisher Scientific | Cat# 18265017 |
| **Chemicals, peptides, and recombinant proteins** | | |
| Camptothecin (CPT) | Sigma-Aldrich | Cat# C9911 |
| Aphidicolin (Aph) | Calbiochem | Cat# 178273 |
| RO3306 | Sigma-Aldrich | Cat# SML0569 |
| Hydroxyurea | Sigma-Aldrich | Cat# H8627 |
| Thymidine | Sigma-Aldrich | Cat#1895 |
| Nocodazole | Sigma-Aldrich | Cat# M1404 |
| BrdU | Thermo Fisher Scientific | Cat#00-013 |
| ProLong™ Gold Antifade Mountant with DAPI | Thermo Fisher Scientific | Cat# P36935 |
| Lipofectamine 2000 | Thermo Fisher Scientific | Cat# 11668027 |
| X-tremeGENE™ HP DNA Transfection Reagent | Sigma-Aldrich | Cat# 6366244001 |
| DNase | Sigma-Aldrich | Cat# AMPD1 |
| Tris-HCl | Himedia | Cat# MB030 |
| NaCl | SRL | Cat# 33205 |
| Sodium lauryl sulphate extrapure AR, ACS, 99% | SRL | Cat# 54468 |
| NP-40 | Sigma-Aldrich | Cat# 492016 |
| Albumin bovine (pH 6–7) fraction V for molecular biology (bovine serum albumin, BSA), 98% | SRL | Cat# 85171 |
| Sodium deoxycholate | Sigma-Aldrich | Cat# D6750 |
| Phosphatase inhibitor | Sigma-Aldrich | Cat# 524636 |
| Dithiotheriotol (DTT) | Sigma-Aldrich | Cat# 11583786001 |
| Magnesium chloride | SRL | Cat# 31196 |
| Protein A/G beads | Santa Cruz | Cat# sc-2003 |
| Glycine | Sigma-Aldrich | Cat# 50046250 G |
| Paraformaldehyde | Sigma-Aldrich | Cat# P6148 |
| BSA | Sigma-Aldrich | Cat# A5611 |
| Potassium chloride extrapure AR, 99.5% | SRL | Cat# 38630 |
| EDTA | SRL | Cat# 43272 |
| DMSO | Sigma-Aldrich | Cat# 276855 |
| DMEM—Dulbecco's Modified Eagle Medium | Thermo Fisher Scientific | Cat# 10569044 |
| Fetal bovine serum | Gibco (By Life Technologies) | Cat# 10270106 |
| Trypsin-EDTA (0.05%) | Sigma-Aldrich | Cat# 25300054 |
| cOmplete Mini, EDTA-free (protease inhibitor cocktail) | Sigma-Aldrich | Cat# 4693159001 |
| Proteinase K | Sigma-Aldrich | Cat# P2308 |

| Reagent or resource | Source | Identifier |
|---|---|---|
| **Critical commercial assays** | | |
| Reverse transcription kit | Applied Biosystems | Cat# 4368814 |
| QuikChange II XL site-directed mutagenesis kit | Agilent Technologies | Cat# 200521 |
| **Deposited data** | | |
| Raw Imaging day | This paper; Mendeley data | |
| **Experimental models: cell lines** | | |
| TDP1+/+ and −/− MEFs | Dr Cornelius F Boerkoel (Centre for Molecular Medicine and Therapeutics, University of British Columbia, Vancouver, British Columbia, Canada) | N/A |
| MCF7 | Developmental Therapeutics Program (NCI, NIH) | N/A |
| **Oligonucleotides** | | |
| *TDP1* siRNA AAGGAGCAGCAAAUGAGCCC | This paper | N/A |
| *CDK1* siRNA duplex 1: (RNA) – CCU AGU ACU GCA AUU CGG GAA AUU U duplex 2: (RNA) – GGA CAA UCA GAU UAA GAA GAU GUA G | This paper | N/A |
| *MUS81* siRNA CAGCCCUGGUGGAUCGAUA | This paper | N/A |
| Primers for human TDP1 (forward: GACGTGGACTGGCTCGTAAA; reverse: GAGCCTTAGCCTCTCTCGCTTATC) | This paper | N/A |
| Primers for human actin (forward GACCCAGATCATGTTTGAGACC; reverse: CATCACGATGCCAGTGGTAC) | PMID: 31723605 | N/A |
| Primers for CFS loci FCR forward 5′-TGTTGGAATGTTAACTCTAT CCCAT-3′; FCR; reverse 5′ATATCTCATCAAGACCGCT G- CA-3′; FDR; forward 5′-CAATGGCTTAAGCAGACATG GT-3′; FDR; reverse 5′-AGTGAATGGCATGGCTGGA ATG-3′; FRA16D; forward 5′-TCCTGTGGAAGGGATATTTA-3′; FRA16D; reverse 5′-CCCCTCATATTCTGCTTCTA-3′; | PMID: 26354865 | N/A |
| **Recombinant DNA** | | |
| pCMV-Tag2B-TDP1 WT (FLAG-TDP1^WT) | PMID: 29718323 | N/A |
| pCMV-Tag2B-TDP1-S61A (FLAG-TDP1^S61A) | This paper | N/A |
| pCMV-Tag2B-TDP1 S61D (FLAG-TDP1^S61D) | This paper | N/A |
| pET15b-His-TDP1 WT (His-TDP1^WT) | PMID: 29718323 | N/A |
| pET28A-His-TDP1-S61A (His-TDP1^S61A) | This paper | N/A |
| pET28A-His-TDP1 S61D (His-TDP1^S61D) | This paper | N/A |
| pCDNA3-HA-CDK1 WT (HA-CDK1^WT) | Gift from Dr. Sohrab Dalal (ACTREC) | N/A |
| **Software and algorithms** | | |
| ImageJ | ImageJ | https://imagej.nih.gov/ij/, RRID:SCR_003070 |

| Reagent or resource | Source | Identifier |
|---|---|---|
| LAS AF | Leica | https://www.leica-microsystems.com/products/microscope-software/p/leica-las-x-ls/ RRID:SCR_013673 |
| Origin | Origin | http://www.originlab.com/index.aspx?go=PRODUCTS/Origin RRID:SCR_014212 |
| GraphPad Prism | GraphPad Software, Inc | https://www.graphpad.com:443/, RRID:SCR_002798 |

## Mass spectrometry

The mass spectrometry proteomics data have been deposited to the ProteomeXchange Consortium via the PRIDE partner repository with the dataset identifier PXD053309.

## Cell culture, treatment, and transfections

Cell cultures were maintained at 37 °C under 5% $CO_2$ in Dulbecco's modified Eagle's medium containing 10% fetal bovine serum (Life Technologies, Rockville, MD, USA). The human embryonic kidney origin (HEK293) and human breast cancer (MCF7) cell lines were obtained from the Developmental Therapeutics Program (NCI, NIH/ USA). TDP1^+/+ and TDP1^−/− primary mouse embryonic fibroblast (MEF) cells were a kind gift from Dr Cornelius F Boerkoel (University of British Columbia, Vancouver, British Columbia, Canada). Cells were treated with the indicated concentrations of different drugs as detailed in the schematic representations in Figures. Plasmid DNAs and Si RNAs were transfected with Lipofectamine 2000 (Invitrogen) according to the manufacturer's protocol. TDP1^+/+ and TDP1^−/− MEF cells were transfected with the FLAG-TDP1 constructs using X-tremeGENE HP DNA transfection reagent (Roche) according to the manufacturer's protocol.

## Cell extracts, immunoblotting, and immunoprecipitation

Preparation of whole-cell extracts, immunoprecipitation, and immunoblotting were carried out as described previously (Rehman et al, 2018). Briefly, cells were lysed in a lysis buffer (10 mM Tris-HCl (pH 8), 150 mM NaCl, 0.1% SDS, 1% NP-40, 0.5% Na-deoxycholate supplemented with complete protease inhibitor cocktail) (Roche Diagnostics, Indianapolis, IN) and phosphatase inhibitors (Phosphatase Inhibitor Cocktail 1 from Sigma). After thorough mixing and incubation at 4 °C for 2 h, lysates were centrifuged at $12,000 \times g$ at 4 °C for 20 min. Supernatants were collected, aliquoted, and stored at −80 °C. For immunoprecipitation, cells were lysed in a lysis buffer (50 mM Tris-HCl (pH 7.4), 300 mM NaCl, 0.4% NP-40, 10 mM $MgCl_2$, 0.5 mM dithiothreitol supplemented with protease and phosphatase inhibitors). Supernatants of cell lysates were obtained by centrifugation at $15,000 \times g$ at 4 °C for 20 min and precleared with 50 µl of protein A/G-PLUS agarose beads (Santa Cruz, CA, USA). About 5 mg of precleared lysate was incubated overnight at 4 °C with indicated antibodies (2–5 µg/ml) and 50 µl of protein A/G-PLUS agarose beads. Isolated immunocomplexes were recovered by centrifugation, washed thrice

with lysis buffer, and were subjected to electrophoresis on 10% Tris–glycine gels and immunoblot analysis. Immunoblottings were carried out following standard procedures, and immunoreactivity was detected using ECL chemiluminescence reaction (Amersham) under ChemiDoc™ MP System (Bio-Rad, USA). Densitometric analyses of immunoblots were performed using ImageJ software.

## Mass spectrometry analysis of TDP1

Ectopic FLAG-TDP1[WT] complexes were immunoprecipitated with anti-FLAG antibody as described above (Rehman et al, 2018). To induce DNA damage cells expressing FLAG-TDP1 were treated with CPT (5 μM/3 h) prior to anti-FLAG immunoprecipitation and were subjected to tryptic digestion at 37 °C, overnight, followed by lyophilization, reconstitution, and fractionation applying strong cation exchange (SCX) liquid chromatography (LC) and mass spectrometry analysis as previously described (Rehman et al, 2018).

## siRNA transfection

Transfections were performed as described previously (Rehman et al, 2018). In brief, cells ($1.5 \times 10^5$) were transfected with control siRNA or 100 nM CDK1, MUS81 or TDP1 siRNA (GE Dharmacon, SiRNA-SMARTpool) using lipofectamine 2000 (Invitrogen) according to the manufacturer's protocol. Time course experiments revealed a maximum suppression of CDK1, MUS81 or TDP1 protein expression at day 3 after transfection, as analyzed by western blotting.

## Expression constructs and site-directed mutagenesis

Human FLAG-tagged full-length TDP1 (FLAG-TDP1[WT]) and His-tagged-TDP1 constructs were described previously (Das et al, 2009; Das et al, 2014). The HA-CDK1 construct were a kind gift from Dr. Sorab Dalal (Tata Memorial Centre Advanced Centre for Treatment, Research and Education in Cancer, India). The point mutations: TDP1[S61A] and in TDP1[S61D] FLAG as well as His-TDP1[S61A] and His-TDP1[S61D] were created using the "QuickChange" protocol (Stratagene, La Jolla, CA, USA). All PCR-generated constructs were confirmed by DNA sequencing.

## Immunocytochemistry and confocal microscopy

Immunofluorescence staining and confocal microscopy were performed as described previously (Das et al, 2009; Das et al, 2014). Briefly, cells were grown and drug treated on chamber slides (Thermo Scientific™ Nunc™ Lab-Tek™ II Chamber slides) followed by fixation with 4% paraformaldehyde for 10 min at room temperature and permeabilisation with absolute ethanol overnight. Primary antibodies against phosphopeptide-TDP1 (pS61), control immunopeptide TDP1, FLAG, γH2AX, Top1, MUS81, BrdU and Top1cc were detected using anti-rabbit or anti-mouse IgG secondary antibodies labeled with Alexa 488/568 (Invitrogen). Primary and secondary antibodies were used at 1:300 and 1:500 dilutions respectively. Cells were mounted in antifade solution with 4',6-diamidino-2-phenylindole (DAPI) (Vector Laboratories, Burlingame, CA, USA) and examined under Leica TCS SP8 confocal laser-scanning microscope (Germany) with a 63×/1.4 NA oil objective. Images were collected and processed using the Leica software and sized in Adobe Photoshop 7.0. The

intensity per nucleus for the different proteins of interest were determined with Adobe Photoshop 7.0 by measuring the fluorescence intensities normalized to the number of cell count (Das et al, 2009; Das et al, 2014).

## λ-phosphatase assay

To validate the phosphorylation at TDP1-S61 residue induced by nocodazole treatment (200 ng/ml) FLAG-TDP1[WT] lysates were pretreated with lambda protein phosphatase (λ-phosphatase; New England Biolabs) prior to immunoprecipiation. For this, cell lysates were supplemented with benzamidine (1.25 μl/ml) and N-Ethylmaleimide (10 mM) and mixed with λ-phosphatase in a 30 μl volume of 1× NEB buffer for PMP (New England Biolabs) supplemented with 1 mM $MnCl_2$. Reactions were incubated at 30 °C for 30 min and stopped by adding 5× Laemmli sample buffer followed by immunoprecipitation with anti-FLAG antibody. Samples were analyzed by western blotting along with mock-treated samples.

## In vitro kinase assays

Recombinant His-TDP1 (WT, S61A or S61D) proteins (1μg) as indicated were incubated with immunoprecipitated HA-CDK1[WT] kinase or immunoprecipitated endogenous CDK2, 0.05 mM ATP and in 1X kinase assay buffer (25 mM Tris-HCl (pH 7.5), 5 mM beta-glycerophosphate, 2 mM dithiothreitol (DTT), 0.1 mM $Na_3VO_4$, 10 mM $MgCl_2$) buffer for 30 min at 37 °C as described previously (Das et al, 2009). Reactions were stopped by adding 2× Laemmli sample buffer and heating at 95 °C for 5 min. Samples were loaded onto a 10% SDS-PAGE gel and run at 25 mA for 2 h. Phosphorylation reaction products were separated by SDS-PAGE, transferred onto PVDF membrane and analyzed by western blotting using anti-pS61-TDP1 and anti-TDP1 antibodies.

## Flow cytometry-based cell cycle analysis

For cell cycle profile analysis, cell samples are harvested by scraping or trypsinisation followed by pelleting the cells at 1500 rpm. The cell pellets were washed with 1× PBS before fixation with 70% ethanol. On the day of the flow cytometry, the cell suspension in 70% ethanol was pelleted and washed using 1× PBS. Next, the flow cytometry samples each containing ~$1 \times 10^6$ cells in suspension are prepared. The samples are centrifuged, and the supernatant is decanted, leaving a pellet of cells in each sample tube. In total, 0.5 mL of FxCycle™ PI/RNase Staining Solution stain is added to each flow cytometry sample, mixed well. The samples are incubated for 15–30 min at room temperature, protected from light. The samples are analyzed the samples without washing, using 488-nm, 532-nm, or similar excitation, and collect emission using a 585/42 bandpass filter or equivalent following the manufacturer's protocol.

## Cell fractionation and isolation of chromatin-bound protein

For cell fractionation and isolation of chromatin-bound proteins (Wu and Wang, 2021), cells were washed with 1× PBS followed by washing with hypotonic buffer containing 20 mM HEPES, pH 7.5, 20 mM NaCl, 5 mM $MgCl_2$ and suspended in hypotonic buffer

(10 ml). Post 10 min incubation on ice, cells were lysed to free nuclei by 45 strokes of a Dounce homogenizer and were centrifuged at $1500 \times g$ at 4 °C for 5 min to isolate the supernatant from the nuclear pellet. This whole-cell lysate was used as the input fraction for western blotting. Nuclei were further suspended in extraction buffer containing 50 mM HEPES, pH 7.5, 100 mM KCl, 0.25% Triton X-100, 2.5 mM MgCl₂, 1 mM dithiothreitol, aprotinine (1 µM), leupeptine (50 µM), 4-(2-aminoethyl)-bezenesulfonylfluoride/HCl (1 mM) and NaF (10 mM) followed by centrifugation at $600 \times g$ at 4 °C for 3 min. Nuclei were further suspended thrice in extraction buffer for complete lysis of the nuclear envelope and full extraction. Supernatants were pooled to yield nucleosolic proteins and the residual pellet contained all DNA and structure-bound proteins (chromatin fraction).

## Chromatin immunoprecipitation (ChIP)

Cells were cultured overnight at a density of $1 \times 10^7$ per 100 mm petri dish and subjected to transfections or treatments for mitotic synchronization as indicated in the experimental protocol outlines. Chromatin and proteins were cross-linked by incubating cells in 1% formaldehyde for 15 min at room temperature, and the reaction was stopped by 10 min incubation with 125 mM glycine. Cells were collected and washed sequentially with solution A (10 mM HEPES [pH 7.5], 10 mM EDTA, 0.5 mM EGTA, 0.75% Triton X-100) and solution B (10 mM HEPES [pH 7.5], 200 mM NaCl, 1 mM EDTA, 0.5 mM EGTA). The cell pellets were resuspended in 1 ml lysis buffer (25 mM Tris-HCl [pH 7.5], 150 mM NaCl, 0.1% SDS, 1% Triton X-100, 0.5% deoxycholate freshly supplemented with protease inhibitor cocktail (Roche) and sonicated on ice by 10 s pulses at 25% of maximal power on a sonicator. After centrifugation at 13,000 rpm for 15 min to remove any debris, the supernatant was precleared with protein-G-sepharose/salmon sperm DNA beads at 4 °C for 1 h. For each immunoprecipitation, 600 µl of the precleared chromatin was incubated overnight at 4 °C with 6 mg of antibodies specific for TDP1, FLAG, Top1, MUS81, and γH2AX. A reaction containing an equivalent amount of Goat/rabbit IgG was included as the background control. In total, 10% of the precleared chromatin was taken as input control. Antibody-chromatin complexes were pulled down by adding 50 µl of protein-G-sepharose/salmon sperm DNA beads and incubated for 4 h at 4 °C. The beads were washed for 10 min each with the lysis buffer followed by high-salt wash buffer (0.1% SDS, 1% Triton X-100, 2 mM EDTA, 20 mM Tris-HCl [pH 8.1], 500 mM NaCl), LiCl wash buffer (250 mM LiCl, 1% NP-40, 1% deoxycholate, 1 mM EDTA, 10 mM Tris-HCl [pH 8.0]), and TE buffer (10 mM Tris-HCl [pH 8.0], 1 mM EDTA). Finally, DNA was eluted with elution buffer (1% SDS, 100 mM NaHCO₃). Elutes were incubated at 65 °C for overnight with the addition of 5 M NaCl to a final concentration of 200 mM to reverse the formaldehyde cross-linking and digested at 55 °C for 3 h with proteinase K at a final concentration of 50 mg/ml. Following phenol/chloroform extraction and ethanol precipitation, sheared DNA fragments served as templates in semi-quantitative and real-time PCR analysis.

## Metaphase spread

For metaphase spreads, cells were transfected as per experimental requirements treated with or without low dose of CPT (15 nM for 24 h) and later incubated with 1 g/ml colcemid for the last 4 h. Cells were then harvested and treated with hypotonic solution (75 mM KCl) for 12 min, washed with chilled fixative (methanol/acetic acid 3:1), and left overnight at 4 °C. Cells were later dropped onto a chilled glass slide, air-dried and stained with 5% aqueous Giemsa. For each case, 150 metaphase plates were scored for defects.

## Analysis of micronuclei

$TDP1^{+/+}$ or $TDP1^{-/-}$ MEFs transfected with EV or FLAG-TDP1 variants (WT or S61A) and/or Si MUS81 were grown on chamber slides for analysis of micronuclei as described previously (Di Marco et al, 2017). Cell culture medium was supplemented 16 h before fixation with 2 µg/ml cytochalasin B (Sigma-Aldrich) to block cells in cytokinesis. Cells were fixed with 4% formaldehyde for 10 min and mounted with Vectashield mounting medium containing DAPI. Slides were analyzed by confocal fluorescent microscope. For quantification, only DAPI-stained binucleated cells were counted, and distinct micronuclei in the vicinity of these cells were considered as positive. At least 50 binucleated cells were scored in each experiment.

## Analysis of bulky anaphase bridges

Bulky anaphase bridges were detected as described previously (Boleslavska and Oravetzova, 2022). Briefly, $TDP1^{+/+}$ or $TDP1^{-/-}$ MEFs transfected with EV or FLAG-TDP1 variants (WT or S61A) and/or Si MUS81 were grown on chamber slides were synchronized with 9 µM RO3306 for 16 h, followed by three washes with 1× PBS for 5 min at RT and subsequent incubation in DMEM medium for a total time of 2 h at 37 °C. Cells were fixed with 4% (v/v) paraformaldehyde in PBS for 15 min at RT, followed by staining with DAPI (1 µg/ml). Cell images were acquired in a confocal microscope using a 63×/1.4 NA objective with oil immersion. The percentage of anaphase cells with bulky bridges was determined using ImageJ. At least 25 anaphase cells were scored per condition in each experiment.

## Detection of mitotic BrdU foci (mitotic DNA synthesis)

For detection of mitotic BrdU foci in MEFs, cells were transfected with empty vector (EV) or FLAG-TDP1 variants (WT or S61A) as detailed in experimental protocol, synchronized at the G1/S transition with 1.5 mM thymidine (Sigma-Aldrich) for 16 h, washed three times with PBS, released in fresh medium containing 6 µM RO3306 (Sigma-Aldrich) for 10.5 h (in the absence of aphidicolin) or for 17.5 h (in the presence of low dosage of CPT (15 nM) alone or combined with aphidicolin (1 µM)) and treated as indicated. Cells were then washed three times with warm medium and released in medium containing 100 ng/mL nocodazole (Sigma-Aldrich) and 1× BrdU (Thermo Fisher) for 60–90 min before being processed following a protocol described earlier. To suppress replication in S-phase cells that could potentially contaminate the mitotic shake-off 2 mM HU (Sigma-Aldrich) was added during the final 3 h as described previously (Groelly et al, 2022). Briefly, cells were fixed after removal of BrdU-containing medium with 4% paraformaldehyde for 10 min. The cells were washed and incubated with methanol for 15 min at −20 °C. Fixed cells were stored in 70% ethanol at 4 °C for up to a week. At the time of antibody staining, ethanol was removed, and cells were

washed twice with PBS and incubated for 1 h with 8% BSA in PBS to block nonspecific binding. After a 5-min PBS wash, the cells were incubated for 3 h with anti-γ-H2AX antibody diluted in 3% BSA in PBS. Slides were washed twice with PBS and then incubated with anti-rabbit antibody conjugated with Alexa Fluor 568 for 2 h. After a PBS wash, the cells were again fixed with 4% paraformaldehyde for 5 min, followed by 10-min incubation with 1.5 M HCl at 37 °C to denature the DNA. Cells were washed again, incubated with 0.5% Tween 20 in PBS for 5 min, and incubated with NGS for 20 min. Anti-BrdU (conjugated to alexa 488) was diluted in blocking buffer and incubated for 2 h in a humid environment. Microscopy was done on Leica TCS SP8 confocal laser-scanning microscope (Germany) with 63×/1.4 NA oil objective. Leica software was used for image processing which was later sized in AdobePhotoshop7.0. The γH2AX or BrdU foci/intensity per nucleus was measured by the fluorescence intensities normalized to the number of cell counts in Adobe Photoshop 7.0. Cells analyzed for γH2AX without nucleotide were fixed with 4% paraformaldehyde and stored in 70% ethanol at 4 °C. Antibody staining was done according to the protocol outlined above until the secondary antibody; after which cells were washed and incubated with. Preparations were mounted and imaged as described above. The γH2AX fluorescence intensity was measured as the average pixel intensity (Adobe Photoshop 7.0) of 25 cells from each sample.

## EdU labeling and detection in mitotic cells

Asynchronously growing TDP1$^{-/-}$ MEFs were transfected with Si MUS81 followed by transfection with EV or FLAG-TDP1 variants (WT or S61A) after 24 h. Cells were synchronized in late G2 phase of the cell cycle by incubation with 9 µM RO3306 for 16 h along with 15 nM CPT. Cells were then washed three times with warm medium and released in medium containing 100 ng/mL nocodazole (Sigma-Aldrich) and 20 µM 5-ethynyl-2′-deoxyuridine (EdU) for 60–90 min before being processed following a protocol described earlier. To suppress replication in S-phase cells that could potentially contaminate the mitotic shake-off 2 mM HU (Sigma-Aldrich) was added during the final 3 h as described previously (Groelly et al, 2022). This was followed by EdU detection using Click-IT Plus EdU Alexa fluor 488 Imaging Kit (Thermo Fisher Scientific). Chromosomes were stained using DAPI (Vectashield; Vector Labs). Images were captured using a Leica SP8 inverted confocal laser-scanning microscope (63×/1.40 OIL) objective.

## Neutral COMET assays

To compare the levels of DNA damage in TDP1$^{+/+}$ or TDP1$^{-/-}$ MEFs transfected with the different TDP1 variants (WT or S61A) and/or Si MUS81, samples were subjected to neutral comet assays according to the manufacturer's instructions (Trevigen, Gaithersburg, MD) as described previously (Das et al, 2009; Das et al, 2010; Das et al, 2014). Briefly, after treatment with 15 nM CPT and mitotic arrest by nocodazole treatment, cells were collected and mixed with low melting agarose. Slides were immersed in lysis solution at 4 °C for 1 h. After a rinse with deionized water, slides were immersed in a 4 °C electrophoresis solution (50 mM NaOH, 1 mM EDTA, and 1% dimethyl sulfoxide) for 1 h. Electrophoresis was carried out at a constant voltage of 25 V for 30 min at 4 °C. After electrophoresis, slides were neutralized in 0.4 M Tris-HCl

(pH 7.5), dehydrated in ice-cold 70% ethanol for 5 min, and air-dried. DNA was stained with ethidium bromide (EtBr) purchased from Sigma (USA). The relative length and intensity of EtBr-stained DNA, tails to heads, is proportional to the amount of DNA damage present in the individual nucleus. Comet length was measured using the TriTek Comet Score software (TriTek Corp, Sumerduck, VA) and was scored for at least 50 cells. Distributions of comet lengths were compared using ANOVA.

## Cell survival assays

MCF7 cells (1–2 × 10³) were transfected with control or MUS81 siRNA (100 nM) as described above and seeded in 96-well plates. After 24 h, cells were treated with CPT at the indicated concentrations and kept further for 48 h. In a related experiment the TDP1$^{+/+}$ and TDP1$^{-/-}$ MEFs ectopically expressing FLAG-TDP1 variants (WT or S61A) and/or Si MUS81 and treated with varying doses of CPT for 72 h. In all independent cases the transient knockdown was performed 24 h prior to the DNA transfections and cell survival was then assessed by 3- (4,5-dimethylthiazol-2-yl)-2,5-diphenyltetrazolium bromide (MTT) purchased from Sigma, USA as described previously (Rehman et al, 2018). Plates were analyzed on Molecular Devices SpectraMax M2 Microplate Reader at 570 nm. The percent inhibition of viability for each concentration of CPT was calculated with respect to the control. Data are represented mean values ± SD for three independent experiments.

## Data availability

The mass spectrometry proteomics data have been submitted to the ProteomeXchange Consortium via the PRIDE partner repository under the dataset number PXD053309. The raw data for each figure and the numerical data for this study are provided with the manuscript. Correspondence and material requests should be directed to Dr. Benu Brata Das, pcbbd@iacs.res.in. This article includes expanded view data, reagent table and appendices.

The source data of this paper are collected in the following database record: biostudies:S-SCDT-10_1038-S44318-024-00169-3.

## Peer review information

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

## Acknowledgements

BBD team is supported by SERB core research grant (CRG/2022/001322), BRNS grant (54/14/10/2022-BRNS/11014) and ICMR grant (2021-11299/CMB/ADHOC-BMS) and IACS intramural funds. SPC is the recipient of the IACS senior research fellowship, India.

## Author contributions

**Srijita Paul Chowdhuri**: Resources; Data curation; Formal analysis; Validation; Investigation; Visualization; Methodology; Writing—original draft; Project administration. **Benu Brata Das**: Conceptualization; Resources; Formal analysis; Supervision; Funding acquisition; Writing—original draft; Project administration; Writing—review and editing.

Source data underlying figure panels in this paper may have individual authorship assigned. Where available, figure panel/source data authorship is listed in the following database record: biostudies:S-SCDT-10_1038-S44318-024-00169-3.

## Disclosure and competing interests statement

The authors declare no competing interests.

