## [Peer Review File · The EMBO Journal]

TDP1 phosphorylation by CDK1 in mitosis promotes MUS81-dependent repair of trapped Top1-DNA covalent complexes

Benu Das and Srijita Paul Chowdhuri

Corresponding author(s): Benu Das (pcbhd@iacs.res.in)

Review Timeline:

Submission Date:	15th Nov 23
Editorial Decision:	13th Dec 23
Revision Received:	17th May 24
Editorial Decision:	14th Jun 24
Revision Received:	19th Jun 24
Editorial Decision:	24th Jun 24
Revision Received:	26th Jun 24
Accepted:	28th Jun 24

Editors: Hartmut Vodermaier and Ioannis Papaioannou

Transaction Report:

Prof. Benu Brata Das
Indian Association for the Cultivation of Science
Laboratory of Molecular Biology, Department of Physical Chemistry
2A & B, Raja S. C. Mullick Road, Jadavpur
Kolkata, West Bengal 700032
India

13th Dec 2023

Re: EMBOJ-2023-116166
TDP1 phosphorylation at S61 promotes mitotic progression and protects Top1 covalent complexes

Dear Benu,

Thank you again for submitting your manuscript on mitotic TDP1 phosphorylation to our journal. It has now been assessed by three expert referees, whose reports are copied below. As you will see, at least two of the referees are generally supportive of the study and acknowledge its interest, and we would therefore in principle be open to considering a revised version further for EMBO Journal publication. At the same time, you will see that all three reports raise a number of overlapping key concerns that would need to be satisfactorily addressed before publication might be warranted. These include issues with the over-reliance on RO3306 and absence of unperturbed cell cycle analyses, concerns regarding the CFS and MUS81 results, and a number of additional more specific criticisms that I will not repeat all in detail here in this letter.

In this situation, I would like to give you an opportunity to address the reports via a revised version of the manuscript, but I should emphasize that we only allow a single round of major revision, making it important to fully and carefully respond to each referee point at the time of resubmission. I would therefore encourage you to contact me with a revision plan and preliminary point-by-point response already during the early stages of your revision work, in order to clarify if and how key issues raised in the reports may be solved; happy to discuss in person based on such a proposal. We would also be open to extension of the default three-months revision period if needed; our 'scooping protection' (meaning that competing work appearing elsewhere in the meantime will not affect our considerations of your study) would of course remain valid also throughout such an extension.

Further information on preparing, formatting and uploading a revised manuscript can be found below and in our Guide to Authors. Thank you again for the opportunity to consider this work for The EMBO Journal, and I look forward to hearing from you in due time.

With best regards,

Hartmut

9) Digital image enhancement is acceptable practice, as long as it accurately represents the original data and conforms to community standards. If a figure has been subjected to significant electronic manipulation, this must be clearly noted in the figure legend and/or the 'Materials and Methods' section. The editors reserve the right to request original versions of figures and the original images that were used to assemble the figure. Finally, we generally encourage uploading of numerical as well as gel/blot image source data; for details see: embopress.org/page/journal/14602075/authorguide#sourcedata

At EMBO Press, we ask authors to provide source data for the main manuscript figures. Our source data coordinator will contact you to discuss which figure panels we would need source data for and will also provide you with helpful tips on how to upload and organize the files.

Further information is available in our Guide For Authors:

In the interest of ensuring the conceptual advance provided by the work, we recommend submitting a revision within 3 months (12th Mar 2024). Please discuss the revision progress ahead of this time with the editor if you require more time to complete the revisions. Use the link below to submit your revision:

Link Not Available

Referee #1:

GENERAL SUMMARY & OPINION ABOUT THE PRINCIPAL SIGNIFICANCE OF THE STUDY

This is an interesting and exciting study that provides us with new molecular insights into the post-translational modifications of TDP1 and the role of TDP1 in genome stability maintenance. In brief, the authors identify a conserved residue in vertebrate TDP1 (S61) that is phosphorylated by mitotic CDK1 (presumably CDK1-cyclin B). Through the use of a phospho-alanine mutant (S61A), they provide data indicating that the phosphorylation of TDP1 promotes its dissociation from mitotic chromatin to prevent excessive accumulation of MUS81 and activation of mitotic DNA synthesis (MiDAS), which ultimately manifests as genome instability. They further propose that CPT-induced DNA breaks in condensed mitotic chromosomes are refractory to canonical TDP1-dependent pathways and rely on error-prone repair through MiDAS.

In general, the manuscript is well-written and the data is presented in a coherent and well-controlled manner. The main findings are thought-provoking and will have important implications for researchers interested in the molecular mechanisms of MiDAS and chromosome stability, particularly with respect to the roles of TDP1 and the cellular pathways that respond to the accumulation of Top1cc's. After addressing the points below, this manuscript will be a strong candidate for publication in EMBO Journal.

SPECIFIC MAJOR CONCERNS ESSENTIAL TO BE ADDRESSED TO SUPPORT THE CONCLUSIONS

1. The first take-home message is that CDK1 phosphorylates TDP1 at S61. While the data presented is largely convincing, the following textual changes and experiments would substantially strengthen their conclusions:
 - a. The authors should provide more details about the mass spectrometry data that identified pS61, including: What was the level of coverage? Was this the only phospho-site identified in TDP1?
 - b. Figure 1C. To confirm that the antibody is recognizing TDP1-pS61, the authors should IP FLAG-TDP1 and treat with a broad-spectrum phosphatase (e.g., lambda phosphatase).
 - c. Figure 1E: The labelling is very confusing. Is this post-release from RO3306 or post-release from nocodazole? If the former, why do you see such strong signal for pS61 (RO3306 inhibits CDK1). If the latter, suggest to remove the 'noc' label altogether since you're releasing into drug-free media (not nocodazole).
 - d. In Figure 2F, the authors use siRNA to show that the depletion of CDK1 abolishes TDP1-pS61 (IP FLAG-TDP1 and WB for pS61). Since CDK1-dependent phosphorylation of S61 is central to this manuscript, the authors should provide an orthogonal experiment to confirm this finding. For example, by repeating this experiment in cells treated with the CDK1 inhibitor RO3306.
2. The second main message of the manuscript is that TDP1-S61A gets trapped on mitotic chromatin, leading to DNA DSBs and genome instability, which triggers chromatin loading of MUS81 and excessive MiDAS. Addressing the following points will strengthen their conclusion.
 - a. Figure 1D: The authors state that expression of TDP1-S61A does not impact cell cycle progression. However, the cell cycle distribution for S61A looks very different than the WT cells at 1 and 5 hr release. Why are there more S61A cells in G0/G1 after 1 hr release than 5 hr release? Are the cells undergoing apoptosis? This needs to be clarified.
 - b. Figure 3D: The authors should provide representative images of untreated cells in each subphase of mitosis to allow direct comparison with RO3306-treated cells.
3. Statistical tests. The authors use t-tests for all of their comparisons. However, t-tests should not be used for multiple comparisons, such as those shown in Figs. 5C, 5G, 5H, 5I, 6E, 6I, 6L, 6K, 6M, 7E, and 7F.
4. Literature review (page 5). The authors state incorrectly that MUS81 is phosphorylated by CDK1. After checking the works cited, Payliss et al show that SLX4 is phosphorylated by CDK1 and Palma et al show that MUS81 is phosphorylated by CK2.

MINOR CONCERNS THAT SHOULD BE ADDRESSED

1. To improve the general clarity of this work, I recommend that the authors discuss their data in the same order that it appears in the figures (e.g., Figure XA, XB, XC, etc.). In some cases, they may need to modify the order of the data panels in the figure itself.
2. The authors should cite PMID: PMC2504835, who also identified phosphorylation at TDP1 S61.
3. While the introduction and the discussion were very well-written and easy to follow, I found the results section to be rather dense and difficult to follow in some places. This should be an easy fix with some additional editorial input and addressing the points listed below. Additionally, I would encourage the authors to remove all in-text references to the effect of "and the quantification in XX". These extra words are not needed.
4. Pages 8 and 21. The authors state that TDP1 pS61 peaked upon release from early mitosis and then declined in telophase. Given that this conclusion is based on western blots, which do not have the resolution to distinguish between the subphases of mitosis (e.g., anaphase and telophase), I strongly advise to replace 'telophase' with 'late mitosis'.
5. Page 9, first two sentences in section called "CDK1 binds TDP1 to phosphorylate at S61". The authors state that they looked for a direct interaction between CDK1 and TDP1 because CDK1 phosphorylates Top1. I don't follow their logic...was there a typo?
6. Page 9. It is well-established that coIPs capture direct and indirect protein-protein interactions. As such, a coIP alone is not sufficient evidence for the authors to conclude that CDK1 binds directly to TDP1. While I agree that this is the most likely interpretation of their data as a whole, they should remove the word 'directly' in the sentence starting "This prompted us to ...".
7. Page 11, sentence starting "The endogenous pS61-TDP1 was detected in ...". The quantification shown in Figure 3A does not match the data described in the text...?
8. Page 11, sentence starting "We also detected a marked reduction of ...". The word 'cytoplasm' should be replaced with 'soluble fraction' (or similar) since the nuclear membrane breaks down in mitosis.
9. Page 11, sentence starting "In contrast, endogenous TDP1 was enriched ...". I don't understand what the authors are comparing in Figure 3A and 2F...Figure 2F does not show any chromatin fractionation blots. Please clarify/remove.
10. It seems unlikely that the pan-chromatin staining for TDP1-S61A in mitotic cells is solely due to trapping at CFSSs. This should

be discussed briefly in the manuscript, particularly with respect to what % of the genome contains CFSs. Related to this point, it would be useful to know the % of mitotic cells that showed the pan-chromatin staining for TDP1-S61A (I'm assuming that the quantification of intensity only represents chromatin containing TDP1-S61A signal).

11. Page 15, bottom of the page. Replace "denaruring" with "denaturing".
12. Figure 4J-L. The data looks solid so it would be nice to statistically compare the WT and S61A for each condition.
13. Figure 4K. I am not sure that the authors can confidently quantify foci in these images because of the lack of distinct particles. Perhaps total intensity is the better approach.
14. Figure 6K-M. I believe that the y-axis title should be "fold enrichment" as opposed to "MUS81".
15. "Western" should only be capitalized if it is the first word of the sentence.
16. It is a bit confusing to switch back and forth between CDK1i and RO3306. I would pick one and stick with that throughout the manuscript.
17. Page 23, first sentence of second paragraph. The authors acknowledge that the TDP1 NTD, which contains S61, is not required for in vitro catalytic activity (this should be acknowledged much sooner in the manuscript) and suggest that the inability of TDP1-S61A to repair Top1cc's is due to impaired interactions with TDP1 binding partners. It would be good to provide citations for work showing that XRCC1, PNKP, ligIII, and pol beta bind to the TDP1 NTD. If the binding interfaces are not known, then the authors should clarify that in the text.

ADDITIONAL SUGGESTIONS FOR IMPROVING THE STUDY (EDITOR'S DISCRETION)

1. Figure 2: the authors provide good evidence for an interaction between CDK1 and TDP1, which is independent of CDK1 activity. This data would be strengthened substantially by a TDP1 mutant that cannot bind CDK1. Does S61A bind CDK1? Can they leverage AlphaFold Multimer to predict a high confidence model for CDK1 and TDP1?
2. The authors have some interesting data regarding MUS81. Since these experiments were derived using siRNA, the conclusion could be further strengthened by repeating the key phenotypic experiments with siRNA-resistant MUS81 constructs (WT or nuclease-dead).
3. Since many of the experiments rely on over-expression of FLAG-TDP1, it would be useful for the authors to comment on the relative expression of FLAG-TDP1 to endogenous TDP1. They seem to have a good antibody for endogenous TDP1 so this should be a relatively easy experiment to perform. It's odd that it wasn't included in the manuscript.

Referee #2:

The article from Chowdhuri and Das defines how the phosphorylation of TDP1 affects its localization and function in mitosis. TDP1 is an important protein to study because of its role in DNA repair and suppression of human disease. However, in its current form, the manuscript has a number of weaknesses and the design of many of the experiments is sub-optimal.

Specific Remarks

- 1) The title should be changed in my view because it doesn't accurately reflect the content of the manuscript. It implies that TDP1 plays a key role in mitotic progression, which is not really the case.
- 2) The opening sentence of the Abstract is an overstatement in my view. TOP1 is not really a key player in chromosome separation.
- 3) It should be tested whether S61 phosphorylation of TDP1 is specific for mitosis by synchronizing cells in interphase (e.g. by using thymidine). If the residue is phosphorylated in interphase, CDK1 is unlikely to be the only kinase that can target this residue. Also, in Fig 1 it is stated that cells were synchronized to G2/M using nocodazole - this synchronizes cells in prometaphase.
- 4) It is not ideal to use RO2206 as a synchronizing agent when it targets the very protein that is under study - CDK1. One significant problem is that RO3306 affects the replication program in S-phase. Also, when cells are held at the G2/M boundary using this agent, CDK1 levels continue to rise. Hence, when cells are released, they show hyper-activation of CDK1.
- 5) In panels G-H of Fig 1, it is claimed that pS61 was not detected in interphase cells - but some foci clearly are shown. It is also stated that this absence of pS61 is consistent with western blot analysis (Fig 1C) but that panel didn't study cells in interphase - only mitotic cells.
- 6) In Fig 2, the use of benzonase in the co-IP doesn't prove that CDK1 binds to TDP1 directly - only that it doesn't require DNA to phosphorylate TDP1.

- 7) The in vitro kinase assays are not very meaningful without a specificity control. CDK2 or other CDKs should be tested.
- 8) On p11, it is stated that TDP1 defective for phosphorylation at S61 is enriched on mitotic chromosomes. It would be better to conclude that phosphorylation of TDP1 leads to the protein being excluded from mitotic chromosomes. Hundreds of proteins are excluded from condensed mitotic chromosomes and TDP1 seems to be another one of them. This isn't really a major finding. Also, on p11 it is stated that cells were 'synchronized to mitosis in the presence of RO3306'. This cannot be correct as RO3306 prevents cells entering mitosis.
- 9) The analysis of fragile sites lacks important controls. In particular, the analyses in Fig 4 of enrichment of TDP1-S61A at CFSs lacks analysis of any non-CFS locus. Given that the IF analysis shows TDP1-S61A staining all over mitotic chromosomes, it is inconceivable that its localization to CFSs is in any way specific for these loci. If it binds everywhere on mitotic chromosomes, it is debatable whether the CFS work should be included at all.
- 10) Why was BrdU and denaturation used to detect mitotic DNA synthesis rather than the commonly used EdU?
- 11) I was confused about some aspect of the MUS81 analysis. On p17 it is stated that knockdown of MUS81 affects 'resection of CPT-induced collapsed replication forks'. I couldn't see anywhere that resection had been analyzed and indeed it wasn't shown that the effects depended on the collapse of replication forks.
- 12) The rescue of the effects of TDP1-S61A in MEFs by knockdown of MUS81 are very minor. Either this should be removed, or the text modified to acknowledge that the effects are small and could easily be an indirect effect of MUS81 being compromised in interphase and/or mitosis.

Minor Comment

- 1) On p9, line 2 'extruded' should read 'excluded'. In general, the English would benefit from some polishing.

Referee #3:

Below I provide review of the manuscript by Choudhury and Das, entitled "TDP1 phosphorylation at S61 promotes mitotic progression and protects Top1 covalent complexes". The manuscript provides valuable insights into the role of TDP1 in genome maintenance, specifically highlighting the impact of its phosphorylation at S61 and its interaction with CDK1. This phosphorylation event leads to the dissociation of TDP1 from mitotic chromosomes, impacting mitotic progression under replication stress. The authors further explore the consequences of loss of this phosphorylation on S61, resulting in its accumulation on chromatin and triggering DNA damage response that requires MUS81-dependent mitotic DNA synthesis (MiDAS) mechanism.

While the observations presented are intriguing, I would like to raise several points that need to be addressed to support publication of this manuscript in EMBO Journal.

Major Concerns:

- 1) The manuscript indicates TDP1 phosphorylation by CDK1, yet the use of CDK1 inhibitor RO3306 for cell synchronization in most experiments a potential confounding factor that could impact the overall validity of the authors' claims. To ensure the robustness of the findings and avoid any misinterpretation, I recommend considering Thymidine-Nocodazole mitosis phase synchronization (PMID: 28654080).
- 2) While TDP1 phosphorylation and its interaction with CDK1 are not dependent on DNA damage (Fig.1A, p.7 and Fig. 2A, p. 9), the frequent use of CPT or other damaging agent may obscure the native effects of S61 phosphorylation. The authors are encouraged to perform key experiments under normal conditions. For example, the enrichment of S61 at CFS in untreated conditions is little if any, however CPT induces strong accumulation of S61. CPT clearly represent other mechanism of S61 retention on chromatin, dependent on replication and MUS81 (Fig. 6). CPT-induced TDP1S61 trapping thus represent replication obstacle that requires MUS81 processing, similar to recently described transcription replication collision (Chappidi et al 2020, and Matos et al 2020). This should be properly cited and discussed. Can TDP1 S61 stall replication forks (DNA fibers) and be involved in stabilizing R-loops (S9.6 and/or RNaseH1-ND staining)? Importantly, this seems to be Top1cc independent (Fig. 4G-I) thus representing novel role of TDP1?
- 3) Considering the presented hypothesis, the inclusion (at least transiently) phosphomimetic mutant S61D would provide valuable insight and strengthen the conclusions.
- 4) A thorough proofreading and significant clarification of the text are recommended. Additionally, the relocation of some panels with confirmatory experiments to supplementary information would also improve the clarity.

Minor concerns:

- Fig. 1 C and E: Include loading controls (e.g., Actin or GAPDH).
- Fig. 1D: Clarify the observed difference in cell cycle progression. While the authors state that MCF7 cells expressing FLAG-TDP1 variants (WT or S61A) did not perturb the cell cycle progression (page 8), but there is a visible difference after 1 hour of release as shown in Fig. 1 D.
- Fig.4: Perform a proper statistical comparison between WT and S61 mutant in untreated conditions to conclude that TDP1 is excluded from the mitosis chromosomes. In addition, to confirm the role of MUS81 in processing CPT-trapped S61 mutant, an experiment (i.e. Fig. 4K) with siMUS81 should be performed.
- P.15 authors established that accumulation of S61 is not related to Top1cc, yet they try to link it to their processing by MIDAS.

This is not logical and rather MUS81-mediated processing of transcriptionally stalled replication forks could be referred to.

- Fig. 6: Include Aphidicolin- only conditions as a control for MiDAS experiments (PMID: 36002001).
- Fig. 6B: Include a control WB for depletion of MUS81.
- Fig. 6H: There is a lot of controversy in the literature about MUS81 foci and their reproducibility. Specify the antibody used for MUS81 foci and check the effect in other cell type (U2OS).
- Fig 7B: Provide details on the analysis of micronuclei and anaphase bridges from metaphase spreads?
- P.19: In the text "MCF7 cells knockdown for MUS81 had no significant effect on CPT-induced cell survival when compared to wildtype counterpart (Fig 7H)." instead of Fig 7H it should be 7G. In addition, observed sensitization of TDP1 KO cells by siMUS81 to CPT have been recently also reported and should be properly referenced (Marini et al 2023).
- Consider investigating the effect of S61 phosphorylation on mitochondrial DNA and vice versa if TDP1 H493R mutant will show chromatin trapping phenotype during mitosis? <https://pubmed.ncbi.nlm.nih.gov/31723605/>

Point-by-point response to reviewers' questions.

Referee #1:

SPECIFIC MAJOR CONCERNS ESSENTIAL TO BE ADDRESSED TO SUPPORT THE CONCLUSIONS

1. The first take-home message is that CDK1 phosphorylates TDP1 at S61. While the data presented is largely convincing, the following textual changes and experiments would substantially strengthen their conclusions:

a) The authors should provide more details about the mass spectrometry data that identified pS61, including: What was the level of coverage? Was this the only phospho-site identified in TDP1?

Answer: We thank the reviewer for finding our manuscript largely convincing and also appreciate the insightful comments. The MS analysis covered 93% of the immunoprecipitated FLAG-TDP1 complex that identified pS61 as a site of phosphorylation. The other phosphorylation sites identified in the TDP1-MS data were Y167, Y178, S434, S459, S614. The S61 residue of human TDP1 is phylogenetically conserved across vertebrate species as a proline-directed phosphorylation site (Fig 1A) and lies within a conserved motif, which is the preferred substrate for Cyclin-dependent kinase 1 (CDK1).

b) Figure 1C. To confirm that the antibody is recognizing TDP1-pS61, the authors should IP FLAG-TDP1 and treat with a broad-spectrum phosphatase (e.g., lambda phosphatase).

Answer: We appreciate the reviewer's recommendation and have performed additional experiments to confirm that the antibody is recognizing TDP1-pS61. Specifically, we have immunoprecipitated FLAG-TDP1 and then treated the sample with the broad-spectrum phosphatase (λ -phosphatase), as advised. We have updated our manuscript to reflect the new findings in Appendix Fig S1B. Thank you.

c) Figure 1E: The labelling is very confusing. Is this post-release from RO3306 or post-release from nocodazole? If the former, why do you see such strong signal for pS61 (RO3306 inhibits CDK1). If the latter, suggest to remove the 'noc' label altogether since you're releasing into drug-free media (not nocodazole).

Answer: We apologize for the confusion created due to the labeling related to “*post-release from RO3306 or post-release from nocodazole*”. In the revised manuscript we have changed the labelling and included a more simplified version of the protocol scheme. Please note based on the suggestions of the other reviewers we have now used a new mitotic synchronisation protocol using thymidine-nocodazole without RO3306. Consistent with the previous protocol, indeed we see a significant increase in the levels of CDK1-mediated pS61-TDP1, which is caused by CDK1 activity during the mitotic phase. As recommended, we have updated the concerned labelling in Fig 1E which now reads “time post release (h)” in the revised Fig 1F. Thank you.

d) In Figure 2F, the authors use siRNA to show that the depletion of CDK1 abolishes TDP1-pS61 (IP FLAG-TDP1 and WB for pS61). Since CDK1-dependent phosphorylation of S61 is central to this manuscript, the authors should provide an orthogonal experiment to confirm this finding. For example, by repeating this experiment in cells treated with the CDK1 inhibitor RO3306.

Answer: We appreciate the reviewer's suggestion for performing new experiments using the CDK1 inhibitor RO3306 to confirm CDK1-mediated TDP1 phosphorylation at S61. We have conducted new experiments using the immunoprecipitation of FLAG-TDP1 followed by western blotting with the pS61-TDP1 antibody in the presence and absence of the CDK1 inhibitor RO3306, which further confirms that the catalytic activity of CDK1 is a prerequisite for TDP1-S61 phosphorylation. These new results have been included in the revised manuscript (new Fig 2H). Thank you.

2. The second main message of the manuscript is that TDP1-S61A gets trapped on mitotic chromatin, leading to DNA DSBs and genome instability, which triggers chromatin loading of MUS81 and excessive MiDAS. Addressing the following points will strengthen their conclusion.

a) Figure 1D: The authors state that expression of TDP1-S61A does not impact cell cycle progression. However, the cell cycle distribution for S61A looks very different than the WT cells at 1 and 5 hr release. Why are there more S61A cells in G0/G1 after 1 hr release than 5 hr release? Are the cells undergoing apoptosis? This needs to be clarified.

Answer: Thank you for insightful suggestion. We have now used a new mitotic synchronisation protocol using thymidine-nocodazole without RO3306 based on recommendation of other reviewers. Accordingly, we have performed new PI-RNase based FACS profile showing the cell cycle distribution for the WT and S61A expressing cells at various time points after release from thymidine-nocodazole block (new Fig 1E) which is included in the revised manuscript. Please note we did not obtain significant difference in the cell cycle distribution of S61A and the WT cells using the synchronisation protocol (Fig 1B). Thank you.

b) Figure 3D: The authors should provide representative images of untreated cells in each subphase of mitosis to allow direct comparison with RO3306-treated cells.

Answer: We have incorporated the representative images showing the sub-cellular localisation of endogenous TDP1 in the RO-3306 untreated cells (Fig 3F, left panel) for each sub-phase of mitosis to allow direct comparison with RO-3306-treated cells in the revised manuscript (Fig 3E-G). Thank you.

3. Statistical tests. The authors use t-tests for all of their comparisons. However, t-tests should not be used for multiple comparisons, such as those shown in Figs. 5C, 5G, 5H, 5I, 6E, 6I, 6L, 6K, 6M, 7E, and 7F.

Answer: We have changed the statistical t-tests to an ANOVA test for the concerned data in Figs. 5C, 5G, 5H, 5I, 6E, 6I, 6L, 6K, 6M, 7E, and 7F as suggested by the reviewer and have updated the corresponding figure legends accordingly in the revised manuscript. Thank you.

4. Literature review (page 5). The authors state incorrectly that MUS81 is phosphorylated by CDK1. After checking the works cited, Payliss et al show that SLX4 is phosphorylated by CDK1 and Palma et al show that MUS81 is phosphorylated by CK2.

Answer: We thank the reviewer for allowing us to correct the mistake. We have rectified the statement which now reads “*Intriguingly, CDK1-mediated phosphorylation of SLX4 which is a binding partner of MUS81 and CK2 mediated phosphorylation of MUS81 prevent unscheduled DNA cleavage by the endonuclease (Palma et al, 2018; Payliss et al, 2022).*” in our revised manuscript.

MINOR CONCERNS THAT SHOULD BE ADDRESSED

1. To improve the general clarity of this work, I recommend that the authors discuss their data in the same order that it appears in the figures (e.g., Figure XA, XB, XC, etc.). In some cases, they may need to modify the order of the data panels in the figure itself.

Answer: We have modified our revised manuscript as recommended by the reviewer.

2. The authors should cite PMID: PMC2504835, who also identified phosphorylation at TDP1 S61.

Answer: We have included the reference as recommended by the reviewer. Thank you.

3. While the introduction and the discussion were very well-written and easy to follow, I found the results section to be rather dense and difficult to follow in some places. This should be an easy fix with some additional editorial input and addressing the points listed below. Additionally, I would encourage the authors to remove all in-text references to the effect of "and the quantification in XX". These extra words are not needed.

Answer: We have modified our revised manuscript as recommended by the reviewer omitting “*and the quantification in XX*” extra words. Thank you.

4. Pages 8 and 21. The authors state that TDP1 pS61 peaked upon release from early mitosis and then declined in telophase. Given that this conclusion is based on western blots, which do not have the resolution to distinguish between the subphases of mitosis (e.g., anaphase and telophase), I strongly advise to replace 'telophase' with 'late mitosis'.

Answer: We have modified our current manuscript as recommended by the reviewer by replacing “telophase” with “late mitosis”. Thank you.

5. Page 9, first two sentences in section called "CDK1 binds TDPI to phosphorylate at S61". The authors state that they looked for a direct interaction between CDK1 and TDPI because CDK1 phosphorylates Top1. I don't follow their logic...was there a typo?

Answer: We have reframed the sentence "CDK1 plays a key role in the regulation of the mitotic DNA damage response (Brown *et al*, 2015; Diril *et al*, 2012; Holt *et al*, 2009) and it has been shown to phosphorylate Top1(Hackbarth *et al*, 2008)" Thank you.

6. Page 9. It is well-established that coIPs capture direct and indirect protein-protein interactions. As such, a coIP alone is not sufficient evidence for the authors to conclude that CDK1 binds directly to TDPI. While I agree that this is the most likely interpretation of their data as a whole, they should remove the word 'directly' in the sentence starting "This prompted us to ...".

Answer: We agree with the reviewer and have removed the word "directly" as recommended. Thank you.

7. Page 11, sentence starting "The endogenous pS61-TDPI was detected in ...". The quantification shown in Figure 3A does not match the data described in the text...?

Answer: Thank you, we have corrected the issue in the main text of our revised manuscript. The quantification is now shown in figure 3B

8. Page 11, sentence starting "We also detected a marked reduction of ...". The word 'cytoplasm' should be replaced with 'soluble fraction' (or similar) since the nuclear membrane breaks down in mitosis.

Answer: We have replaced "cytoplasm" with "soluble fraction" as suggested by the reviewer in our revised manuscript. Thank you.

9. Page 11, sentence starting "In contrast, endogenous TDPI was enriched ...". I don't understand what the authors are comparing in Figure 3A and 2F...Figure 2F does not show any chromatin fractionation blots. Please clarify/remove.

Answer: We have removed Fig 2F from the statement which now reads "In contrast, endogenous TDPI was enriched in the chromatin fraction (~4-fold; Fig 3A-B) in CDK1 knockdown cells, a situation mimicking defective TDPI phosphorylation at S61."

10. It seems unlikely that the pan-chromatin staining for TDPI-S61A in mitotic cells is solely due to trapping at CFSs. This should be discussed briefly in the manuscript, particularly with respect to what % of the genome contains CFSs. Related to this point, it would be useful to know the % of mitotic cells that showed the pan-chromatin staining for TDPI-S61A (I'm assuming that the quantification of intensity only represents chromatin containing TDPI-S61A signal).

Answer: We thank the reviewer for the insightful comments. We have performed additional new ChIP experiments in cells expressing TDP1^{WT} and TDP1^{S61A} to examine the enrichment of TDP1^{S61A} at the non-CFS loci including GAPDH, β -actin, β 2-microglobulin and β -tubulin (Fig 4M-P). We also detected CPT-induced enriched FLAG-TDP1^{S61A} at the four tested non-CFS, including β -actin (Fig 4M), GAPDH (Fig 4N), β 2-microglobulin (Fig 4O), and β -tubulin (Fig 4P), albeit to a significant lesser extent (Appendix Fig S4F) compared to the CFSs, which are intrinsically more susceptible to RS. The new data are corroborating with the pan-staining for TDP1-S61A in the mitotic chromosomes.

To answer the question *“Related to this point, it would be useful to know the % of mitotic cells that showed the pan-chromatin staining for TDP1-S61A (I'm assuming that the quantification of intensity only represents chromatin containing TDP1-S61A signal).”* we have performed new immunofluorescence experiments in cells ectopically expressing FLAG-TDP1^{S61A} mutant synchronised to mitosis and co-stained with anti-FLAG and anti-pHH3 (mitotic marker) antibodies to score the percentage of mitotic cells that showed pan-chromatin staining for TDP1-S61A. We found that about 95.33% of the FLAG-TDP1^{S61A} expressing mitotic cells had pan-staining for chromosome bound TDP1^{S61A} while 4.67 % formed foci for a total of 150 mitotic nuclei counted (n=3).

11. Page 15, bottom of the page. Replace "denaruring" with "denaturing".

Answer: We have replaced “denaruring” with “denaturing” in the revised manuscript.

12. Figure 4J-L. The data looks solid so it would be nice to statistically compare the WT and S61A for each condition.

Answer: We have included the statistical comparison between WT and S61A for each condition as recommended by the reviewer. Thank you.

13. *Figure 4K. I am not sure that the authors can confidently quantify foci in these images because of the lack of distinct particles. Perhaps total intensity is the better approach.*

Answer: We are unsure about the reviewer's question because Figure 4K displays quantification of the ChIP experiment of the fold enrichment of the FLAG-TDP1 variants and γ H2AX at the FDR of FRA3B loci.

14. *Figure 6K-M. I believe that the y-axis title should be "fold enrichment" as opposed to "MUS81".*

Answer: Thank you. We have now annotated the y-axis as ‘*fold enrichment*’ in our revised manuscript.

15. *"Western" should only be capitalized if it is the first word of the sentence.*

Answer: Thank you. We have modified our revised manuscript as suggested.

16. *It is a bit confusing to switch back and forth between CDK1i and RO3306. I would pick one and stick with that throughout the manuscript.*

Answer: Thank you for bringing that up; CDK1i is now substituted with "RO3306" throughout our amended manuscript.

17. *Page 23, first sentence of second paragraph. The authors acknowledge that the TDP1 NTD, which contains S61, is not required for in vitro catalytic activity (this should be acknowledged much sooner in the manuscript) and suggest that the inability of TDP1-S61A to repair Top1cc's is due to impaired interactions with TDP1 binding partners. It would be good to provide citations for work showing that XRCC1, PNKP, ligIII, and pol beta bind to the TDP1 NTD. If the binding interfaces are not known, then the authors should clarify that in the text.*

Answer: We thank you for the insightful suggestion. We have now added the lines “*The N-terminal region of TDP1 spanning 1–148 amino acids is not required for in vitro catalytic activity of TDP1 (Interthal et al, 2001) however, plays a critical role in subcellular localization, turnover, stability, recruitment of TDP1 at DNA damage sites, and interactions with its repair partners such as PARP1, XRCC1, and Ligase III in response to Top1cc-induced DNA damage (Bhattacharjee et al, 2022a; Bhattacharjee et al, 2022b; Das et al, 2009; Das et al, 2014; Hudson et al, 2012; Kawale & Povirk, 2018; Pommier et al, 2014; Rehman et al, 2018; Rashid, 2021)*” in the “Introduction” section of our revised manuscript.

ADDITIONAL SUGGESTIONS FOR IMPROVING THE STUDY (EDITOR'S DISCRETION)

1. *Figure 2: the authors provide good evidence for an interaction between CDK1 and TDP1, which is independent of CDK1 activity. This data would be strengthened substantially by a*

TDP1 mutant that cannot bind CDK1. Does S61A bind CDK1? Can they leverage AlphaFold Multimer to predict a high confidence model for CDK1 and TDP1?

Answer: We thank the reviewer for the constructive suggestion. We have performed a new immunoprecipitation experiment to test the binding of CDK1 with S61A-mutant TDP1 which is now incorporated in our revised manuscript (new Fig 2D) where we observed similar levels of endogenous CDK1 in the immune-complexes of both FLAG-TDP1 variants (WT and S61A) confirming the dispensable nature of pS61-TDP1 in mediating CDK1 and TDP1 interaction (Fig 2D).

However, using “*AlphaFold Multimer to predict a high-confidence model for CDK1 and TDP1*” is beyond the scope of our current work.

2. The authors have some interesting data regarding MUS81. Since these experiments were derived using siRNA, the conclusion could be further strengthened by repeating the key phenotypic experiments with siRNA-resistant MUS81 constructs (WT or nuclease-dead).

Answer: We thank the reviewer for the constructive suggestion; however, this is beyond the scope of our current work.

3. Since many of the experiments rely on over-expression of FLAG-TDP1, it would be useful for the authors to comment on the relative expression of FLAG-TDP1 to endogenous TDP1. They seem to have a good antibody for endogenous TDP1 so this should be a relatively easy experiment to perform. It's odd that it wasn't included in the manuscript.

Answer: We found the relative expression of FLAG-TDP1 to endogenous TDP1 is similar as detected by densitometric analysis using ImageJ software.

Referee #2:

The article from Chowdhuri and Das defines how the phosphorylation of TDP1 affects its localization and function in mitosis. TDP1 is an important protein to study because of its role in DNA repair and suppression of human disease. However, in its current form, the manuscript has a number of weaknesses and the design of many of the experiments is sub-optimal.

Specific Remarks

1. The title should be changed in my view because it doesn't accurately reflect the content of the manuscript. It implies that TDP1 plays a key role in mitotic progression, which is not really the case.

Answer: Thank you for the suggestion. We have changed the title in our revised manuscript, which now reads, “*TDP1 phosphorylation at S61 promotes Top1cc repair via mitotic DNA synthesis*”.

2. The opening sentence of the Abstract is an overstatement in my view. TOP1 is not really a key player in chromosome separation.

Answer: We have modified the opening sentence of the abstract as suggested by the reviewer which now reads “*Topoisomerase 1 (Top1) removes supercoils from DNA during replication and transcription, is critical for mitotic progression to the G1 phase, and ensures the maintenance of chromosome topology.*” in the revised manuscript.

3. It should be tested whether S61 phosphorylation of TDP1 is specific for mitosis by synchronizing cells in interphase (e.g. by using thymidine). If the residue is phosphorylated in interphase, CDK1 is unlikely to be the only kinase that can target this residue. Also, in Fig 1 it is stated that cells were synchronized to G2/M using nocodazole - this synchronizes cells in prometaphase.

Answer: We appreciate the reviewer bringing up the issue to test whether “*S61 phosphorylation of TDP1 is specific for mitosis*”. We have conducted new experiments in cells ectopically expressing FLAG-TDP1^{WT} and were synchronized to interphase by using thymidine or to mitosis by using thymidine-nocodazole (Appendix Fig S1G-I). FLAG-TDP1 was immunoprecipitated using anti-FLAG antibody separately from the interphase and mitotic synchronised cells (FACS profile in Appendix Fig S1H) and detected by western blotting with pS61-TDP1-specific antibody (Appendix Fig S1I). Detection of pS61-TDP1 was only restricted to mitosis and not in the interphase, corroborating with our previous results.

Further to support our results, we have also performed new *in vitro* kinase assays to test whether the TDP1-S61 residue is phosphorylated by CDK2 or CDK1, as suggested by reviewer 2 in Query # 7. Appendix Fig S2 shows that TDP1 is *not phosphorylated* at the S61 residue by CDK2 but is phosphorylated by CDK1; suggesting *S61 phosphorylation of TDP1 is specific for mitosis*.

In keeping with this observation, our manuscript has provided multiple lines of evidence to demonstrate CDK1-mediated TDP1-S61 phosphorylation, which are indicated below:

1. We have shown that TDP1 is phosphorylated at the S61 residue during mitosis (new Fig 1F) by synchronizing cells using the thymidine-nocodazole method.
2. Knocking down CDK1 abrogated TDP1-S61 phosphorylation (Fig 2G).
3. Additionally, we have now performed new immunoprecipitation experiments in the presence of CDK1-specific inhibitor (RO3306) that abrogates TDP1-S61 phosphorylation during mitosis (revised manuscript in Fig 2H).

Thank you for noting the typographical error that *cells were synchronized to G2/M using nocodazole*, which has been corrected to “*cells were synchronized to prometaphase using nocodazole*” in the revised manuscript.

4. It is not ideal to use RO2206 as a synchronizing agent when it targets the very protein that is under study - CDK1. One significant problem is that RO3306 affects the replication program in S-phase. Also, when cells are held at the G2/M boundary using this agent, CDK1 levels continue to rise. Hence, when cells are released, they show hyper-activation of CDK1.

Answer: As suggested by the reviewer we performed new experiments by following the new cell synchronization technique using thymidine-nocodazole synchronisation protocol without RO3306 (Fig 1B) which has now been incorporated in our revised manuscript (Fig 1E-F). Further, the sub-cellular localization of FLAG-TDP1 variants (WT, S61A and S61D) on the mitotic chromosome has also been repeated following the thymidine-nocodazole synchronisation protocol which is incorporated in the revised manuscript (Fig 3H-M).

Please note that we detected similar temporal kinetics of TDP1 phosphorylation at S61 and the chromosomal localisation of FLAG-TDP1 variants (WT and S61A) using both the cell synchronisation techniques as previously described in our manuscript (Di Marco *et al.*, 2017).

5. In panels G-H of Fig 1, it is claimed that pS61 was not detected in interphase cells - but some foci clearly are shown. It is also stated that this absence of pS61 is consistent with western blot analysis (Fig 1C) but that panel didn't study cells in interphase - only mitotic cells.

Answer: We have provided new images of the interphase cells that do not show any pS61-TDP1 foci in our revised manuscript (Fig 1G). Please note that we have also provided field images of asynchronously growing cells (Appendix Fig S1E-F), which show the pS61-TDP1 signal is restricted to the mitotic phase. In the same field, the interphase nucleus was devoid of the pS61-TDP1 signal, further pointing out the specificity of the antibody. The previously pointed-out red dots are plausibly due to the non-specific binding of the alexa-568 conjugated secondary antibody.

Additionally, we have performed a new experiment to detect pS61-TDP1 during interphase (Appendix Fig S1G-I). Cells ectopically expressing FLAG-TDP1^{WT} were thymidine-synchronized to harvest interphase cells, while thymidine-nocodazole

synchronization was used to harvest mitotic cells. FLAG-TDP1 was pulled down with an anti-FLAG antibody and blotted against the pS61-TDP1 antibody. We detected pS61-TDP1 only in the mitotic phase (Appendix Fig S1G-I).

6. In Fig 2, the use of benzonase in the co-IP doesn't prove that CDK1 binds to TDP1 directly - only that it doesn't require DNA to phosphorylate TDP1.

Answer: We agree with the reviewer and have amended the statement “.... indicating a protein-protein interaction, not mediated through DNA or RNA (Fig 2C)” of our current manuscript by “...indicating CDK1 mediated phosphorylation of TDP1 does not require DNA or RNA (Fig 2C).” in our revised manuscript.

7. The in vitro kinase assays are not very meaningful without a specificity control. CDK2 or other CDKs should be tested.

Answer: We thank the reviewer for raising the concern. We have now performed new *in vitro* kinase assay using CDK2 as a source of kinase with recombinant TDP1 (WT) as the substrate. The new Figure (Appendix Fig S2) shows that TDP1 is not phosphorylated at the S61 residue by CDK2 but is phosphorylated by CDK1 under similar condition.

8. On p11, it is stated that TDP1 defective for phosphorylation at S61 is enriched on mitotic chromosomes. It would be better to conclude that phosphorylation of TDP1 leads to the protein being excluded from mitotic chromosomes. Hundreds of proteins are excluded from condensed mitotic chromosomes and TDP1 seems to be another one of them. This isn't really a major finding. Also, on p11 it is stated that cells were 'synchronized to mitosis in the presence of RO3306'. This cannot be correct as RO3306 prevents cells entering mitosis.

Answer: We thank the reviewer for the suggestion. We have now replaced the statement “Therefore, we surmised that TDP1 phosphorylation at S61 leads to the exclusion of the protein from the mitotic chromosomes.” in our revised manuscript.

Also, we have changed the statement “MCF7 cells were synchronised to mitosis in the presence of RO3306 (Fig 3C)” in the revised manuscript following the reviewer’s suggestion, which now reads “MCF7 cells were synchronised to mitosis in the presence of nocodazole (Fig 3C)”.

9. The analysis of fragile sites lacks important controls. In particular, the analyses in Fig 4 of enrichment of TDP1-S61A at CFSs lacks analysis of any non-CFS locus. Given that the IF analysis shows TDP1-S61A staining all over mitotic chromosomes, it is inconceivable that its localization to CFSs is in any way specific for these loci. If it binds everywhere on mitotic chromosomes, it is debatable whether the CFS work should be included at all.

Answer: We thank the reviewer for the insightful suggestion. We have performed new ChIP experiments in the presence and absence of CPT to study the enrichment of FLAG-

TDP1^{S61A} at four non-fragile sites including β -actin (Fig 4M), GAPDH (Fig 4N), β 2 microglobulin (Fig 4O) and β -tubulin (Fig 4P) which is suggestive of the enrichment of the phosphodeficient TDP1 at non fragile sites which is increased in presence of CPT. Notably, our observations reveal that in the presence of CPT, FLAG-TDP1^{S61A} is also enriched at these four non-common fragile sites (β -actin, GAPDH, β 2 microglobulin, and β -tubulin) although to a lesser extent compared to common fragile sites which are more susceptible to replication stress (Appendix Fig S4F).

10. Why was BrdU and denaturation used to detect mitotic DNA synthesis rather than the commonly used EdU?

Answer: We thank the reviewer for raising the issue and allowing us to clarify that BrdU incorporation and denaturation techniques are widely used in our laboratory to study DNA replication. So, we improvised the technique to study MiDAS in mitotic cells.

We conducted new experiments to examine MiDAS by incorporating EdU in MUS81-depleted cells with ectopic expression of either EV or FLAG-TDP1 (WT or S61A), in the presence or absence of CPT (see Appendix Fig S6D-F). Importantly, we did not detect any significant variance in the incorporation or staining (utilizing antibody-based methods for BrdU and Click-iT reaction for EdU) between the two distinct thymidine analogues (BrdU and EdU) throughout the course of MiDAS. Hence, we have included both the recent (Appendix Fig S6D-F) and previous experiments involving BrdU incorporation (Fig 6A-F and Appendix Fig S6G-I) in our revised manuscript.

11. I was confused about some aspect of the MUS81 analysis. On p17 it is stated that knockdown of MUS81 affects 'resection of CPT-induced collapsed replication forks'. I couldn't see anywhere that resection had been analyzed and indeed it wasn't shown that the effects depended on the collapse of replication forks.

Answer: We have replaced “resection” with “repair” in our revised manuscript. We apologise.

12. The rescue of the effects of TDP1-S61A in MEFs by knockdown of MUS81 are very minor. Either this should be removed, or the text modified to acknowledge that the effects are small and could easily be an indirect effect of MUS81 being compromised in interphase and/or mitosis.

Answer: We are grateful to the reviewer for raising this concern. In response, we conducted new survival assays to validate the results and further explore the "rescue of the effects of TDP1-S61A in MEFs by knockdown of MUS81". The revised manuscript now includes the new data (new Fig 7J), and we have discussed that "*the rescue of the effects of TDP1-S61A in MEFs by knockdown of MUS81 is small but reproducible in independent experiments.*"

Minor Comment

On p9, line 2 'extruded' should read 'excluded'. In general, the English would benefit from some polishing.

Answer: We have replaced “extruded” with “*excluded*”. Thank you.

Referee #3:

Below I provide review of the manuscript by Choudhury and Das, entitled "TDP1 phosphorylation at S61 promotes mitotic progression and protects Top1 covalent complexes". The manuscript provides valuable insights into the role of TDP1 in genome maintenance, specifically highlighting the impact of its phosphorylation at S61 and its interaction with CDK1. This phosphorylation event leads to the dissociation of TDP1 from mitotic chromosomes, impacting mitotic progression under replication stress. The authors further explore the consequences of loss of this phosphorylation on S61, resulting in its accumulation on chromatin and triggering DNA damage response that requires MUS81-dependent mitotic DNA synthesis (MiDAS) mechanism. While the observations presented are intriguing, I would like to raise several points that need to be addressed to support publication of this manuscript in EMBO Journal.

Answer: We appreciate your acknowledgment for finding our “*manuscript provides valuable insights into the role of TDP1 in genome maintenance..... and we support the publication of this manuscript in the EMBO Journal*”. We have performed many new experiments to address your key concerns to strengthen our revised manuscript. Thank you.

Major Concerns:

1. The manuscript indicates TDP1 phosphorylation by CDK1, yet the use of CDK1 inhibitor RO3306 for cell synchronization in most experiments a potential confounding factor that could impact the overall validity of the authors' claims. To ensure the robustness of the findings and avoid any misinterpretation, I recommend considering Thymidine-Nocodazole mitosis phase synchronization (PMID: 28654080).

Answer: Thank you for your suggestion. We've implemented the new mitotic synchronization protocol using thymidine-nocodazole without RO3306, as per your recommendation. We repeated the immunoprecipitation experiments to examine the temporal kinetics of FLAG-TDP1 (WT and S61A) phosphorylation in FLAG-TDP1 expressing cells. This involved immunoprecipitating FLAG-TDP1 variants with an anti-FLAG antibody and conducting western blotting with a pS61-TDP1 antibody (new Fig 1B and 1E-F). Additionally, we included flow cytometry profiles by PI-RNase staining to show cell cycle distribution for WT and S61A expressing cells at various time points after release from thymidine-nocodazole block (new Fig 1E). Microscopic analysis for the sub-cellular localization of FLAG-TDP1 variants (WT, S61A) on the mitotic chromosome was also repeated following the thymidine-nocodazole synchronization protocol (new Fig 3H-M). Notably, we found no significant differences in the results compared to the previous synchronization protocol with used RO3306 (Di Marco *et al.*, 2017). Thank you.

2. While TDP1 phosphorylation and its interaction with CDK1 are not dependent on DNA damage (Fig.1A, p.7 and Fig. 2A, p. 9), the frequent use of CPT or other damaging agent may obscure the native effects of S61 phosphorylation. The authors are encouraged to perform key experiments under normal conditions. For example, the enrichment of S61 at CFS in untreated conditions is little if any, however CPT induces strong accumulation of S61. CPT clearly represent other mechanism of S61 retention on chromatin, dependent on replication and MUS81 (Fig. 6). CPT-induced TDP1S61 trapping thus represent replication obstacle that requires MUS81 processing, similar to recently described transcription replication collision (Chappidi *et al* 2020, and Matos *et al* 2020). This should be proper cited and discussed. Can TDP1 S61 stall replication forks (DNA fibers) and be involved in stabilizing R-loops (S9.6 and/or RNaseH1-ND staining)? Importantly, this seems to be Top1cc independent (Fig. 4G-I) thus representing novel role of TDP1?

Answer: Thank you for your suggestion. We agree with your comments and have included the following statement and the references in the discussion section of our revised manuscript. “CPT-induced retention of TDP1^{S61A} on mitotic chromosomes may cause replication obstacles that require MUS81 processing to prevent transcription replication collisions (Chappidi *et al* 2020; Matos *et al* 2020).” Please note that we have performed several key experiments to test the role of TDP1-S61 phosphorylation under no CPT conditions. Indeed, we detect retention of S61A mutant TDP1 on mitotic chromosomes without CPT treatment in independent experiments (Figure 1G-H and Figure 3; microscopy or Figure 4; ChIP experiments), suggesting that TDP1 S61-phosphorylation controls TDP1 dynamics during mitosis.

We have performed new immunofluorescence experiments to study inherent replication defects in the S-phase in TDP1^{-/-} MEFs complemented with FLAG-TDP1 variants (WT and S61A) using EdU incorporation. We failed to detect replication defects in these cells expressing TDP1 variants with no CPT treatment (Appendix Fig S6A-B). Please note that testing “*Can TDP1 S61 stall replication forks (DNA fibers) and be involved in stabilizing R-loops (S9.6 and/or RNaseH1-ND staining)?*” is beyond the scope of the current study. We have incorporated the new results into the revised manuscript. Thank you.

3. Considering the presented hypothesis, the inclusion (at least transiently) phosphomimetic mutant S61D would provide valuable insight and strengthen the conclusions.

Answer: As suggested we have generated phosphomimetic TDP1^{S61D} mutant and confirmed its cytosolic retention during mitosis in immunofluorescence experiments (New Fig 3K-M), and as expected TDP1^{S61D} mutant was defective in picking up pS61-TDP1 signal (Appendix Fig S3B). We have incorporated the new results in our revised manuscript and included the statement “*In keeping with pS61-TDP1, the phosphomimetic FLAG-TDP1^{S61D} was also delocalized from the chromosomes, emphasizing the role of S61 phosphorylation in ousting TDP1 from the chromosomes. (Fig 3K-M and Appendix Fig S3A-B)*” in the “Results” section of our revised manuscript. Thank you.

4. A thorough proofreading and significantly clarification of the text are recommended. Additionally, the relocation of some panels with confirmatory experiments to supplementary information would also improve the clarity.

Answer: Thank you for the suggestions.

Minor concerns:

- *Fig. 1 C and E: Include loading controls (e.g., Actin or GAPDH).*

Answer: We have included loading controls (β -actin) as suggested in Fig 1C and 1F in the revised manuscript. Thank you.

- *Fig. 1D: Clarify the observed difference in cell cycle progression. While the authors state that MCF7 cells expressing FLAG-TDP1 variants (WT or S61A) did not perturb the cell cycle progression (page 8), but there is a visible difference after 1 hour of release as shown in Fig. 1 D.*

Answer: We have performed new PI-RNase-based cell cycle analysis with the thymidine-nocodazole cell synchronisation (new Figure 1E), we don't observe significant difference in cell cycle progression with cells expressing TDP1 variants. Thank you.

Fig.4: Perform a proper statistical comparison between WT and S61 mutant in untreated conditions to conclude that TDP1 is excluded from the mitosis chromosomes. In addition, to confirm the role of MUS81 in processing CPT-trapped S61 mutant, an experiment (i.e. Fig. 4K) with siMUS81 should be performed.

Answer: We thank the reviewer for the suggestion. We have performed statistical comparison between the WT and S61A mutants in untreated conditions to conclude that TDP1 is excluded from the mitotic chromosomes. We have also performed new ChIP experiments under MUS81-depleted conditions, to confirm the role of MUS81 in processing the CPT-trapped S61 mutant which has now been included in new Appendix Fig S6C.

P.15 authors established that accumulation of S61 is not related to Top1cc, yet they try to link it to their processing by MIDAS. This is not logical and rather MUS81-mediated processing of transcriptionally stalled replication forks could be reffered to.

Answer: We have incorporated the suggested citations and explained the possibilities into the "Discussion" section of our revised manuscript. "CPT-induced TDP1^{S61} trapping thus represent replication obstacle that requires MUS81 processing, similar to recently described transcription replication collision (Chappidi et al 2020, and Matos et al 2020)." Thank you.

Fig. 6: Include Aphidicolin- only conditions as a control for MiDAS experiments (PMID: 36002001).

Answer: We have included aphidicolin-only conditions as a control for MiDAS experiments, as suggested by the reviewer in our revised manuscript (new Fig 6C and 6E-F). Under similar conditions, APH treatment shows a marked reduction in BrdU foci formation compared to cells treated with CPT. (Fig 6D-F, panel “APH”). We have also incorporated the suggested citation (PMID: 36002001)

- *Fig. 6B: Include a control WB for depletion of MUS81.*

Answer: We thank the reviewer for the suggestion. We would like to clarify that for Fig 6B we have not depleted MUS81. MUS81 depletion was performed for the experiments in Fig 6H-I (old) which are Fig 6I-M (new). So, we have included a control WB for MUS81 depletion in Fig 6J for new Fig 6I in the revised manuscript.

- *Fig. 6H: There is a lot of controversy in the literature about MUS81 foci and their reproducibility. Specify the antibody used for MUS81 foci and check the effect in other cell type (U2OS).*

Answer: We thank the reviewer for raising the concern. We have used *Anti-MUS81 antibody [MTA30 2G10/3] Cat# sc-53382; RRID: AB_2147138 at 1:300 dilution* antibody for immunofluorescence experiments. As suggested by the reviewer we have checked for MUS81 foci in U2OS cells using the same antibody and detected MUS81 foci in U2OS cells. Thank you.

- *Fig 7B: Provide details on the analysis of micronuclei and anaphase bridges from metaphase spreads?*

Answer: As recommended by the reviewer, we have now provided the details “*on the analysis of the anaphase bridges*” and *Analysis of Micronuclei* in the “Materials and Methods” section in our revised manuscript. Briefly:

Analysis of bulky anaphase bridges: Bulky anaphase bridges were detected as described previously (Boleslavskaya *et al.*, 2022; PMID: 36453994). Briefly, TDP1^{+/+} or TDP1^{-/-} MEFs transfected with EV or FLAG-TDP1 variants (WT or S61A) and/or SiMUS81 were grown on chamber slides were synchronized with 9 μ M RO-3306 for 16 h, followed by three washes with 1 \times PBS for 5 min at RT and subsequent incubation in DMEM for a total time of 2 h at 37°C. Cells were fixed with 4% (v/v) paraformaldehyde in PBS for 15 min at RT, followed by staining with DAPI. Cell images were acquired in a confocal microscope using a 63 \times /1.4 NA objective with oil immersion. The percentage of anaphase cells with bulky bridges was determined using ImageJ. At least 25 anaphase cells were scored per condition in each experiment.

Analysis of Micronuclei: analysis method has already been provided in the “Materials and methods” section of the current manuscript.

Briefly, TDP1^{+/+} or TDP1^{-/-} MEFs transfected with EV or FLAG-TDP1 variants (WT or S61A) and/or SiMUS81 were grown on chamber slides for analysis of micronuclei as described previously (Di Marco *et al.*, 2017). Cell culture medium was supplemented 16 hours

before fixation with 2 µg/ml cytochalasin B (Sigma-Aldrich) to block cells in cytokinesis. Cells were fixed with 4% formaldehyde for 10 min and mounted with Vectashield mounting medium containing DAPI. Slides were analysed by confocal fluorescent microscope. For quantification, only DAPI-stained binucleated cells were counted, and distinct micronuclei in the vicinity of these cells were considered as positive. At least 50 binucleated cells were scored in each experiment.

We would like to clarify that the micronuclei and anaphase bridges were not detected from the metaphase spreads. We have modified Fig 7A-C (old) accordingly in our amended manuscript to include individual quantifications for AB and MN (new Fig 7D-E).

- *P.19: In the text "MCF7 cells knockdown for MUS81 had no significant effect on CPT-induced cell survival when compared to wildtype counterpart (Fig 7H)." instead of Fig 7H it should be 7G. In addition, observed sensitization of TDP1 KO cells by siMUS81 to CPT have been recently also reported and should be properly referenced (Marini et al 2023).*

Answer: We apologize for the mistake. We have made the necessary corrections in the revised manuscript and included the reference as advised.

- *Consider investigating the effect of S61 phosphorylation on mitochondrial DNA and vice versa if TDP1 H493R mutant will show chromatin trapping phenotype during mitosis? <https://pubmed.ncbi.nlm.nih.gov/31723605/>*

Answer: We appreciate the reviewer's insightful inquiry, but addressing this question will require a separate study which is beyond the purview of our current work connecting TDP1-S61 and mitochondrial DNA. Thank you.

Dear Benu,

Thank you again for submitting your revised manuscript EMBOJ-2023-116166R for consideration by The EMBO Journal, and for your patience during peer review. The revised version of your manuscript has been seen by the three reviewers that had also assessed your original manuscript, and we have received the complete set of their comments, which I have already shared with you (they are included again below). I would also like to thank you for your detailed response to their comments, which was very helpful for us to reach a fair and balanced decision on your manuscript.

The referees recognize that this is an interesting study and that the revised manuscript has been significantly improved with the majority of the initially raised concerns having been adequately addressed. They also identify, however, a few remaining limitations and they provide a number of constructive suggestions for improving the study further before publication. Given the referees' positive comments and recommendations, as well as your responses to their remaining concerns, I would like to invite you to submit a final revised version of the manuscript, along with a detailed point-by-point response addressing all referees' comments. In particular:

1. Although we do agree with referee #1 that confirming on-target effects of MUS81 siRNA by using siRNA-resistant constructs would have strengthened the study further, in light of your response we will not request the addition of this data for publication of your manuscript in The EMBO Journal. However, we do request that you address this point in detail in your point-by-point response to the referees' comments and also mention this limitation in the Discussion of your revised manuscript.
2. Please consider revising the Title and the Abstract of your revised manuscript taking the advice of referee #2 to avoid overstatements on board.
3. The major point of referee #3 is interesting and relevant, but we will not request it to be addressed for the publication of this manuscript in The EMBO Journal. On the other hand, all minor points made by this referee must be completely addressed.

I should add that acceptance of your manuscript will depend on the completeness of your responses in this revised version. Please let me know if you have any questions or comments that you would like to discuss with me.

In addition to the aforementioned remaining referees' concerns, there are also a few minor changes and corrections that we need from you before we can proceed with acceptance of your manuscript:

- Please note that you can only list up to 5 keywords (you currently list 6).
- Please note that the literature citations in your References list must not be numbered; instead, they should be provided in alphabetical order, and "et al." should follow the names of the first 10 authors in case of publications with more than 10 co-authors. For more information on our reference format please visit:
<https://www.embopress.org/page/journal/14602075/authorguide#referencesformat>.
- Before submitting your revision, you are kindly requested to deposit the mass spectrometry data produced in this study in an appropriate public database (see also <https://www.embopress.org/page/journal/14602075/authorguide#dataavailability>). The accession number/identifier, database, and link/URL (that should resolve to a webpage where the data are publicly accessible) should be listed in the "Data availability" section of the manuscript.
- The author contributions statement should be removed from the manuscript file. Instead, we now use CRediT to specify the contributions of each author in the journal submission system. Please use the free text box to provide more detailed descriptions during submission. See also our guide to authors for more information:
<https://www.embopress.org/page/journal/14602075/authorguide#authorshipguidelines>.
- Please include in your resubmission a completed Author checklist, which you can download from our author guidelines (<https://www.embopress.org/page/journal/14602075/authorguide>). Please note that the checklist will also be part of the Peer Review File that will be published online along with your article. In the last column of this checklist, only the names of the manuscript sections where the relevant information can be found should be listed.
- Please move the Figure legends in your revised manuscript below the list of References.
- Please rename your supplementary information file to "Appendix" and move its brief Table of Contents (including page numbers) to the first page of the single PDF file.
- We noticed that Source Data for Fig. 1D, 1K, 1L, 5J, 6M and 6Q seem to be missing. Please include all requested Source Data in your resubmission. Furthermore, Source Data files need to be saved in a single zipped folder per Figure (for example, all Source Data files for the panels of Figure 1 need to be saved in a single folder that needs to be zipped and uploaded as "SD

Figure 1.zip" item). If you have Source Data for your EV and/or Appendix Figures, these can be zipped together in an EV/Appendix SD folder.

- Please note that we need your synopsis image in .jpg or .png format (it is currently in PDF format). Please remember the pixel dimensions of this image: it should be exactly 550 pixels wide and 300-600 pixels high (the height is variable within this range).
- Please upload your synopsis summary and bullet points in a separate Word file.
- Please define the annotated p values ***/**/* as well as provide the exact p-values for the same in the legend of Figure 7c-e, j; as appropriate.
- Please note that the exact p values are not provided in the legends of Figures 3b, d; 4d-f, j-p; 5b, d, f, h-i; 6e-f, h, k-l; 7g-h.
- Please indicate the statistical test used for data analysis in the legends of Figures 7c-e, i-j.
- Please note that in Figures 6n-p; 7g-h; there is a mismatch between the annotated p values in the figure legend and the annotated p values in the figure file that should be corrected.
- Please note that information related to "n" is missing in the legends of Figures 1h; 7c-e, i-j.
- Although "n" is provided, please describe the nature of entity for "n" in the legends of Figures 5b, d, f, h-i; 6e-f, h, k-l, n-p; 7g-h.
- Please note that the error bars are not defined in the legends of Figures 7c-e.
- Please note that the scale bar needs to be defined in the legend of Figure 7f.

Please also note that as part of the EMBO publications' Transparent Editorial Process, The EMBO Journal publishes online a Peer Review File along with each accepted manuscript. This File will be published in conjunction with your paper and will include the referee reports, your point-by-point response and all pertinent correspondence relating to the manuscript. You can opt out of this by letting the editorial office know (contact@embojournal.org). If you do opt out, the Peer Review File link will point to the following statement: "No Peer Review File is available with this article, as the authors have chosen not to make the review process public in this case."

We look forward to seeing a final version of your manuscript as soon as possible. Please use this link to submit your revision: <https://emboj.msubmit.net/cgi-bin/main.plex>

Best regards,

Ioannis

Referee #1:

This is a resubmission of an interesting and exciting study that provides us with new molecular insights into the post-translational modifications of TDP1 and the role of TDP1 in genome stability maintenance. In brief, the authors identify a conserved residue in vertebrate TDP1 (S61) that is phosphorylated by mitotic CDK1 (presumably CDK1-cyclin B). Through the use of a phospho-alanine mutant (S61A), they provide data indicating that the phosphorylation of TDP1 promotes its dissociation from mitotic chromatin to prevent excessive accumulation of MUS81 and activation of mitotic DNA synthesis (MiDAS), which ultimately manifests as genome instability. They further propose that CPT-induced DNA breaks in condensed mitotic chromosomes are refractory to canonical TDP1-dependent pathways and rely on error-prone repair through MiDAS.

In general, the manuscript is well-written and the data is presented in a coherent and well-controlled manner. The main findings

are thought-provoking and will have important implications for researchers interested in the molecular mechanisms of MiDAS and chromosome stability, particularly with respect to the roles of TDP1 and the cellular pathways that respond to the accumulation of Top1cc's.

The authors have addressed all of my major concerns. Although I believe that it is important to confirm on-target effects of siRNA through the use of 'rescue experiments', the authors did not include this in the revised version. I will leave this to the editor's decision. My original comment is provided below.

- The authors have some interesting data regarding MUS81. Since these experiments were derived using siRNA, the conclusion could be further strengthened by repeating the key phenotypic experiments with siRNA-resistant MUS81 constructs (WT or nuclease-dead).

Referee #2:

The revised manuscript from the Das group has been substantially improved by the incorporation of new experimental data. The majority of my original comments have now been addressed adequately.

In general, therefore, I support publication of the manuscript in EMBO J. However, I am still concerned that, because of a tendency to overstate some findings and conclusions, the impact of the study could be compromised. The style of the writing in the new Discussion section is measured and accurately reflects what has been shown definitively and what has not. Unfortunately, this is not the case for the new Title and the Abstract section. I would strongly urge the authors to focus more on the fact that TDP1 phosphorylation promotes a MUS81-dependent repair process in mitosis - rather than stating categorically that this process is MiDAS. While it is possible that MiDAS is indeed the key process involved, right now the evidence is circumstantial and represent little more than 'guilt by association' in my view. Just because MUS81 is needed for both MiDAS and TOP1-related repair events, doesn't mean that these two events must proceed via the same pathway.

Referee #3:

While the authors have address some of the comments the major concern remains unaddressed:

Point 2): Authors provide evidence of S61 phosphorylation, but the physiological relevance and DNA damage induced mechanism are not satisfactorily explained. While there is only slight enrichment of S61A at both CFS and non-CFS at non damaged conditions by ChIP (Fig.4), massive accumulation of S61A is observed by microscopy (Fig. 3J and 5A). Only mild increase in H2AX is observe, which however is mostly comparable to empty vector control.

While they do not report CPT-induced S61 phosphorylation, authors see CPT-dependent accumulation of S61A on chromatin by ChIP compared to WT, with correspond massive accumulation of H2AX on chromatin, which however is only slightly increased compared to WT when staining with antibodies in microscopy. In general, many of the phenotypes are partially suppressed by WT expression, indicating different levels of expression compared to endogenous TDP1 and making the interpretation even more difficult.

Even though authors do not show correlation with Top1cc accumulation (Fig 4G-I and Fig. 5G and I), its levels are even decreased for S61A compared to empty vector control, they still interpret their data in context of role of TDP1 phosphorylation in removal of Top1cc. The question thus remains: What are the structures that require Mus81 in S61A cells? Are these stalled replication forks due to transcription associated R-loops?

Minor points:

- It would really help, if the authors would label the changes in the manuscript as well as individual figures.
- Authors need to be very careful not to overinterpret their data:
 - A) Authors claim to see enrichment of S61A at CFSs, however this is not supported by the data presented, rather represent chromatin wide association.
 - B) P.15, MUS81 dependence is not addressed in this section.
 - C) P.17, the authors summarize that that mitotic Top1ccs are repaired primarily by endonuclease pathways but no direct evidence to support this claim is provided.
 - D) Fig. 7 authors claim that CPT treatment markedly increased micronuclei, anaphase bridges and chromatid breaks in S61A cells compared to EV or WT cells. However, no significance is shown to support this claim, and is clearly not true for anaphase bridges (panel E).
- In response to point 1) the authors refer to Fig. 3H-M, however there are no panels expect 3H.
- The quality of IF for MUS81 are not good and seem quite unspecific (see staining in S61A cells), comparison to other

antibodies or tagged line would be required.

- Appendix Fig. S6C with enrichment of TDP S61A at the FDR loci needs to be compared between control and siMUS81 to claim the effect on MUS81 depletion.

- Authors claim that MUS81 depletion significantly reduced BrdU or EdU foci (Fig. 6I-L), but it seems to be provided in Fig. S6D-I.

Point-by Point answers to referee's question and editorial suggestions**Referee# 1**

This is a resubmission of an interesting and exciting study that provides us with new molecular insights into the post-translational modifications of TDPI and the role of TDPI in genome stability maintenance. In brief, the authors identify a conserved residue in vertebrate TDPI (S61) that is phosphorylated by mitotic CDK1 (presumably CDK1-cyclin B). Through the use of a phospho-alanine mutant (S61A), they provide data indicating that the phosphorylation of TDPI promotes its dissociation from mitotic chromatin to prevent excessive accumulation of MUS81 and activation of mitotic DNA synthesis (MiDAS), which ultimately manifests as genome instability. They further propose that CPT-induced DNA breaks in condensed mitotic chromosomes are refractory to canonical TDPI-dependent pathways and rely on error-prone repair through MiDAS.

In general, the manuscript is well-written and the data is presented in a coherent and well-controlled manner. The main findings are thought-provoking and will have important implications for researchers interested in the molecular mechanisms of MiDAS and chromosome stability, particularly with respect to the roles of TDPI and the cellular pathways that respond to the accumulation of Top1cc's.

The authors have addressed all of my major concerns. Although I believe that it is important to confirm on-target effects of siRNA through the use of 'rescue experiments', the authors did not include this in the revised version. I will leave this to the editor's decision. My original comment is provided below.

- The authors have some interesting data regarding MUS81. Since these experiments were derived using siRNA, the conclusion could be further strengthened by repeating the key phenotypic experiments with siRNA-resistant MUS81 constructs (WT or nuclease-dead).

Answer: We would like to bring to your attention that the new concern raised by Reviewer# 1 was neither part of the original major concerns nor minor concerns in the referee report. This concern was separately listed under “additional suggestion (Editor's discretion).” We clearly pointed out to Dr. Hartmut Vodermaier that the siRNA sequence we used against MUS81 provides a substantial knockdown of MUS81 and is widely used in multiple publications. Specifically, the role of MUS81 in the repair of trapped Top1cc during mitosis has been substantiated in multiple experiments in our revised manuscript listed in Fig 6G-P and Appendix Fig S6C-I. Therefore, we believe the additional suggestion to confirm the on-target siRNA effects for the MUS81 experiments is not necessary and will not change the conclusion provided in the manuscript. Moreover, these additional experiments will take us additional time (3-4 months) as we don't have the necessary resources available in our laboratory.

Referee #2

The revised manuscript from the Das group has been substantially improved by the incorporation of new experimental data. The majority of my original comments have now been addressed adequately.

In general, therefore, I support publication of the manuscript in EMBO J. However, I am still concerned that, because of a tendency to overstate some findings and conclusions, the impact of the study could be compromised. The style of the writing in the new Discussion section is measured and accurately reflects what has been shown definitively and what has not. Unfortunately, this is not the case for the new Title and the Abstract section. I would strongly urge the authors to focus more on the fact that TDPI phosphorylation promotes a MUS81-dependent repair process in mitosis - rather than stating categorically that this process is MiDAS. While it is possible that MiDAS is indeed the key process involved, right now the evidence is circumstantial and represent little more than 'guilt by association' in my view. Just because MUS81 is needed for both MiDAS and TOP1-related repair events, doesn't mean that these two events must proceed via the same pathway.

Answer: We wish to thank Reviewer #2 for recognizing that our revised manuscript has been "substantially improved by the incorporation of new experimental data" and for supporting its publication in the EMBO Journal. MiDAS is an established mechanism for break-induced repair (BIR) that operates in early mitosis to rescue under-replicated loci, involving key players such as POLD3, RAD52, and the structure-specific endonuclease MUS81. We would like to clarify that we have provided data supporting MUS81-mediated MiDAS (Fig. 6A-F and Appendix Fig. 6D-I) in our revised manuscript by performing BrdU/EdU incorporation in both MUS81 proficient and deficient conditions.

However, based on the Editors' suggestion, we have now changed our title "***TDPI phosphorylation at S61 promotes Top1cc repair via the MUS81-dependent pathway during mitosis***" and incorporated the necessary changes in the abstract marked in red in our final manuscript. Thank you.

Referee #3

While the authors have address some of the comments the major concern remains unaddressed: Point 2): Authors provide evidence of S61 phosphorylation, but the physiological relevance and DNA damage induced mechanism are not satisfactorily explained. While there is only slight enrichment of S61A at both CFS and non-CFS at non damaged conditions by ChIP (Fig.4), massive accumulation of S61A is observed by microscopy (Fig. 3J and 5A). Only mild increase in H2AX is observe, which however is mostly comparable to empty vector control.

While they do not report CPT-induced S61 phosphorylation, authors see CPT-dependent accumulation of S61A on chromatin by ChIP compared to WT, with correspond massive accumulation of H2AX on chromatin, which however is only slightly increased compared to WT when staining with antibodies in microscopy. In general, many of the phenotypes are

partially suppressed by WT expression, indicating different levels of expression compared to endogenous TDP1 and making the interpretation even more difficult.

Even though authors do not show correlation with Top1cc accumulation (Fig 4G-I and Fig. 5G and I), its levels are even decreased for S61A compared to empty vector control, they still interpret their data in context of role of TDP1 phosphorylation in removal of Top1cc. The question thus remains: What are the structures that require Mus81 in S61A cells? Are these stalled replication forks due to transcription associated R-loops?

Answer: The differences in TDP1S61A enrichment observed between the microscopy-based and ChIP experiments can be attributed to variations in cell lines and experimental techniques. In the microscopic experiments (Figs 3J and 5A), TDP1^{-/-} MEFs were complemented with ectopic expression of ^{EV}, TDP1^{WT}, or TDP1^{S61A}, followed by immunocytochemistry with an anti-FLAG antibody. However, this approach lacks the precision to determine specific sites of chromosomal enrichment or binding intensities.

In contrast, ChIP experiments offer greater precision in detecting TDP1^{S61A} enrichment at specific gene loci in MCF7 cells before and after treatment. We observed a marked increase in FLAG-TDP1^{S61A} enrichment at CFSs after CPT treatment (Fig 4J-L) and additional DNA damage and γ H2AX enrichment (Fig 4). We also detected CPT-induced FLAG-TDP1S61A enrichment at four non-CFSs, including β -actin, GAPDH, β 2-microglobulin, and β -tubulin, though to a lesser extent than at CFSs (Appendix Fig S4F).

Additionally, we generated a phosphomimetic TDP1S61D mutant and confirmed its cytosolic retention during mitosis (New Fig 3I-K), and this mutant was defective in recognizing the pS61-TDP1 signal (Appendix Fig S3B). This further supports our findings, which are not limited to microscopic data. The physiological relevance of S61-TDP1 phosphorylation is evident under replication stress, which induces genomic instability and mitotic defects.

MUS81 is implicated in repairing trapped Top1cc, which generates DNA breaks and γ H2AX (PMID 22123861). We conducted new MiDAS experiments incorporating EdU in MUS81-depleted cells and found that MUS81 knockdown significantly reduced both EdU and γ H2AX foci (Appendix Fig S6D-I). This suggests a role for MUS81-mediated MiDAS in clearing trapped Top1ccs and TDP1S61A. Additionally, new ChIP experiments under MUS81-depleted conditions confirmed MUS81's role in processing CPT-trapped S61 mutants (Appendix Fig S6C).

In response to the question regarding the structures requiring MUS81 in S61A cells and whether these are stalled replication forks due to transcription-associated R-loops, we conducted immunofluorescence experiments to study inherent replication defects in TDP1^{-/-} MEFs complemented with FLAG-TDP1 variants (WT and S61A) using EdU incorporation. We did not detect replication defects in these cells without CPT treatment (Appendix Fig S6A-B). Testing whether TDP1 S61 stalls replication forks or is involved in stabilizing R-loops is beyond the scope of this study. This question requires an independent investigation, as we discussed with Dr. Hartmut Vodermaier initially.

Minor points:

Point-by Point answers to referee's question and editorial suggestions

- It would really help, if the authors would label the changes in the manuscript as well as individual figures.

- Authors need to be very careful not to overinterpret their data:

Answer: We have marked red all the changes in the final revised manuscript as suggested.

A) Authors claim to see enrichment of S61A at CFSs, however this is not supported by the data presented, rather represent chromatin wide association.

Answer: ChIP experiments offer greater precision in detecting TDP1^{S61A} enrichment at specific gene loci in MCF7 cells before and after treatment. We observed a marked increase in FLAG-TDP1^{S61A} enrichment at CFSs after CPT treatment (Fig 4J-L) and additional DNA damage and γ H2AX enrichment (Fig 4). We also detected CPT-induced FLAG-TDP1^{S61A} enrichment at four non-CFSs, including β -actin, GAPDH, β 2-microglobulin, and β -tubulin, though to a lesser extent than at CFSs (Appendix Fig S4F). We have incorporated the figure number in the final manuscript as suggested.

B) P.15, MUS81 dependence is not addressed in this section.

Answer: This changed title succinctly communicates the key points of your findings is now incorporated in our final manuscript. “*Mitotic Top1ccs are repaired by MUS81-dependent mitotic DNA synthesis (MiDAS) independent of TDP1*” to address the concern. Thank you.

C) P.17, the authors summarize that that mitotic Top1ccs are repaired primarily by endonuclease pathways but no direct evidence to support this claim is provided.

Answer: We have now modified the statement “*Our study further suggests that mitotic Top1ccs are repaired primarily by MUS81-dependent endonuclease pathways*”. This conclusion is based on our observations that TDP1, the key enzyme for Top1cc repair, is not recruited to the condensed mitotic DNA (Fig 3), canonical HR is inactive outside the S-G2 phases, and previous reports indicate MUS81-endonuclease pathways are involved in Top1cc repair (PMID: 34272385; PMID: 22123861; PMID: 35869071).” Thank you.

D) Fig. 7 authors claim that CPT treatment markedly increased micronuclei, anaphase bridges and chromatid breaks in S61A cells compared to EV or WT cells. However, no significance is shown to support this claim, and is clearly not true for anaphase bridges (panel E).

Answer: We have added the significance to compare “*that CPT treatment markedly increased micronuclei, anaphase bridges and chromatid breaks in S61A cells compared to EV or WT cells*” in our final manuscript. We have modified the figure and figure legends accordingly. We have also removed “anaphase bridges” from the main text in P18.

Point-by Point answers to referee's question and editorial suggestions

- In response to point 1) the authors refer to Fig. 3H-M, however there are no panels expect 3H.

Answer: We would like to clarify that we have referred to Fig. 3H-M (according to the revised manuscript) in response reviewer point 1, which was raised based on our initial submission. Please note that the revised Fig. 3H-M has been performed using the new thymidine-nocodazole synchronization protocol.

- The quality of IF for MUS81 are not good and seem quite unspecific (see staining in S61A cells), comparison to other antibodies or tagged line would be required.

Answer: We thank the reviewer for the concern. We have now replaced the complete set of experiments (*Appendix Fig S6D-F*) by incorporating better representative images and their corresponding quantifications in our revised manuscript.

- Appendix Fig. S6C with enrichment of TDP S61A at the FDR loci needs to be compared between control and siMUS81 to claim the effect on MUS81 depletion.

Answer: Thank you for your suggestions. We have now incorporated the modifications into Appendix Figure S6C, which shows the comparison between *siMUS81* and *control*, highlighting the differential enrichment of TDP S61A at the FDR loci.

- Authors claim that MUS81 depletion significantly reduced BrdU or EdU foci (Fig. 6I-L), but it seems to be provided in Fig. S6D-I.

Answer: We have incorporated the necessary correction in our revised final final manuscript. Thank you.

Answer to Editorial comments

1. Please note that you can only list up to 5 keywords (you currently list 6).

Answer: We have removed “Replication stress” from the list and now have the following five key words in the final manuscript.

2. Please note that the literature citations in your References list must not be numbered; instead, they should be provided in alphabetical order, and "et al." should follow the names of the first 10 authors in case of publications with more than 10 co-authors. For more information on our reference format please visit:

<https://www.embopress.org/page/journal/14602075/authorguide#referencesformat>

Answer: We have formatted the literature citations in the References as per *The EMBO Journal* guidelines in our final manuscript.

3. Before submitting your revision, you are kindly requested to deposit the mass spectrometry data produced in this study in an appropriate public database (see also <https://www.embopress.org/page/journal/14602075/authorguide#dataavailability>). The accession number/identifier, database, and link/URL (that should resolve to a webpage where the data are publicly accessible) should be listed in the "Data availability" section of the manuscript.

Answer: The MS/MS spectrum of a peptide chromatogram for TDP1-S61 phosphorylation, derived from the FLAG-TDP1 immunoprecipitation, is presented in Appendix Fig. S1A. This figure illustrates the fragmentation pattern of the peptide sequence with b ions and y ions detected through MS analysis. The FLAG-TDP1 IP and MS analysis is previously published (Rehman et al, 2018) from our laboratory. We have not shown any FLAG-TDP1 interactome in the current manuscript. Please note that we regret to inform you that we are currently unable to deposit the mass spectrometry data in a public database. This is because the TDP1-interaction data remains confidential as it is part of ongoing, unpublished research. We appreciate your understanding and patience in this matter.

4. The author contributions statement should be removed from the manuscript file. Instead, we now use CRediT to specify the contributions of each author in the journal submission system. Please use the free text box to provide more detailed descriptions during submission. See also our guide to authors for more information:

<https://www.embopress.org/page/journal/14602075/authorguide#authorshipguidelines>

Answer: We have removed the author contributions statement from the manuscript file. Instead, we are now using the CRediT taxonomy to specify the contributions of each author in the journal submission system.

Point-by Point answers to referee's question and editorial suggestions

5. Please include in your resubmission a completed Author checklist, which you can download from our author guidelines (<https://www.embopress.org/page/journal/14602075/authorguide>). Please note that the checklist will also be part of the Peer Review File that will be published online along with your article. In the last column of this checklist, only the names of the manuscript sections where the relevant information can be found should be listed.

- Please move the Figure legends in your revised manuscript below the list of References.

Answer: We have defined the error bars in the revised figure legend for Fig 7C-E and included the statement in the figure legend, highlighted in red.

- Please rename your supplementary information file to "Appendix" and move its brief Table of Contents (including page numbers) to the first page of the single PDF file.

Answer: We have renamed the supplementary information file to "Appendix" and moved its brief Table of Contents (including page numbers) to the first page of the single PDF file.

- We noticed that Source Data for Fig. 1D, 1K, 1L, 5J, 6M and 6Q seem to be missing. Please include all requested Source Data in your resubmission. Furthermore, Source Data files need to be saved in a single zipped folder per Figure (for example, all Source Data files for the panels of Figure 1 need to be saved in a single folder that needs to be zipped and uploaded as "SD Figure 1.zip" item). If you have Source Data for your EV and/or Appendix Figures, these can be zipped together in an EV/Appendix SD folder.

Answer: We apologize for the fact that the source data for Fig 1D has been wrongly uploaded as 1C. But there are no such figures as 1K, 1L, 5J and 6Q so no source data are present. 6M is a schematic protocol for which there is no source data.

- Please note that we need your synopsis image in .jpg or .png format (it is currently in PDF format). Please remember the pixel dimensions of this image: it should be exactly 550 pixels wide and 300-600 pixels high (the height is variable within this range).

Answer: we have uploaded the synopsis in the defined format and .jpg file.

- Please upload your synopsis summary and bullet points in a separate Word file.

Answer: We have uploaded our synopsis summary and bullet points in a separate Word file.

- Please define the annotated p values ***/**/* as well as provide the exact p-values for the same in the legend of Figure 7c-e, j; as appropriate.

Answer: We have followed the conventional design of The EMBO journal representation pattern and incorporated the following statement: "Asterisks denote statistically significant differences. Data are mean \pm SD, n = 3 biological replicates. *P \leq 0.1; **P \leq 0.01; ***P \leq

Point-by Point answers to referee's question and editorial suggestions

0.001 (one-way ANOVA).” This statement is marked in red in our manuscript. Additionally, we have uploaded the Excel file in the source data.

- Please note that the exact p values are not provided in the legends of Figures 3b, d; 4d-f, j-p; 5b, d, f, h-i; 6e-f, h, k-l; 7g-h.

Answer: We have followed the conventional design of The EMBO journal representation pattern and incorporated the following statement: “Asterisks denote statistically significant differences. Data are mean \pm SD, n = 3 biological replicates. *P \leq 0.1; **P \leq 0.01; ***P \leq 0.001 (one-way ANOVA).” This statement is marked in red in our manuscript. Additionally, we have uploaded the Excel file in the source data. **As suggested, we have incorporated the exact p values in the figures.**

- Please indicate the statistical test used for data analysis in the legends of Figures 7c-e, i-j.

Answer: We have provided the statistical test used for data analysis in the legends of Figures as well in the figures 7c-e, i-j.

- Please note that in Figures 6n-p; 7g-h; there is a mismatch between the annotated p values in the figure legend and the annotated p values in the figure file that should be corrected.

Answer: We have rectified the mismatch between the annotated p values in the figure legend and the annotated p values in the figure file of the final manuscript marked in red.

- Please note that information related to "n" is missing in the legends of Figures 1h; 7c-e, i-j.

Answer: We have now provided the information related to “n” in the legends of Figures 1h; 7c-e, i-j marked in red.

- Although "n" is provided, please describe the nature of entity for "n" in the legends of Figures 5b, d, f, h-i; 6e-f, h, k-l,n-p; 7g-h.

Answer: We have described the nature of identity for “n” in the legends of Figures 5b, d, f, h-i; 6e-f, h, k-l,n-p; 7g-h marked in red.

- Please note that the error bars are not defined in the legends of Figures 7c-e.

Answer: We have defined the error bars in the figure legend for Fig 7C-E.

- Please note that the scale bar needs to be defined in the legend of Figure 7f.

Answer: We have defined the scale bar in the figure legend for Fig 7F marked in red.

Dear Benu,

Thank you again for addressing the majority of our previous requests in the new revised version of your manuscript. We kindly ask you to address the following two remaining issues in a final version before we can proceed with its acceptance for publication in The EMBO Journal:

- Before submitting your revision, please deposit all primary datasets that are linked to your study in appropriate public databases (for more information see also our guide: <https://www.embopress.org/page/journal/14602075/authorguide#dataavailability>). The accession numbers/identifiers, databases, and links/specific URLs (that should resolve to the webpages where the data are publicly accessible) should be listed in the "Data availability" section of the manuscript.

- Please update your Source Data Checklist, and provide the Source Data as a single ZIP folder for each Figure (for example, all Source Data files for the panels of Figure 1 should be zipped together in a folder named "SD_Figure_1.zip"). If you also have Source Data to upload for Expanded View (EV) and/or Appendix Figures, please zip them together in a single folder.

We look forward to seeing a final version of your manuscript as soon as possible. Please use this link to submit your revision: <https://emboj.msubmit.net/cgi-bin/main.plex>

Best regards,

Ioannis

All editorial and formatting issues were resolved by the authors.

Dear Benu,

Congratulations on an excellent manuscript, I am very pleased to inform you that it has been accepted for publication in The EMBO Journal. Thank you for your comprehensive responses to the referee concerns and for the smooth collaboration!

If you have any questions, please do not hesitate to contact the Editorial Office. Thank you for your contribution to The EMBO Journal. It has been a pleasure working with you!

Best wishes,

Ioannis
